# AC-Sampler: Accelerate and Correct Diffusion Sampling with Metropolis-Hastings Algorithm

**Minsang Park**[1][*] **Gyuwon Sim**[1], **Hyungho Na**[2], **Jiseok Kwak**[1], **Sumin Lee**[1],
**Richard Lee Kim**[1], **Dongyhyeok Shin**[1], **Byeonghu Na**[1], **Yeongmin Kim**[1], **Il-Chul Moon**[1,3][*]
[1]KAIST, [2]UNIST, [3]summary.ai

## Abstract

Diffusion-based generative models have recently achieved state-of-the-art performance in high-fidelity image synthesis. These models learn a sequence of denoising transition kernels that gradually transform a simple prior distribution into a complex data distribution. However, requiring many transitions not only slows down sampling but also accumulates approximation errors. We introduce the Accelerator-Corrector Sampler (AC-Sampler), which accelerates and corrects diffusion sampling without fine-tuning. It generates samples directly from intermediate timesteps using the Metropolis–Hastings (MH) algorithm while correcting them to target the true data distribution. We derive a tractable density ratio for arbitrary timesteps with a discriminator, enabling computation of MH acceptance probabilities. Theoretically, our method yields samples better aligned with the true data distribution than the original model distribution. Empirically, AC-Sampler achieves FID 2.38 with only 15.8 NFEs, compared to the base sampler's FID 3.23 with 17 NFEs on unconditional CIFAR-10. On CelebA-HQ 256×256, it attains FID 6.6 with 98.3 NFEs. AC-Sampler can be combined with existing acceleration and correction techniques, demonstrating its flexibility and broad applicability. Our code is available at https://github.com/aailab-kaist/AC-Sampler.

## 1 Introduction

Diffusion-based generative models (Ho et al., 2020; Sohl-Dickstein et al., 2015; Song et al., 2021a) have become one of the most popular approaches in recent years due to their strong ability to generate diverse types of data such as high-fidelity images (Dhariwal & Nichol, 2021; Rombach et al., 2022) and videos (Ho et al., 2022; Voleti et al., 2022). Building on these models, strong pre-trained variants have emerged (Rombach et al., 2022; Karras et al., 2022) In spite of many variations, these models share a fundamental structure: they start by sampling from a simple prior distribution and iteratively transform the samples through a series of learned transition kernels to approximate the complex data distribution. This iterative generation causes two problems. First, the sampling process is slow due to the large number of kernel transition calculations required (Song et al., 2021b; Zhang & Chen, 2023). Second, errors can accumulate during the sampling process if the kernel transition does not accurately reflect the true reverse diffusion process (Xu et al., 2023b).

Speed and accuracy are usually considered separate research topics, as improving both simultaneously is often challenging. Some approaches for acceleration diffusion sampling (Kim & Ye, 2023; Zheng et al., 2023) focus on reducing the NFE while maintaining image quality, but they lack theoretical analysis for converging to the true data distribution, leaving model errors unaddressed. On the contrary, previous correction methods (Kim et al., 2023; Na et al., 2024; Xu et al., 2023b) makes additional computational burden, which makes the methods unscalable in real-world services.

To address both challenges, we propose **Accelerator-Corrector Sampler (AC-Sampler)**. It accelerates and corrects the diffusion sampling process without any fine-tuning of the pre-trained model. AC-Sampler directly proposes samples at intermediate timesteps, which enables acceleration.

---

[*]Corresponding authors: Minsang Park <pagemu@kaist.ac.kr>, Il-Chul Moon <icmoon@kaist.ac.kr>

Using Metropolis-Hastings correction, these proposals are guaranteed to theoretically follow the true marginal distribution. Since the pre-trained model approximates the score function, we can construct an effective proposal distribution using Langevin dynamics (Grenander & Miller, 1994). For computing the acceptance probability, we only train a time-dependent discriminator (Kim et al., 2023; Na et al., 2024), which can be learned at a much lower cost than the diffusion model itself.

Real-world services (xAI, 2025; Midjourney, Inc., 2025; Microsoft, 2025) and diverse downstream tasks (Hiranaka et al., 2025; Trabucco et al., 2024; Azizi et al., 2023) typically require generating multiple samples quickly and accurately. Our method meets this demand by producing multiple samples that are both faster and more accurate, effectively overcoming the limitations of previous methods while offering the flexibility to be integrated with various sampling methods.

Our contributions are as follows:

- We propose the **AC-Sampler**, which accelerates diffusion sampling by generating samples from intermediate timesteps rather than the initial prior distribution. It also corrects the accumulated error in the sampling process with Metropolis-Hastings (MH) algorithms.

- We provide a theoretical analysis showing that training the discriminator and following the MH chain in our method leads to a tighter bound on the data distribution compared to that of a pre-trained diffusion model. Furthermore, we provide a theoretic analysis of the expected reduction in the number of function evaluations (NFE).

- We validate our theoretical claim through experiments on benchmark datasets and toy settings, and demonstrate that our method applies effectively to diverse pre-trained models in both unconditional and conditional settings.

- Our contribution is orthogonal to advances in training-free samplers, so the two gains are complementary and can be realized simultaneously. Also, the utilize discriminator is simple and does not require ad-hoc structures on the pre-trained diffusion model.

## 2 RELATED WORK

**Acceleration Methods**    To reduce the computational burden associated with additional training, acceleration methods have been developed to speed up the sampling process without modifying the original diffusion model. DDIM (Song et al., 2021a) reformulates the reverse diffusion as a deterministic ODE, achieving significant speedups with fewer steps while preserving the pre-trained network. Building on this foundation, various works have further improved ODE solvers through high-order numerical methods and exponential integration, leading to significant gains in sampling efficiency (Liu et al., 2022; Lu et al., 2022a;b; Dockhorn et al., 2022a; Karras et al., 2022; Zhang & Chen, 2023; Zhao et al., 2023; Zheng et al., 2023; Tong et al., 2025).

In parallel, other lines of work have explored fundamentally different perspectives on diffusion model acceleration. For example, PDS (Ma et al., 2022) treats diffusion sampling as an Markov Chain Monte Carlo (MCMC) process, incorporating frequency-domain preconditioning to improve high-frequency details. DLG (Kim & Ye, 2023) formulates the sampling process over the product space of data and time, enabling joint Langevin-based Gibbs sampling. This sampling process identifies the intermediate perturbed data with low noise for the initialization of the reverse process, which shortens the subsequent diffusion trajectory. However, as shown in Appendix A.1, DLG lacks theoretical convergence guarantees, leaving room for improvement.

**Correction Methods**    Several studies have been conducted to improve the sampling quality of pre-trained diffusion models. DG (Kim et al., 2023) proposes a correction method using a time-dependent discriminator when score estimation is inaccurate, thereby improving the accuracy of the transition kernel. Restart (Xu et al., 2023b) theoretically demonstrates that repeating forward and backward steps within a fixed time interval $[t_{\min}, t_{\max}]$ in a pre-trained model can reduce sampling error. DiffRS (Na et al., 2024) aims to sample from the true distribution by applying a rejection sampling scheme with a time-dependent discriminator. ES (Ning et al., 2024) proposes a training-free correction schedule to compensate for the scale gap in score norms between the training and sampling phases. However, there is no theoretical guarantee that simply matching the norms leads to distributional equivalence. While these methods focus on sampling correction, they do not reduce the base NFE and may even increase it, leading to slower sampling.

## 3 PRELIMINARY

### 3.1 METROPOLIS-HASTINGS ALGORITHM AND LANGEVIN PROPOSAL

The Metropolis-Hastings algorithm (MH algorithm) (Metropolis et al., 1953; Hastings, 1970) is a MCMC method used to sample from a target distribution when direct sampling is not possible. It constructs a Markov chain whose stationary distribution is the target distribution by satisfying the detailed balance condition. Based on this condition, the algorithm defines an acceptance probability, which is used to determined whether to accept proposals drawn from a simple proposal distribution.

Given a target distribution $q(\cdot)$ and a proposal distribution $p_{\text{proposal}}(\cdot \mid \cdot)$, the acceptance probability $\alpha$ for a proposed sample $\tilde{\mathbf{x}}$ and current sample $\mathbf{x}$ is

$$\alpha = \min\left(1, \frac{q(\tilde{\mathbf{x}})\, p_{\text{proposal}}(\mathbf{x} \mid \tilde{\mathbf{x}})}{q(\mathbf{x})\, p_{\text{proposal}}(\tilde{\mathbf{x}} \mid \mathbf{x})}\right),$$

which guarantees that $q(\cdot)$ is the stationary distribution of the Markov chain.

To improve mixing and convergence of MH algorithm, gradient-based proposals have been studied (Parisi, 1988; Neal et al., 2011), i.e. the Langevin proposal. Specifically, the proposal is derived from the Euler–Maruyama discretization of the overdamped Langevin dynamics (Roberts & Tweedie, 1996). Formally, with the target distribution $q(\cdot)$, the Langevin proposal is defined as

$$p_{\text{proposal}}(\cdot|\mathbf{x}) := \mathcal{N}\left(\mathbf{x} + \frac{\eta}{2}\nabla_{\mathbf{x}}\log q(\mathbf{x}), \eta\mathbf{I}\right),$$

where $\eta$ is the step size and $\nabla_{\mathbf{x}}\log q(\mathbf{x})$ is the score function of $q(\mathbf{x})$. This method is commonly referred to as the Metropolis-Adjusted Langevin Algorithm (MALA) (Grenander & Miller, 1994).

### 3.2 DIFFUSION MODELS

Diffusion models (Ho et al., 2020; Song et al., 2021b) are probabilistic generative models that approximate data distributions by adding noise and reversing this process. They consist of a forward process that corrupts data into noise and a reverse process that removes noise to generate samples.

Let $q_0(\mathbf{x}_0)$ denote the true data distribution and $p_0^{\boldsymbol{\theta}}(\mathbf{x}_0)$ denote the distribution of generated samples from the model. In particular, the forward process is a fixed Markov chain where Gaussian noise is added using a pre-defined variance schedule. This creates a sequence of random variables $\mathbf{x}_{1:T}$:

$$q_{1:T|0}(\mathbf{x}_{1:T}|\mathbf{x}_0) := \prod_{t=1}^{T} q_{t|t-1}(\mathbf{x}_t|\mathbf{x}_{t-1}), \tag{1}$$

where each $q_{t|t-1}(\mathbf{x}_t|\mathbf{x}_{t-1})$ is a Gaussian transition with increasing noise levels. This process transforms the data distribution into a tractable prior distribution $p_T(\mathbf{x}_T)$ as $t \to T$. The reverse process is modeled as a Gaussian distribution that denoises a $p_T(\mathbf{x}_T)$ iteratively:

$$p_{0:T}(\mathbf{x}_{0:T}) := p_T(\mathbf{x}_T) \prod_{t=1}^{T} p_{t-1|t}^{\boldsymbol{\theta}}(\mathbf{x}_{t-1}|\mathbf{x}_t), \tag{2}$$

where $p_{t-1|t}^{\boldsymbol{\theta}}$ is the transition kernel that generate data from prior distribution $p_T$. The model is trained by maximizing a variational bound on the log-likelihood of the data. In practice, this can be achieved via denoising score matching loss (Vincent, 2011; Ho et al., 2020), given by:

$$\mathcal{L}_{\text{DSM}}(\boldsymbol{\theta}) = \mathbb{E}_{t \sim U[0,1], \mathbf{x}_0 \sim q(\mathbf{x}_0), \mathbf{x}_t \sim q(\mathbf{x}_t|\mathbf{x}_0)}\left[\left\|\mathbf{s}^{\boldsymbol{\theta}}(\mathbf{x}_t, t) - \nabla_{\mathbf{x}_t}\log q_{t|0}(\mathbf{x}_t|\mathbf{x}_0)\right\|_2^2\right], \tag{3}$$

With this loss, $\mathbf{s}^{\boldsymbol{\theta}}(\mathbf{x}_t, t)$ optimizes to follow the true score $\nabla_{\mathbf{x}_t}\log q_t(\mathbf{x}_t)$. The transition kernel is parameterized as a Gaussian distribution whose mean is a function of the score function:

$$p_{t-1|t}^{\boldsymbol{\theta}}(\mathbf{x}_{t-1}|\mathbf{x}_t) := \mathcal{N}(\mathbf{x}_{t-1}; \boldsymbol{\mu}_t(\mathbf{x}_t, \mathbf{s}^{\boldsymbol{\theta}}(\mathbf{x}_t, t)), \sigma_t^2\mathbf{I}), \tag{4}$$

where $\sigma_t^2\mathbf{I}$ denotes time-dependent variance. After training, samples are generated by iteratively applying the transition kernel from $t = T$ to $t = 0$.

We assume that we have access to a pre-trained diffusion model, denoted by $\boldsymbol{\theta}$. Let $q_t(\mathbf{x}_t)$ and $p_t^{\boldsymbol{\theta}}(\mathbf{x}_t)$ denote the marginal distributions at timestep $t$, defined by forward diffusion processes starting from

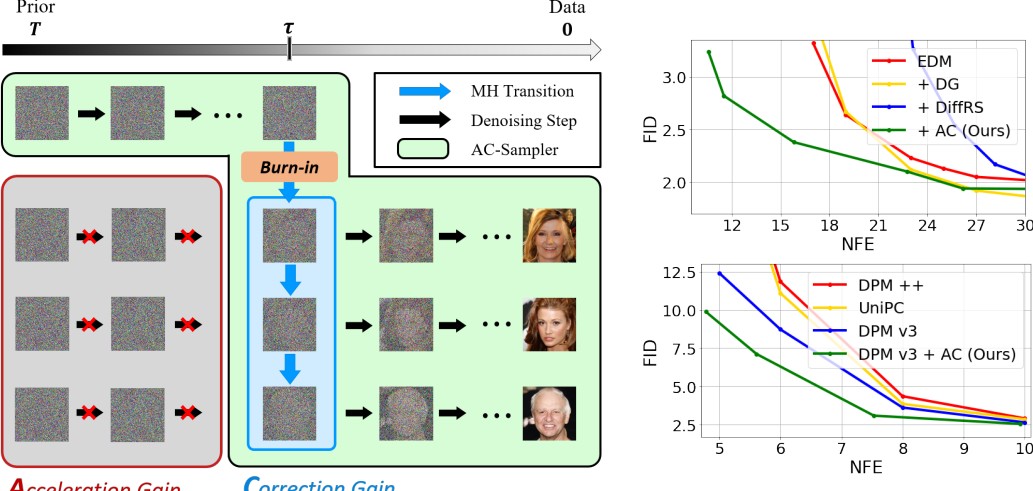

Figure 1: Overall figure of AC-Sampler.

Figure 2: FID–NFE graph on uncond. CIFAR-10: (Top) Correction methods (Bottom) Acceleration methods.

$q_0(\mathbf{x}_0)$ and $p_0^{\boldsymbol{\theta}}(\mathbf{x}_0)$, respectively. Pre-trained diffusion model provides $\mathbf{s}^{\boldsymbol{\theta}}(\mathbf{x}_t, t) \approx \nabla_{\mathbf{x}_t} \log q_t(\mathbf{x}_t)$. The mean of the transition kernel depends on both $\mathbf{x}_t$ and the score function $\mathbf{s}^{\boldsymbol{\theta}}(\mathbf{x}_t, t)$. Therefore, we treat the score function and the transition kernel $p_{t-1|t}^{\boldsymbol{\theta}}(\mathbf{x}_{t-1}|\mathbf{x}_t)$ as equivalent parametrizations, since both are derived from the same model and same input $\mathbf{x}_t, t$.

Due to this structure, diffusion models are inherently limited by slow sampling speed from the large number of transition steps, and by the accumulation of approximation errors in the transition kernels.

## 4 METHOD

To address the two key challenges previously discussed, we propose **AC-Sampler**, a novel diffusion sampling framework. The overall sampling procedure is as follows. First, we perform denoising from the prior distribution up to a target timestep $\tau$, which serves as the initial sample of the MCMC chain. Starting from this initial sample, we repeatedly generate new candidates using a score-based proposal distribution. At each step, MH correction is applied. These steps are performed as described in Algorithm. 1. After sufficient burn-in period, the resulting samples are corrected to true marginal distribution, $q_\tau$. Finally, each accepted sample is further denoised to obtain the final outputs. The overall sampling process is illustrated in Fig. 1.

This approach has two advantages. First, samples are created directly at $\tau$ without denoising from $T$. This enables faster sampling (denoted as *Acceleration Gain* in Fig. 1). Second, because of MH correction, the resulting samples follow a true marginal distribution $q_\tau$. (denoted as *Correction Gain* in Fig. 1). Our method accelerates and corrects the sampling process without requiring any fine-tuning of the underlying diffusion model. For this process, we require the design of the proposal distribution (Sec. 4.1) and the computation of the acceptance probability (Sec. 4.2).

### 4.1 PROPOSAL DISTRIBUTION

To sample from the intermediate timestep $t$, we use MALA. With target distribution $q_t(\mathbf{x}_t)$, we construct the proposal distribution using pre-trained score model $\mathbf{s}^{\boldsymbol{\theta}}(\mathbf{x}_t, t)$ as:

$$p_{\text{proposal},t}^{\boldsymbol{\theta}}(\cdot|\mathbf{x}_t) = \mathcal{N}\left(\mathbf{x}_t + \frac{\eta}{2}\mathbf{s}^{\boldsymbol{\theta}}(\mathbf{x}_t, t), \eta\mathbf{I}\right), \tag{5}$$

where $\eta$ is Langevin step size. We adaptively set the value of $\eta$ to maintain a constant signal-to-noise ratio (SNR) during sampling (Song & Ermon, 2019; Song et al., 2021b). An important advantage of our framework is that both the denoising step and the proposal distribution rely on the same score value. As a result, a single network evaluation is sufficient for both operations, enabling efficient integration of denoising and exploration within AC-Sampler.

---

**Algorithm 1** `MALAOneStep`

---

**Input:** Target timestep $\tau$, Previous sample $\mathbf{x}_\tau$, Score output $\mathbf{s} := \mathbf{s}^{\boldsymbol{\theta}}(\mathbf{x}_\tau, \tau)$,
    Likelihood ratio $L_\tau^\phi := \frac{d^\phi(\mathbf{x}_\tau, \tau)}{1 - d^\phi(\mathbf{x}_\tau, \tau)}$, Score network $\mathbf{s}^{\boldsymbol{\theta}}$, Discriminator $d^\phi$
**Output:** Next sample $\tilde{\mathbf{x}}_\tau$
 1: **repeat**
 2:    Propose $\tilde{\mathbf{x}}_\tau$ from proposal distribution $p_{\text{proposal},\tau}^{\boldsymbol{\theta}}(\cdot \mid \mathbf{x}_\tau)$ (Eq. 5)
 3:    Get score $\tilde{\mathbf{s}} \leftarrow \mathbf{s}^{\boldsymbol{\theta}}(\tilde{\mathbf{x}}_\tau, \tau)$, and likelihood ratio $\tilde{L}_\tau^\phi \leftarrow \frac{d^\phi(\tilde{\mathbf{x}}_\tau, \tau)}{1 - d^\phi(\tilde{\mathbf{x}}_\tau, \tau)}$
 4:    Calculate acceptance probability $\alpha \leftarrow \hat{\alpha}(\mathbf{x}_\tau, \tilde{\mathbf{x}}_\tau, \mathbf{s}, \tilde{\mathbf{s}}, L_\tau^\phi, \tilde{L}_\tau^\phi)$ (Eq. 9)
 5:    Sample $u \sim \mathcal{U}(0, 1)$
 6: **until** $u < \alpha$
 7: **return** $\tilde{\mathbf{x}}_\tau, \tilde{\mathbf{s}}, \tilde{L}_\tau^\phi$

---

### 4.2 ACCEPTANCE PROBABILITY

The acceptance probability for target distribution $q_t(\cdot)$ is $\alpha = \min\left(1, \frac{q_t(\tilde{\mathbf{x}}_t)p_{\text{proposal},t}^{\boldsymbol{\theta}}(\mathbf{x}_t|\tilde{\mathbf{x}}_t)}{q_t(\mathbf{x}_t)p_{\text{proposal},t}^{\boldsymbol{\theta}}(\tilde{\mathbf{x}}_t|\mathbf{x}_t)}\right)$, where
$\tilde{\mathbf{x}}_t$ is a sample from the proposal distribution $p_{\text{proposal},t}^{\boldsymbol{\theta}}(\cdot|\mathbf{x}_t)$. To make the acceptance probability tractable, we first decompose $q_t(\tilde{\mathbf{x}}_t)/q_t(\mathbf{x}_t)$ as stated in the following theorem.

**Theorem 4.1.** *Let $\mathbf{x}_t$ and $\tilde{\mathbf{x}}_t$ be two arbitrary samples at diffusion timestep $t$. Then, for any fixed $\mathbf{x}_{t-1}$, the density ratio of the true marginal distribution $q_t$ is given by:*

$$\frac{q_t(\tilde{\mathbf{x}}_t)}{q_t(\mathbf{x}_t)} = \frac{q_{t|t-1}(\tilde{\mathbf{x}}_t \mid \mathbf{x}_{t-1})}{q_{t|t-1}(\mathbf{x}_t \mid \mathbf{x}_{t-1})} \cdot \frac{L_t(\tilde{\mathbf{x}}_t, t)}{L_t(\mathbf{x}_t, t)} \cdot \frac{p_{t-1|t}^{\boldsymbol{\theta}}(\mathbf{x}_{t-1} \mid \mathbf{x}_t)}{p_{t-1|t}^{\boldsymbol{\theta}}(\mathbf{x}_{t-1} \mid \tilde{\mathbf{x}}_t)}, \tag{6}$$

*where $L_t(\mathbf{x}_t, t) := q_t(\mathbf{x}_t)/p_t^\theta(\mathbf{x}_t)$ denotes the likelihood ratio between the data and model marginal distributions at timestep $t$.*

Proof is provided in Appendix A.2. Let $\boldsymbol{\mu}_t(\mathbf{x}_t, \mathbf{s}^{\boldsymbol{\theta}}(\mathbf{x}_t, t))$ denote the mean of the reverse transition kernel $p_{t-1|t}^{\boldsymbol{\theta}}$. It is possible to choose $\hat{\mathbf{x}}_{t-1} := \frac{1}{2}\left(\boldsymbol{\mu}_t(\mathbf{x}_t, \mathbf{s}^{\boldsymbol{\theta}}(\mathbf{x}_t, t)) + \boldsymbol{\mu}_t(\tilde{\mathbf{x}}_t, \mathbf{s}^{\boldsymbol{\theta}}(\tilde{\mathbf{x}}_t, t))\right)$, since Theorem 4.1 holds for arbitrary $\mathbf{x}_{t-1}$, With this choice, the transition kernel terms in Eq. 6 are $\frac{p_{t-1|t}^{\boldsymbol{\theta}}(\hat{\mathbf{x}}_{t-1}|\mathbf{x}_t)}{p_{t-1|t}^{\boldsymbol{\theta}}(\hat{\mathbf{x}}_{t-1}|\tilde{\mathbf{x}}_t)}$, which cancel in the density ratio because both are Gaussian with the same variance and $\hat{\mathbf{x}}_{t-1}$ is equidistant from their means. With chosen $\hat{\mathbf{x}}_{t-1}$, acceptance probability as follows:

$$\alpha = \min\left(1, \frac{q_{t|t-1}(\tilde{\mathbf{x}}_t \mid \hat{\mathbf{x}}_{t-1})}{q_{t|t-1}(\mathbf{x}_t \mid \hat{\mathbf{x}}_{t-1})} \cdot \frac{L_t(\tilde{\mathbf{x}}_t, t)}{L_t(\mathbf{x}_t, t)} \cdot \frac{p_{\text{proposal},t}^{\boldsymbol{\theta}}(\mathbf{x}_t|\tilde{\mathbf{x}}_t)}{p_{\text{proposal},t}^{\boldsymbol{\theta}}(\tilde{\mathbf{x}}_t|\mathbf{x}_t)}\right) \tag{7}$$

To access $L_t(\mathbf{x}_t, t)$, we use time-dependent discriminator $d^\phi$, following the approach of DG (Kim et al., 2023). The discriminator is trained to distinguish between $q_t$ and $p_t^{\boldsymbol{\theta}}$ at all timesteps. To achieve this, weighted binary cross-entropy loss is used for training the discriminator:

$$\mathcal{L}_{\text{BCE}}(\boldsymbol{\phi}) = \int \lambda(t) \left[\mathbb{E}_{\mathbf{x}_t \sim q_t}[-\log d^\phi(\mathbf{x}_t, t)] + \mathbb{E}_{\mathbf{x}_t \sim p_t^\theta}[-\log(1 - d^\phi(\mathbf{x}_t, t))]\right] dt, \tag{8}$$

The optimal discriminator satisfies $d^{\phi^*}(\mathbf{x}_t, t) = \frac{q_t(\mathbf{x}_t)}{q_t(\mathbf{x}_t) + p_t^\theta(\mathbf{x}_t)}$, so the density ratio $\frac{q_t(\mathbf{x}_t)}{p_t^\theta(\mathbf{x}_t)}$ becomes $\frac{d^{\phi^*}(\mathbf{x}_t, t)}{1 - d^{\phi^*}(\mathbf{x}_t, t)}$. Having access to $\frac{q_t(\mathbf{x}_t)}{p^\theta(\mathbf{x}_t)} \approx \frac{d^\phi(\mathbf{x}_t, t)}{1 - d^\phi(\mathbf{x}_t, t)} =: L_t^\phi(\mathbf{x}_t, t)$, the acceptance probability can be calculated as below:

$$\hat{\alpha}(\mathbf{x}_t, \tilde{\mathbf{x}}_t, \mathbf{s}, \tilde{\mathbf{s}}, L, \tilde{L}) = \min\left(1, \underbrace{\frac{q_{t|t-1}(\tilde{\mathbf{x}}_t \mid \hat{\mathbf{x}}_{t-1})}{q_{t|t-1}(\mathbf{x}_t \mid \hat{\mathbf{x}}_{t-1})}}_{\text{Forward term}} \cdot \underbrace{\frac{\tilde{L}}{L}}_{\text{Likelihood ratio}} \cdot \underbrace{\frac{p_{\text{proposal},t}^{\boldsymbol{\theta}}(\mathbf{x}_t|\tilde{\mathbf{x}}_t)}{p_{\text{proposal},t}^{\boldsymbol{\theta}}(\tilde{\mathbf{x}}_t|\mathbf{x}_t)}}_{\text{Proposal term}}\right) \tag{9}$$

where $\mathbf{s}, \tilde{\mathbf{s}}, L, \tilde{L}$ denotes $\mathbf{s}^{\boldsymbol{\theta}}(\mathbf{x}_t, t), \mathbf{s}^{\boldsymbol{\theta}}(\tilde{\mathbf{x}}_t, t), L_t^{\phi}(\mathbf{x}_t, t), L_t^{\phi}(\tilde{\mathbf{x}}_t, t)$, respectively. The acceptance probability consists of three terms. The forward and proposal terms are tractable Gaussian distributions, and the likelihood ratio is computed using a discriminator. Together, these components make the acceptance probability fully tractable. Note that with any tractable proposal distribution, the acceptance probability also remains tractable.

## 4.3 THEORETICAL ANALYSIS

We prove that our method can theoretically achieve sampling *acceleration* and *correction*.

**Proposition 4.2.** *Let the reverse diffusion process consist of a total number of timesteps $T$, and let the target timestep in AC-Sampler be $\tau$. Suppose that $l$ denotes the length of the Markov chain at timestep $\tau$. Let $NFE_R$ denote the average reduction in NFE per sample achieved by AC-Sampler.*

*If the acceptance probability $\alpha$ satisfies $\alpha > \frac{1}{T-\tau+1}$, then, as $l \to \infty$, the expected NFE reduction remains strictly positive, while its variance converges to zero:*

$$\lim_{l \to \infty} \mathbb{E}[NFE_R] > 0, \quad \lim_{l \to \infty} \mathrm{Var}(NFE_R) = 0. \tag{10}$$

In practice, it is intractable to set $l$ to infinity. However, we experimentally verified that a significant reduction in the NFE can be achieved even with a finite $l$. This demonstrates that with score-guided proposal distribution, the aforementioned condition $\alpha > \frac{1}{T-\tau+1}$ is not a hard constraint. Further discussion of the acceptance probability is provided in Appendix C.

To show that our method not only accelerates sampling but also corrects errors, we theoretically demonstrate that the data distribution induced by our sampler is closer to the true data distribution than that of the baseline model.

**Theorem 4.3.** *Let $p_0^{\boldsymbol{\theta}}, p_0^{\boldsymbol{\theta},\phi^*}$ denote the model distribution and refined distribution by AC-Sampler with optimal discriminator $\phi^*$, respectively. Then, the KL divergence between the true data distribution $q_0$ and the refined distribution $p_0^{\boldsymbol{\theta},\phi^*}$ is bounded by:*

$$D_{KL}(q_0(\mathbf{x}_0)||p^{\boldsymbol{\theta},\phi^*}(\mathbf{x}_0)) \le D_{KL}(q_0(\mathbf{x}_0)||p^{\boldsymbol{\theta}}(\mathbf{x}_0)) \tag{11}$$

**Theorem 4.4.** *Let $T_\tau$ be the transition kernel of MALA at timestep $\tau$. Also, $p_\tau^{\boldsymbol{\theta},\phi^*,(l)}$ denotes marginal distribution at timestep $\tau$ after the $l$-th MALA transition from $p_\tau^{\boldsymbol{\theta}}$, and $p_0^{\boldsymbol{\theta},\phi^*,(l)}$ denotes the data distribution generated from $p_\tau^{\boldsymbol{\theta},\phi^*,(l)}$ with denoising transition kernel $p_{t-1|t}^{\boldsymbol{\theta}}$. $\mathcal{L}^p$ denotes a space of function which satisfies $(\int_{\mathbb{R}} |f|^p d\mathbf{x})^{\frac{1}{p}} < \infty$. If $q_\tau(\mathbf{x}_\tau) \in \mathcal{L}^\alpha, \log\left(\frac{p_\tau^{\boldsymbol{\theta},\phi^*,(l)}(\mathbf{x}_\tau)}{q_\tau(\mathbf{x}_\tau)}\right) \in \mathcal{L}^\beta, T_\tau \in \mathcal{L}^\gamma$, where $\alpha, \beta, \gamma \in [1, \infty]$ satisfy $\frac{1}{\alpha} + \frac{1}{\beta} + \frac{1}{\gamma} = 1$, then the KL divergence between the true data distribution $q_0$ and the refined distribution $p_0^{\boldsymbol{\theta},\phi^*,(l+1)}$ is bounded by:*

$$D_{KL}(q_0(\mathbf{x}_0)||p_0^{\boldsymbol{\theta},\phi^*,(l+1)}(\mathbf{x}_0)) \le D_{KL}(q_0(\mathbf{x}_0)||p_0^{\boldsymbol{\theta},\phi^*,(l)}(\mathbf{x}_0)) \tag{12}$$

The proofs of each theoretical result and detailed analysis are provided in Appendix A.4 and A.5, respectively. When our model has sufficiently converged, Theorem 4.3 suggests that it can generate samples that are closer to the true data distribution than those produced by the baseline model. Theorem 4.4 further shows that applying more MALA steps progressively moves the samples closer to the true data distribution. Note that $l = 0$ denotes the base diffusion model, without MALA.

**Mitigating Empirical Distribution Distortion** Although the theoretical results above are derived under the standard MH formulation, where rejected proposals are retained as the next state, this assumption is suboptimal in practical generative sampling. In a continuous sample space, observing the exact same point twice is a probability-zero event; such repetition would mathematically require the distribution to contain a Dirac delta mass, which is incompatible with the smooth target density $q_\tau$. Thus, repeated states arise only as an artifact of rejection, not as a reflection of the underlying distribution $q_\tau$. Storing these duplicated states leads to a noticeably distorted empirical distribution

Table 1: Performance on unconditional CIFAR-10 generation. Values that are better compared to the baseline are highlighted in bold.

| Model | FID↓ | NFE↓ |
|---|---|---|
| *Unconditional Generation* | | |
| VDM (Kingma et al., 2021) | 7.41 | 1000 |
| DDPM (Ho et al., 2020) | 3.17 | 1000 |
| iDDPM (Nichol & Dhariwal, 2021) | 2.90 | 1000 |
| DDIM (Song et al., 2021a) | 4.16 | 100 |
| ScoreSDE (Song et al., 2021b) | 2.20 | 2000 |
| Soft Truncation (Kim et al., 2022b) | 2.33 | 2000 |
| STF (Xu et al., 2022) | 1.90 | 35 |
| CLD-SGM (Dockhorn et al., 2022b) | 2.25 | 312 |
| INDM (Kim et al., 2022a) | 2.28 | 2000 |
| LSGM (Vahdat et al., 2021) | 2.10 | 138 |
| PFGM++ (Xu et al., 2023c) | 1.93 | 35 |
| PSLD (Pandey & Mandt, 2023) | 2.10 | 246 |
| Flow Matching (Lipman et al., 2023) | 6.35 | 142 |
| Rectified Flow (Liu et al., 2023) | 2.58 | 127 |
| ES (Ning et al., 2024) | 1.95 | 35 |
| EDM (Heun) (Karras et al., 2022) | 2.01 | 35 |
| **EDM (Heun) + AC (Ours)** | **1.97** | **26.19** |
| DDO (Heun)(Zheng et al., 2025) | 1.42 | 35 |
| **DDO (Heun) + AC (Ours)** | **1.41** | **29.41** |
| *Acceleration Method* | | |
| DPM ++ (Lu et al., 2022a) | 24.54 | 5 |
| UniPC (Zhao et al., 2023) | 23.52 | 5 |
| DPM-v3 (Zheng et al., 2023) | 12.41 | 5 |
| **DPM-v3 + AC (Ours)** | **9.88** | **4.78** |
| *Correction Method* | | |
| Restart (Xu et al., 2023b) | 1.95 | 43 |
| DiffRS (Na et al., 2024) | 2.02 | 30.73 |
| DG (Kim et al., 2023) | 1.93 | 27 |
| **DG + AC (Ours)** | **1.84** | **26.19** |

Table 2: Performance on unconditional CIFAR-10 generation with (Top) correction and (Bottom) acceleration methods.

| Method | FID↓ | NFE↓ | FID↓ | NFE↓ | FID↓ | NFE↓ |
|---|---|---|---|---|---|---|
| EDM (Heun) | 2.05 | 27 | 2.23 | 23 | 3.23 | 17 |
| +DiffRS (Na et al., 2024) | 2.17 | 28.15 | 3.26 | 23.13 | 7.79 | 19.87 |
| +DG (Kim et al., 2023) | **1.93** | 27 | 2.12 | 23 | 3.62 | 17 |
| **+AC (Ours)** | 1.97 | **26.19** | **2.10** | **22.78** | **2.38** | **15.81** |

| Method | FID↓ | NFE↓ | FID↓ | NFE↓ | FID↓ | NFE↓ |
|---|---|---|---|---|---|---|
| DPM++ (Lu et al., 2022b) | 11.85 | 6 | 4.36 | 8 | 2.91 | 10 |
| UniPC (Zhao et al., 2023) | 11.10 | 6 | 3.86 | 8 | 2.85 | 10 |
| DPM-v3 (Zheng et al., 2023) | 8.73 | 6 | 3.62 | 8 | 2.65 | 10 |
| **DPM-v3 + AC (Ours)** | **7.12** | **5.61** | **3.09** | **7.53** | **2.54** | **9.88** |

Table 3: FID and NFE on unconditional CelebA-HQ 256 generation.

| Method | FID↓ | NFE↓ | FID↓ | NFE↓ | FID↓ | NFE↓ |
|---|---|---|---|---|---|---|
| ScoreSDE (KAR1 ) | 121.27 | 40 | 122.74 | 98 | 125.15 | 198 |
| +DLG (Kim & Ye, 2023) | 20.19 | 21.21 | 29.12 | 47.21 | 30.72 | 107.21 |
| **+AC (Ours)** | **15.13** | **15.94** | **22.55** | **44.06** | **15.69** | **87.26** |
| ScoreSDE (KAR2) | 83.21 | 40 | 57.28 | 98 | 29.74 | 198 |
| +DLG (Kim & Ye, 2023) | 17.92 | 21.21 | 12.12 | 47.21 | 8.14 | 107.21 |
| **+AC (Ours)** | **8.45** | **20.05** | **9.55** | **40.05** | **6.60** | **98.27** |

Table 4: Performance on conditional ImageNet (Top) 64×64, (Bottom) 256×256 generation.

| Method | FID↓ | NFE↓ | FID↓ | NFE↓ | FID↓ | NFE↓ |
|---|---|---|---|---|---|---|
| EDM (SDE) | 2.30 | 61 | 1.78 | 127 | 1.43 | 511 |
| **+AC (Ours)** | **2.25** | **58.75** | **1.77** | **121.98** | **1.42** | **483.86** |

| Method | FID↓ | NFE↓ | Precision↑ | Recall↑ |
|---|---|---|---|---|
| DiT (DDPM) (Peebles & Xie, 2023) | 2.35 | 250 | **0.829** | 0.576 |
| **+AC (Ours)** | **2.31** | **234.38** | 0.817 | **0.592** |

because the number of samples that can be generated is limited in a realistic setting. To address this, in Algorithm 1 we adopt a propose-until-accept design: proposals are repeatedly drawn until one is accepted, and only the accepted sample is recorded. By contrast, this eliminates such artifacts and uses the entire sampling budget to explore new regions of the space. While this deviates from the strict MH setting assumed in our theory, it yields a substantially better finite-sample approximation in practice. Further discussion is provided in Section 5.3 and Appendix E.

# 5 EXPERIMENTS

**Experimental setting** We employ our methods on various pre-trained networks trained on CIFAR-10 (Krizhevsky, 2009), ImageNet64×64 and 256×256 (Deng et al., 2009), CelebA-HQ 256×256 (Karras et al., 2017). In text-conditioned generation, we use Stable Diffusion v1.5 (Rombach et al., 2022). We report the Fréchet Inception Distance (FID) (Heusel et al., 2017), Precision/Recall (Kynkäänniemi et al., 2019). For text-conditioned generation, we further evaluate CLIP Score (CLIP) (Hessel et al., 2021), ImageReward (IR) (Xu et al., 2023a), GenEval Score (Ghosh et al., 2023) for evaluate text-image alignment. We use Heun, SDE sampler Karras et al. (2022), KAR1(deterministic), KAR2(stochastic) sampler (Kim & Ye, 2023), and DDPM (Ho et al., 2020), DDIM (Song et al., 2021a) sampler. We highlight the best-performing results compare to baseline model in bold. Detailed hyperparameters and settings are provided in Appendix C and H.

## 5.1 UNCONDITIONAL GENERATION

**CIFAR-10** The upper part of Table 1 shows results for unconditional generation on CIFAR-10. Our method is compatible with both EDM and DDO checkpoints. Both methods were re-tested without

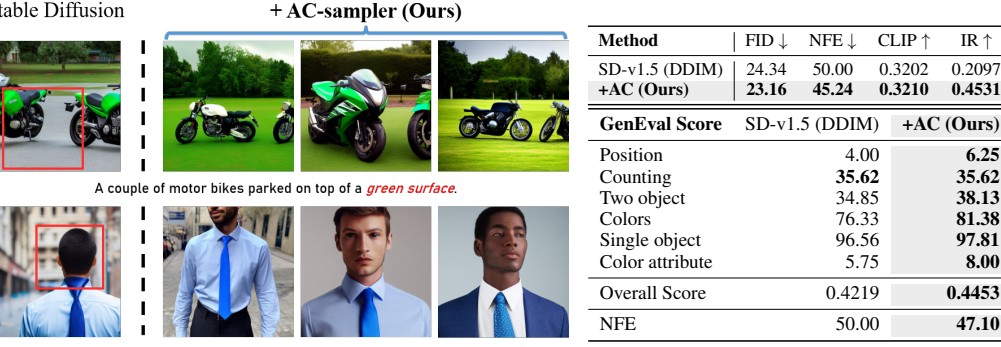

Figure 3: Generated image with SD-v1.5 (left) and AC-sampler (right). The red box indicates base model error.

Table 5: FID, CLIP Score (Top) and GenEval Score (Bottom) for SD-v1.5 and AC-sampler.

| Method | FID ↓ | NFE ↓ | CLIP ↑ | IR ↑ |
|---|---|---|---|---|
| SD-v1.5 (DDIM) | 24.34 | 50.00 | 0.3202 | 0.2097 |
| +AC (Ours) | **23.16** | **45.24** | **0.3210** | **0.4531** |

| GenEval Score | SD-v1.5 (DDIM) | +AC (Ours) |
|---|---|---|
| Position | 4.00 | **6.25** |
| Counting | **35.62** | **35.62** |
| Two object | 34.85 | **38.13** |
| Colors | 76.33 | **81.38** |
| Single object | 96.56 | **97.81** |
| Color attribute | 5.75 | **8.00** |
| Overall Score | 0.4219 | **0.4453** |
| NFE | 50.00 | **47.10** |

applying seed fixing as in the EDM setting. For the EDM checkpoint, our sampler improves the FID from 2.01 with 35 NFE to 1.97 with only 26.19 NFE. For the stronger DDO checkpoint, we achieve 1.41 FID while reducing NFE from 35 to 29.41. Although the gain in FID is marginal for highly capable pre-trained models, our method consistently reduces the NFE, demonstrating its efficiency.

**CelebA-HQ 256×256**   We employ the pre-trained time classifier released by DLG to reproduce their reported performance. We extend the MALA algorithm to the joint space of $(\mathbf{x}_\tau, \tau)$, as detailed in Appendix B, to support effective sampling on high-dimensional benchmark datasets in a restricted NFE setting. As shown in Table 3, the AC-Sampler demonstrates a clear improvement in FID compared to other methods with lower NFE.

**Method Compatibility**   The lower part of Table 1 reports results when applying AC-Sampler on top of existing acceleration and correction methods. When combined with DPM-v3, which originally yields 12.41 FID at 5 NFEs, our method improves performance to 9.88 FID with only 4.78 NFEs. Similarly, DG achieves 1.93 FID at 27 NFEs, while AC-Sampler applied to DG further reduces this to 1.84 FID at 26.19 NFEs. These experiments highlight that our method is orthogonal to existing acceleration and correction methods, and can flexibly enhance them. Table 2 and Figure 2 present the FID–NFE trade-offs of our method compared to existing acceleration and correction techniques. We observe that our method achieves better trade-offs in most NFE regimes.

## 5.2 Conditional Generation

**Class-Conditional**   Table 4 presents the results on conditional ImageNet 64×64 and 256×256 generation. For ImageNet 64×64, we employed a conditional score network, while for ImageNet 256×256, we generate images with classifier-free guidance (Ho & Salimans, 2021) scale $w = 1.5$. We find effective improvements in NFE and slight improvements in FID. Although class-conditioned settings inherently limit the length of the MCMC chain, our results

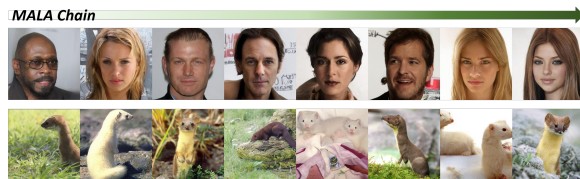

Figure 4: AC-Sampler on CelebA-HQ 256×256 (Top, unconditional) and ImageNet 256×256 (Bottom, condition on "Weasel"). Panel presents the final results of the MALA chains, ordered from left to right.

demonstrate that the proposed method can still be applied effectively under this constraint. We provide a further discussion of the class diversity and related experiments in Appendix F.2.

**Text-Conditional**   Our method is also applicable to high-dimensional text-to-image generation. We conducted experiments on the Stable Diffusion v1.5 (Rombach et al., 2022) with 5,000 randomly selected COCO validation prompts (Lin et al., 2014). Following the design of real-world text-to-image systems (e.g., DALL·E-2 (OpenAI, 2025), Midjourney (Midjourney, Inc., 2025), Bing Image Creator (Microsoft, 2025)), we generated four images per prompt. This setting aligns naturally with our method, which is designed to accelerate and improve the generation process when multiple

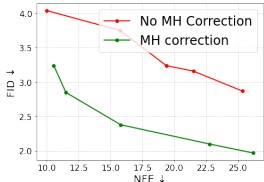

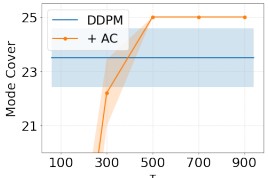

Table 6: Performance of AC-Sampler apply on EDM. Time denotes the wall-clock seconds required to generate 100 images.

Figure 5: Effect of an MH correction on AC-Sampler.

Figure 6: Mode cover with different $\tau$ in 25-Gaussian toy experiment.

| Method | FID↓ | NFE↓ | Precision↑ | Recall↑ | Time↓ |
|---|---|---|---|---|---|
| EDM (Heun) | 2.01 | 35 | **0.704** | 0.627 | 6.46 |
| +AC | **1.97** | **26.19** | 0.703 | **0.628** | **5.26** |
| EDM (Heun) | 2.24 | 23 | 0.703 | 0.625 | 4.30 |
| +AC | **2.21** | **20.67** | **0.707** | **0.632** | **4.15** |
| EDM (Heun) | 3.32 | 17 | 0.683 | 0.622 | 3.20 |
| +AC | **2.41** | **15.78** | **0.699** | **0.623** | **3.19** |

images are generated for a single prompt. Also, we use classifier-free guidance (Ho & Salimans, 2021) scale $w = 7.5$. Figure 3 presents samples generated by our method. For quantitative evaluation, we randomly selected one of the four generated images for each prompt. Our method achieves both faster sampling and improved FID compared to the base sampler as shown in Table 5 (Top).

We also conduct experiment on GenEval(Ghosh et al., 2023) benchmark. GenEval evaluates how well the generated images follow the prompt instructions. Since GenEval requires 4 different images per prompt for evaluation, we make four samples per prompt for computing metric. Ours consistently performs as well as or better than the baseline across all metrics as shown in Table 5 (Bottom).

### 5.3 ABLATION STUDIES

**Distribution Alignment** To validate the necessity of the MH correction, we conducted an ablation study where the MH accept–reject step was omitted (i.e., all proposals were accepted). As illustrated in Fig. 5, the FID deteriorates significantly without the MH correction. This result empirically demonstrates that the MH step is crucial for effectively aligning the generated samples with the target distribution. Furthermore, it validates that the acceptance probabilities derived from the discriminator provide reliable density ratio estimates consistent with the underlying theory.

**Jump Markov Chain** Conventional MH ($\text{MH}_C$) retains rejected proposals as part of the chain, which ensures detailed balance but is inefficient for generative sampling under a limited sample budget. As shown in Table 7, the $\text{MH}_C$ formulation involving rejected samples results in both lower recall and worse FID. In contrast, our propose-until-accept design in Algorithm 1 encourages transitions to new regions of the sample space, thereby improving both diversity and fidelity. This approach can be formally interpreted as a Jump Markov chain (Rosenthal et al., 2021), and further details are provided in Appendix E.

Table 7: Comparison of conventional MH ($\text{MH}_C$) and Alg. 1.

| Method | FID↓ | NFE↓ | Recall↑ |
|---|---|---|---|
| EDM (Base) | 2.05 | 27 | 0.627 |
| +AC with $\text{MH}_C$ | 3.22 | 25.08 | 0.580 |
| +AC with Alg. 1 | **1.97** | **26.19** | **0.628** |

**Wall-clock time** Since our method employs an additional discriminator at inference time, the wall-clock time could potentially be slower even with the same NFE. To evaluate this, we measured the average time (in seconds) required to generate 100 samples using both the base sampler and our method on a single RTX 3090. As shown in Table 6 and Figure 7, our approach not only improved sample quality but also achieved faster wall-clock time across various sampling strategies. Unlike conventional approaches that reduce NFE by enlarging the time step and thereby increasing discretization error, our method generates intermediate samples without coarsening the time grid. As a result, the improvement in sample quality is particularly pronounced in the low-NFE regime.

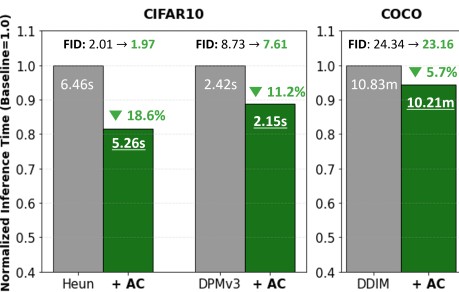

Figure 7: Wall-clock Time Comparison.

**Sample Diversity** Because our method relies on MH, successive samples can be correlated. Thus, assessing whether it still produces diverse samples is crucial. In Table 6, the recall metric, which is an indicator of sample diversity, is comparable to the base sampler. These results indicate that, our method can preserve sample diversity. Together with the improved FID and reduced NFE, these

results demonstrate both the effectiveness and efficiency of our approach. The MALA chains on benchmark datasets are shown in Fig. 4, where we observe that our method successfully generates diverse and high-quality images. Further analysis is provided in Appendix D.

**Toy experiment**   We applied our method to a 25-Gaussian toy task. We generated 100 samples and counted a mode as covered if any sample fell within $2\sigma$ of its mean. Across 10 trials, DDPM covered 23.5 modes on average, whereas our method consistently covered all 25 as in Fig. 6. The solid lines indicate the mean, and the shaded areas are the standard deviation. This confirms that our method effectively performs distribution correction and ensures diverse mode coverage. When low-$\tau$ regimes, more samples are required for sufficient cover. Fig. 8 shows the results with 100 generated samples. Red circles denote uncovered modes. Ours reduces NFE while fully covering all modes of the 25-Gaussian mixture.

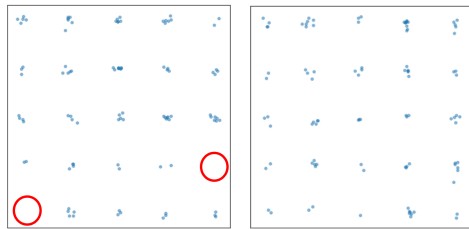

(a) DDPM (NFE=1000)   (b) +AC (NFE=504.5)

Figure 8: Toy experiment results. Red circles denote uncovered mode.

**Hyperparameter**   We primarily tune two parameters: the target timestep $\tau$ and the signal-to-noise ratio (SNR) of the Langevin proposal. The SNR controls the proposal step size $\eta$. Larger SNR yields larger steps. With $T = 18$, we vary $\tau$ and SNR and report the resulting FID, NFE and Recall. As $\tau$ decreases (i.e., closer to the data distribution), the distribution becomes sharper, and MALA mixing deteriorates. Also, if the SNR is too small, proposals change little from the current sample, also leading to slow exploration. Given the constraints of limited number of generation, the choice of the target timestep $\tau$ and the proposal SNR is crucial. A detailed analysis of hyperparameter is provided in Appendix C.

| Hyperparameter | | Metrics | | |
|---|---|---|---|---|
| $\tau$ | SNR | FID ↓ | NFE ↓ | Recall ↑ |
| | 0.10 | 2.89 | 25.13 | 0.550 |
| 13 | 0.20 | 2.06 | 25.65 | 0.620 |
| | 0.23 | 1.97 | 26.19 | 0.628 |
| | 0.10 | 8.77 | 15.21 | 0.200 |
| 8 | 0.20 | 6.14 | 15.75 | 0.422 |
| | 0.23 | 6.73 | 16.70 | 0.441 |
| | 0.10 | 62.67 | 5.29 | 0 |
| 3 | 0.20 | 46.60 | 5.75 | 0 |
| | 0.23 | 43.96 | 6.26 | 0 |

Table 8: Hyperparameter analysis.

**Discriminator NFE**   Representative studies that used time-dependent discriminator, such as DG (Kim et al., 2023) and DiffRS (Na et al., 2024), provide effective correction capabilities. However, these methods rely on the discriminator throughout the whole sampling process, inevitably resulting in high Discriminator NFEs (D-NFE). Furthermore, DG requires gradient computation on the discriminator. In contrast, our method utilizes the discriminator only when constructing the MALA chain, thereby achieving significantly lower D-NFE compared to existing approaches. We analyze the D-NFE corresponding to the comparisons in Table 2 (Top) and visualize the results in Figure 9.

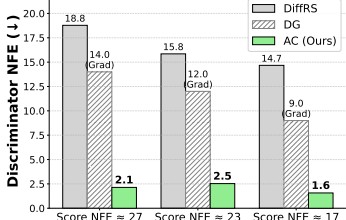

Figure 9: Discriminator NFE.

(Grad) denotes gradient computation. Compared to the baselines, our method requires significantly fewer D-NFE, which directly contributes to the wall-clock time advantages discussed earlier.

## 6   CONCLUSION

We introduced AC-Sampler, which accelerates and corrects diffusion sampling via Metropolis–Hastings with a Langevin proposal. By sampling at intermediate diffusion timesteps and using only time-dependent discriminators for density-ratio estimation, our method improves sample quality and provides faster inference without requiring any fine-tuning of the base diffusion model. AC-Sampler theoretically achieves reduced NFE and decreases the KL divergence from the true marginal distribution at each refinement step. Empirically, AC-Sampler attains better FID with fewer NFEs across multiple datasets and is applicable to high-dimensional text-to-image generation. It also integrates seamlessly with various existing acceleration and correction methods.

## ACKNOWLEDGMENTS

This work was supported by the IITP (Institute of Information & Communications Technology Planning & Evaluation)-ITRC (Information Technology Research Center) grant funded by the Korea government (Ministry of Science and ICT) (IITP-2026-RS-2024-00437268). (50 %) This research was supported by AI Technology Development for Commonsense Extraction, Reasoning, and Inference from Heterogeneous Data(IITP) funded by the Ministry of Science and ICT(RS-2022-II220077) (50 %)

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

## A  PROOFS AND MATHEMATICAL EXPLANATIONS

In this section, we provide a mathematical explanation and formal derivation of the theorems presented in the main body of the paper.

### A.1  THEORETICAL ANALYSIS OF DLG (KIM & YE, 2023)

**Proposition A.1.** *Let $q(\mathbf{x}, t)$ be the true joint distribution over data $\mathbf{x}$ and diffusion timestep $t$, and $p^{\boldsymbol{\theta}}(\mathbf{x}, t)$ be the model joint distribution .*

*Suppose that there is an optimal time classifier $\boldsymbol{\psi}^*$, i.e., $p^{\boldsymbol{\psi}^*}(t \mid \mathbf{x}) = q(t \mid \mathbf{x}), \forall \mathbf{x}, t$. If the model marginal distribution $p^{\boldsymbol{\theta}}(\mathbf{x})$ does not match the true marginal distribution, i.e., $\exists\ \mathbf{x}$ s.t. $p^{\boldsymbol{\theta}}(\mathbf{x}) \neq q(\mathbf{x})$, then the Markov chain defined by Gibbs sampling between $p^{\boldsymbol{\theta}}(\mathbf{x} \mid t)$ and $p^{\boldsymbol{\psi}^*}(t \mid \mathbf{x})$ does not have $q(\mathbf{x}, t)$ as its stationary distribution.*

*Proof.* Let's assume that the true joint distribution $q(\mathbf{x}_t, t)$ is a stationary distribution of Gibbs Sampling, whose transition kernel is $T((\mathbf{x}, t) \to (\mathbf{x}', t')) = p^{\boldsymbol{\theta}}(\mathbf{x}' \mid t)p^{\boldsymbol{\psi}^*}(t' \mid \mathbf{x}')$. Let $(\mathbf{x}, t) \sim q(\cdot, \cdot)$, and $(\mathbf{x}', t')$ is a sample drawn from the transition kernel $T((\mathbf{x}, t) \to (\cdot, \cdot))$. Since the stationary distribution should satisfy the invariance condition, for arbitrary $\mathbf{x}'$ the following equation holds:

$$q(\mathbf{x}', t') = \iint q(\mathbf{x}, t)T((\mathbf{x}, t) \to (\mathbf{x}', t'))\, d\mathbf{x}\, dt \tag{13}$$

$$= \iint q(\mathbf{x}, t)p^{\boldsymbol{\theta}}(\mathbf{x}' \mid t)p^{\boldsymbol{\psi}^*}(t' \mid \mathbf{x}')\, d\mathbf{x}\, dt \tag{14}$$

$$= \iint q(\mathbf{x}, t)p^{\boldsymbol{\theta}}(\mathbf{x}' \mid t)q(t' \mid \mathbf{x}')\, d\mathbf{x}\, dt \tag{15}$$

$$= \int q(t)p^{\boldsymbol{\theta}}(\mathbf{x}' \mid t)q(t' \mid \mathbf{x}')\, dt \tag{16}$$

$$= \int q(t)\frac{p^{\boldsymbol{\theta}}(t \mid \mathbf{x}')p^{\boldsymbol{\theta}}(\mathbf{x}')}{p^{\boldsymbol{\theta}}(t)}q(t' \mid \mathbf{x}')\, dt \tag{17}$$

$$= \int p^{\boldsymbol{\theta}}(t \mid \mathbf{x}')p^{\boldsymbol{\theta}}(\mathbf{x}')q(t' \mid \mathbf{x}')\, dt \tag{18}$$

$$= p^{\boldsymbol{\theta}}(\mathbf{x}')q(t' \mid \mathbf{x}')\int p^{\boldsymbol{\theta}}(t \mid \mathbf{x}')\, dt \tag{19}$$

$$= p^{\boldsymbol{\theta}}(\mathbf{x}')q(t' \mid \mathbf{x}') \tag{20}$$

Note that the marginal distribution $q(t) = p^{\boldsymbol{\theta}}(t)$ for every $t$ is an uniform distribution. To satisfy the invariance, $\forall \mathbf{x}'$, $p^{\boldsymbol{\theta}}(\mathbf{x}') = q(\mathbf{x}')$ so the proof holds due to the contradiction. $\square$

Proposition A.1 states that even if the time-classifier in DLG(Kim & Ye, 2023) is optimal, it is impossible to sample from the true distribution. So we leverage Metropolis-Hastings Correction with this proposal distribution. Detailed explanation is given in Appendix B.

### A.2  PROOF OF THEOREM 4.1

**Theorem 4.1.** *Let $\mathbf{x}_t$ and $\tilde{\mathbf{x}}_t$ be two arbitrary samples at diffusion timestep $t$. Then, for any fixed $\mathbf{x}_{t-1}$, the density ratio of the true marginal distribution $q_t$ is given by:*

$$\frac{q_t(\tilde{\mathbf{x}}_t)}{q_t(\mathbf{x}_t)} = \frac{q_{t|t-1}(\tilde{\mathbf{x}}_t \mid \mathbf{x}_{t-1})}{q_{t|t-1}(\mathbf{x}_t \mid \mathbf{x}_{t-1})} \cdot \frac{L_t(\tilde{\mathbf{x}}_t, t)}{L_t(\mathbf{x}_t, t)} \cdot \frac{p^{\boldsymbol{\theta}}_{t-1|t}(\mathbf{x}_{t-1} \mid \mathbf{x}_t)}{p^{\boldsymbol{\theta}}_{t-1|t}(\mathbf{x}_{t-1} \mid \tilde{\mathbf{x}}_t)}, \tag{6}$$

*where $L_t(\mathbf{x}_t, t) := q_t(\mathbf{x}_t)/p^{\theta}_t(\mathbf{x}_t)$ denotes the likelihood ratio between the data and model marginal distributions at timestep $t$.*

*Proof.* We derive the marginal density ratio as follows:

$$\frac{q_t(\tilde{\mathbf{x}}_t)}{q_t(\mathbf{x}_t)} = \frac{q_{t-1}(\mathbf{x}_{t-1})}{q_{t-1}(\mathbf{x}_{t-1})} \cdot \frac{q_{t|t-1}(\tilde{\mathbf{x}}_t \mid \mathbf{x}_{t-1})}{q_{t|t-1}(\mathbf{x}_t \mid \mathbf{x}_{t-1})} \cdot \frac{q_{t-1|t}(\mathbf{x}_{t-1} \mid \mathbf{x}_t)}{q_{t-1|t}(\mathbf{x}_{t-1} \mid \tilde{\mathbf{x}}_t)} \tag{21}$$

$$= \frac{q_{t|t-1}(\tilde{\mathbf{x}}_t \mid \mathbf{x}_{t-1})}{q_{t|t-1}(\mathbf{x}_t \mid \mathbf{x}_{t-1})} \cdot \frac{q_{t-1|t}(\mathbf{x}_{t-1} \mid \mathbf{x}_t)}{q_{t-1|t}(\mathbf{x}_{t-1} \mid \tilde{\mathbf{x}}_t)} \tag{22}$$

$$= \frac{q_{t|t-1}(\tilde{\mathbf{x}}_t \mid \mathbf{x}_{t-1})}{q_{t|t-1}(\mathbf{x}_t \mid \mathbf{x}_{t-1})} \cdot \frac{q_{t-1|t}(\mathbf{x}_{t-1} \mid \mathbf{x}_t)}{p^{\boldsymbol{\theta}}_{t-1|t}(\mathbf{x}_{t-1} \mid \mathbf{x}_t)} \cdot \frac{p^{\boldsymbol{\theta}}_{t-1|t}(\mathbf{x}_{t-1} \mid \tilde{\mathbf{x}}_t)}{q_{t-1|t}(\mathbf{x}_{t-1} \mid \tilde{\mathbf{x}}_t)} \cdot \frac{p^{\boldsymbol{\theta}}_{t-1|t}(\mathbf{x}_{t-1} \mid \mathbf{x}_t)}{p^{\boldsymbol{\theta}}_{t-1|t}(\mathbf{x}_{t-1} \mid \tilde{\mathbf{x}}_t)} \tag{23}$$

Following the derivation process of DiffRS (Na et al., 2024), Eq. 23 can be expressed as follows:

$$= \frac{q_{t|t-1}(\tilde{\mathbf{x}}_t \mid \mathbf{x}_{t-1})}{q_{t|t-1}(\mathbf{x}_t \mid \mathbf{x}_{t-1})} \cdot \frac{L_{t-1}(\mathbf{x}_{t-1}, t-1)}{L_t(\mathbf{x}_t, t)} \cdot \frac{L_t(\tilde{\mathbf{x}}_t, t)}{L_{t-1}(\mathbf{x}_{t-1}, t-1)} \cdot \frac{p^{\boldsymbol{\theta}}_{t-1|t}(\mathbf{x}_{t-1} \mid \mathbf{x}_t)}{p^{\boldsymbol{\theta}}_{t-1|t}(\mathbf{x}_{t-1} \mid \tilde{\mathbf{x}}_t)} \tag{24}$$

$$= \underbrace{\frac{q_{t|t-1}(\tilde{\mathbf{x}}_t \mid \mathbf{x}_{t-1})}{q_{t|t-1}(\mathbf{x}_t \mid \mathbf{x}_{t-1})}}_{\text{Forward term}} \cdot \underbrace{\frac{L_t(\tilde{\mathbf{x}}_t, t)}{L_t(\mathbf{x}_t, t)}}_{\text{Likelihood ratio}} \cdot \underbrace{\frac{p^{\boldsymbol{\theta}}_{t-1|t}(\mathbf{x}_{t-1} \mid \mathbf{x}_t)}{p^{\boldsymbol{\theta}}_{t-1|t}(\mathbf{x}_{t-1} \mid \tilde{\mathbf{x}}_t)}}_{\text{Transition kernel term}} \tag{25}$$

$$\square$$

The Forward term and Transition kernel term are tractable since it is a gaussian distribution. By approximating the Likelihood ratio by a discriminator, we can derive a tractable form of the acceptance probability.

### A.3 PROOF OF PROPOSITION 4.2

**Proposition 4.2.** *Let the reverse diffusion process consist of a total number of timesteps $T$, and let the target timestep in AC-Sampler be $\tau$. Suppose that $l$ denotes the length of the Markov chain at timestep $\tau$. Let $NFE_R$ denote the average reduction in NFE per sample achieved by AC-Sampler.*

*If the acceptance probability $\alpha$ satisfies $\alpha > \frac{1}{T-\tau+1}$, then, as $l \to \infty$, the expected NFE reduction remains strictly positive, while its variance converges to zero:*

$$\lim_{l \to \infty} \mathbb{E}[NFE_R] > 0, \quad \lim_{l \to \infty} \text{Var}(NFE_R) = 0. \tag{10}$$

*Proof.* Let's assume that the acceptance probability $\alpha$ is fixed in $(0, 1]$, and that there is no burn-in process in the AC-Sampler for simplicity.[1]

Let $R$ denote the total NFE required at timestep $\tau$ for the MCMC step. At timestep $\tau$, Algorithm 1 runs $l$ times. Since we make proposals until it is accepted, $R$ follows the Negative Binomial distribution, i.e., $R \sim \text{NB}(l, \alpha)$. Let $\text{NFE}_T$ denote the total NFE required in the AC-Sampler sampling step. Then the following decomposition holds:

$$\text{NFE}_T = \underbrace{T - \tau}_{\text{Initial denoising step}} + \underbrace{R}_{\text{MALA step}} + \underbrace{l(\tau - 1)}_{\text{Denoising step after MALA}}. \tag{26}$$

Since we obtain the score value of each proposal at the MALA step and denoise total of $l$ samples, the denoising step after MALA is $l(\tau - 1)$. Since $R$ follows a negative binomial distribution, we have $\mathbb{E}[\text{NFE}_T] = T - \tau + \frac{l}{\alpha} + l(\tau - 1)$ and $\text{Var}(\text{NFE}_T) = \frac{l(1-\alpha)}{\alpha^2}$. Then, the mean of $\text{NFE}_R$ is as follows:

$$\mathbb{E}[\text{NFE}_R] = T - \frac{\mathbb{E}[\text{NFE}_T]}{l} \tag{27}$$

$$= T - \frac{T - \tau}{l} - \frac{1}{\alpha} - (\tau - 1) \tag{28}$$

$$= T - \tau + 1 - \frac{1}{\alpha} - \frac{T - \tau}{l}. \tag{29}$$

---

[1] We can get same result with burn-in process easily

Taking limits on both sides yields

$$\lim_{l \to \infty} \mathbb{E}[\text{NFE}_R] = T - \tau + 1 - \frac{1}{\alpha} \tag{30}$$

$$> 0 \qquad \left( \because \alpha > \frac{1}{T - \tau + 1} \right) \tag{31}$$

Moreover, $\text{Var}(\text{NFE}_R) = \frac{1-\alpha}{\alpha^2} \cdot \frac{1}{l}$, and thus taking limits gives $\lim_{l \to \infty} \text{Var}(\text{NFE}_R) = 0$, which concludes the proof.

$\square$

### A.4 PROOF OF THEOREM 4.3

**Theorem 4.3.** *Let $p_0^{\boldsymbol{\theta}}$, $p_0^{\boldsymbol{\theta},\boldsymbol{\phi}^*}$ denote the model distribution and refined distribution by AC-Sampler with optimal discriminator $\boldsymbol{\phi}^*$, respectively. Then, the KL divergence between the true data distribution $q_0$ and the refined distribution $p_0^{\boldsymbol{\theta},\boldsymbol{\phi}^*}$ is bounded by:*

$$D_{KL}(q_0(\mathbf{x}_0)||p^{\boldsymbol{\theta},\boldsymbol{\phi}^*}(\mathbf{x}_0)) \le D_{KL}(q_0(\mathbf{x}_0)||p^{\boldsymbol{\theta}}(\mathbf{x}_0)) \tag{11}$$

*Proof.* First, let $\tau$ be a timestep that MALA occurs in AC-Sampler framework. From (Ho et al., 2020), the upper bound of KL divergence between the true data distribution and the model distribution can be written as follows:

$$D_{\text{KL}}[q_0 \| p_0^{\boldsymbol{\theta}}] = \mathbb{E}_{q_0} \left[ \log \frac{q_0(\mathbf{x}_0)}{p_0^{\boldsymbol{\theta}}(\mathbf{x}_0)} \right] \tag{32}$$

$$= \mathbb{E}_{q_0}[-\log p_0^{\boldsymbol{\theta}}(\mathbf{x}_0)] - H(q_0) \tag{33}$$

$$= \mathbb{E}_{q_0} \left[ -\log \int q_{1:\tau|0}(\mathbf{x}_{1:\tau} \mid \mathbf{x}_0) \frac{p_{0:\tau}^{\boldsymbol{\theta}}(\mathbf{x}_{0:\tau})}{q_{1:\tau|0}(\mathbf{x}_{1:\tau} \mid \mathbf{x}_0)} \, d\mathbf{x}_{1:\tau} \right] - H(q_0) \tag{34}$$

$$\le \mathbb{E}_{q_0} \left[ -\int q_{1:\tau|0}(\mathbf{x}_{1:\tau} \mid \mathbf{x}_0) \log \frac{p_{0:\tau}^{\boldsymbol{\theta}}(\mathbf{x}_{0:\tau})}{q_{1:\tau|0}(\mathbf{x}_{1:\tau} \mid \mathbf{x}_0)} \, d\mathbf{x}_{1:\tau} \right] - H(q_0) \tag{35}$$

$$= \mathbb{E}_{q_{0:\tau}} \left[ -\log p_\tau^{\boldsymbol{\theta}}(\mathbf{x}_\tau) - \sum_{i=1}^{\tau} \log \frac{p_{i-1|i}^{\boldsymbol{\theta}}(\mathbf{x}_{i-1} \mid \mathbf{x}_i)}{q_{i-1|i}(\mathbf{x}_{i-1} \mid \mathbf{x}_i)} \cdot \frac{q_i(\mathbf{x}_i)}{q_{i-1}(\mathbf{x}_{i-1})} \right] - H(q_0) \tag{36}$$

$$= \mathbb{E}_{q_{0:\tau}} \left[ \log \frac{q_\tau(\mathbf{x}_\tau)}{p_\tau^{\boldsymbol{\theta}}(\mathbf{x}_\tau)} - \log q_0(\mathbf{x}_0) - \sum_{i=1}^{\tau} \log \frac{p_{i-1|i}^{\boldsymbol{\theta}}(\mathbf{x}_{i-1} \mid \mathbf{x}_i)}{q_{i-1|i}(\mathbf{x}_{i-1} \mid \mathbf{x}_i)} \right] - H(q_0) \tag{37}$$

$$= D_{\text{KL}}[q_\tau \| p_\tau^{\boldsymbol{\theta}}] + \mathbb{E}_{q_{0:\tau}} \left[ \sum_{i=1}^{\tau} \log \frac{q_{i-1|i}(\mathbf{x}_{i-1} \mid \mathbf{x}_i)}{p_{i-1|i}^{\boldsymbol{\theta}}(\mathbf{x}_{i-1} \mid \mathbf{x}_i)} \right] \tag{38}$$

$$= D_{\text{KL}}[q_\tau \| p_\tau^{\boldsymbol{\theta}}] + \sum_{i=1}^{\tau} \mathbb{E}_{q_i} \left[ D_{\text{KL}}[q_{i-1|i}(\mathbf{x}_{i-1} \mid \mathbf{x}_i) \| p_{i-1|i}^{\boldsymbol{\theta}}(\mathbf{x}_{i-1} \mid \mathbf{x}_i)] \right] \tag{39}$$

By substituting $p_\tau^{\boldsymbol{\theta}}$ with $p_\tau^{\boldsymbol{\theta},\boldsymbol{\phi}^*}$, the following bounded relation also holds:

$$D_{\text{KL}}[q_0 \| p_0^{\boldsymbol{\theta},\boldsymbol{\phi}^*}] \le D_{\text{KL}}[q_\tau \| p_\tau^{\boldsymbol{\theta},\boldsymbol{\phi}^*}] + \sum_{i=1}^{\tau} \mathbb{E}_{q_i} \left[ D_{\text{KL}}[q_{i-1|i}(\mathbf{x}_{i-1} \mid \mathbf{x}_i) \| p_{i-1|i}^{\boldsymbol{\theta},\boldsymbol{\phi}^*}(\mathbf{x}_{i-1} \mid \mathbf{x}_i)] \right] \tag{40}$$

Since $p_{i-1|i}^{\boldsymbol{\theta},\boldsymbol{\phi}^*}(\mathbf{x}_{i-1} \mid \mathbf{x}_i) = p_{i-1|i}^{\boldsymbol{\theta}}(\mathbf{x}_{i-1} \mid \mathbf{x}_i) \; \forall i \le \tau$, it is sufficient to show

$$D_{\text{KL}}[q_\tau \| p_\tau^{\boldsymbol{\theta},\boldsymbol{\phi}^*}] \le D_{\text{KL}}[q_\tau \| p_\tau^{\boldsymbol{\theta}}] \tag{41}$$

When the discriminator is optimal and the burn-in process has been sufficiently performed, $p_\tau^{\boldsymbol{\theta},\boldsymbol{\phi}^*} = q_\tau$, so $D_{\text{KL}}[q_\tau \| p_\tau^{\boldsymbol{\theta},\boldsymbol{\phi}^*}] = 0$. Since the KL-Divergence is non-negative, the proof is complete. $\square$

The stationary distribution at timestep $\tau$ is $p_\tau^{\boldsymbol{\theta}} L_\tau^{\boldsymbol{\phi}}$. The gap between the two distributions in Eq. 41 is as follows:

$$D_{\mathrm{KL}}[q_\tau \,\|\, p_\tau^{\boldsymbol{\theta}}] - D_{\mathrm{KL}}[q_\tau \,\|\, p_\tau^{\boldsymbol{\theta},\boldsymbol{\phi}}] = \int q_\tau \log \frac{q_\tau}{p_\tau^{\boldsymbol{\theta}}} d\mathbf{x}_\tau - \int q_\tau \log \frac{q_\tau}{p_\tau^{\boldsymbol{\theta},\boldsymbol{\phi}}} d\mathbf{x}_\tau \tag{42}$$

$$= \int q_\tau \log \frac{q_\tau}{p_\tau^{\boldsymbol{\theta}}} d\mathbf{x}_\tau - \int q_\tau \log \frac{q_\tau}{p_\tau^{\boldsymbol{\theta}} L_\tau^{\boldsymbol{\phi}}} d\mathbf{x}_\tau \tag{43}$$

$$= \int q_\tau \log L_\tau^{\boldsymbol{\phi}} d\mathbf{x}_\tau \tag{44}$$

If the discriminator cannot distinguish between the two distributions $p_t^{\boldsymbol{\theta}}, q_t$ at all, i.e., $L_t^{\boldsymbol{\phi}}(\mathbf{x}_t, t) = 1$ for all $\mathbf{x}_t, t$, then the target distribution of Metropolis-Hastings algorithm becomes $p_\tau^{\boldsymbol{\theta}}$. As a result the gap between the two KL divergences in Eq. 44 becomes 0. By training the discriminator, the gap converges to $D_{\mathrm{KL}}[q_\tau \| p_\tau^{\boldsymbol{\theta}}] (\geq 0)$, indicating that the bound becomes tighter.

## A.5 PROOF OF THEOREM 4.4

**Theorem 4.4.** *Let $T_\tau$ be the transition kernel of MALA at timestep $\tau$. Also, $p_\tau^{\boldsymbol{\theta},\boldsymbol{\phi}^*,(l)}$ denotes marginal distribution at timestep $\tau$ after the l-th MALA transition from $p_\tau^{\boldsymbol{\theta}}$, and $p_0^{\boldsymbol{\theta},\boldsymbol{\phi}^*,(l)}$ denotes the data distribution generated from $p_\tau^{\boldsymbol{\theta},\boldsymbol{\phi}^*,(l)}$ with denoising transition kernel $p_{t-1|t}^{\boldsymbol{\theta}}$. $\mathcal{L}^p$ denotes a space of function which satisfies $(\int_{\mathbb{R}} |f|^p d\mathbf{x})^{\frac{1}{p}} < \infty$. If $q_\tau(\mathbf{x}_\tau) \in \mathcal{L}^\alpha, \log \left( \frac{p_\tau^{\boldsymbol{\theta},\boldsymbol{\phi}^*,(l)}(\mathbf{x}_\tau)}{q_\tau(\mathbf{x}_\tau)} \right) \in \mathcal{L}^\beta, T_\tau \in \mathcal{L}^\gamma$, where $\alpha, \beta, \gamma \in [1, \infty]$ satisfy $\frac{1}{\alpha} + \frac{1}{\beta} + \frac{1}{\gamma} = 1$, then the KL divergence between the true data distribution $q_0$ and the refined distribution $p_0^{\boldsymbol{\theta},\boldsymbol{\phi}^*,(l+1)}$ is bounded by:*

$$D_{KL}(q_0(\mathbf{x}_0)||p_0^{\boldsymbol{\theta},\boldsymbol{\phi}^*,(l+1)}(\mathbf{x}_0)) \leq D_{KL}(q_0(\mathbf{x}_0)||p_0^{\boldsymbol{\theta},\boldsymbol{\phi}^*,(l)}(\mathbf{x}_0)) \tag{12}$$

*Proof.* As in Theorem 4.3, it suffices to show that

$$D_{\mathrm{KL}}[q_\tau \,\|\, p_\tau^{\boldsymbol{\theta},\boldsymbol{\phi}^*,(l+1)}] \leq D_{\mathrm{KL}}[q_\tau \,\|\, p_\tau^{\boldsymbol{\theta},\boldsymbol{\phi}^*,(l)}]$$

Then the below equation holds. We refer to the proof procedure in (Tsvetkov et al., 2017).

$$D_{\mathrm{KL}}[q_\tau \,\|\, p_\tau^{\boldsymbol{\theta},\boldsymbol{\phi}^*,(l+1)}] = \int q_\tau(\mathbf{x}_\tau) \log \frac{q_\tau(\mathbf{x}_\tau)}{p_\tau^{\boldsymbol{\theta},\boldsymbol{\phi}^*,(l+1)}(\mathbf{x}_\tau)} \tag{45}$$

$$= -\int q_\tau(\mathbf{x}_\tau) \log p_\tau^{\boldsymbol{\theta},\boldsymbol{\phi}^*,(l+1)}(\mathbf{x}_\tau) \, d\mathbf{x}_\tau - H(q_\tau) \tag{46}$$

$$= -\int q_\tau(\mathbf{x}_\tau) \log \left\{ \int p_\tau^{\boldsymbol{\theta},\boldsymbol{\phi}^*,(l)}(\tilde{\mathbf{x}}_\tau) T_\tau(\tilde{\mathbf{x}}_\tau \to \mathbf{x}_\tau) \, d\tilde{\mathbf{x}}_\tau \right\} d\mathbf{x}_\tau - H(q_\tau) \tag{47}$$

$$= -\int q_\tau(\mathbf{x}_\tau) \log \left\{ \int \frac{p_\tau^{\boldsymbol{\theta},\boldsymbol{\phi}^*,(l)}(\tilde{\mathbf{x}}_\tau)}{q_\tau(\tilde{\mathbf{x}}_\tau)} q_\tau(\tilde{\mathbf{x}}_\tau) T_\tau(\tilde{\mathbf{x}}_\tau \to \mathbf{x}_\tau) \, d\tilde{\mathbf{x}}_\tau \right\} d\mathbf{x}_\tau - H(q_\tau) \tag{48}$$

$$= -\int q_\tau(\mathbf{x}_\tau) \log \left\{ \int \frac{p_\tau^{\boldsymbol{\theta},\boldsymbol{\phi}^*,(l)}(\tilde{\mathbf{x}}_\tau)}{q_\tau(\tilde{\mathbf{x}}_\tau)} q_\tau(\mathbf{x}_\tau) T_\tau(\mathbf{x}_\tau \to \tilde{\mathbf{x}}_\tau) \, d\tilde{\mathbf{x}}_\tau \right\} d\mathbf{x}_\tau - H(q_\tau) \tag{49}$$

$$= -\int q_\tau(\mathbf{x}_\tau) \log \left\{ q_\tau(\mathbf{x}_\tau) \int \frac{p_\tau^{\boldsymbol{\theta},\boldsymbol{\phi}^*,(l)}(\tilde{\mathbf{x}}_\tau)}{q_\tau(\tilde{\mathbf{x}}_\tau)} T_\tau(\mathbf{x}_\tau \to \tilde{\mathbf{x}}_\tau) \, d\tilde{\mathbf{x}}_\tau \right\} d\mathbf{x}_\tau - H(q_\tau) \tag{50}$$

$$= -\int q_\tau(\mathbf{x}_\tau) \log \left\{ \int \frac{p_\tau^{\boldsymbol{\theta},\boldsymbol{\phi}^*,(l)}(\tilde{\mathbf{x}}_\tau)}{q_\tau(\tilde{\mathbf{x}}_\tau)} T_\tau(\mathbf{x}_\tau \to \tilde{\mathbf{x}}_\tau) \, d\tilde{\mathbf{x}}_\tau \right\} d\mathbf{x}_\tau \tag{51}$$

$$\leq -\iint q_\tau(\mathbf{x}_\tau) \left[ \log \left\{ \frac{p_\tau^{\boldsymbol{\theta},\boldsymbol{\phi}^*,(l)}(\tilde{\mathbf{x}}_\tau)}{q_\tau(\tilde{\mathbf{x}}_\tau)} \right\} T_\tau(\mathbf{x}_\tau \to \tilde{\mathbf{x}}_\tau) \right] d\tilde{\mathbf{x}}_\tau d\mathbf{x}_\tau \tag{52}$$

Since $\frac{1}{\alpha} + \frac{1}{\beta} + \frac{1}{\gamma} = 1$, we can apply Hölder's inequality (Hölder, 1889) in Eq. 52

$$\iint q_\tau(\mathbf{x}_\tau) \left[ \log \left\{ \frac{p_\tau^{\boldsymbol{\theta},\boldsymbol{\phi}^*,(l)}(\tilde{\mathbf{x}}_\tau)}{q_\tau(\tilde{\mathbf{x}}_\tau)} \right\} T_\tau(\mathbf{x}_\tau \to \tilde{\mathbf{x}}_\tau) \right] d\tilde{\mathbf{x}}_\tau d\mathbf{x}_\tau \tag{53}$$

$$\leq \left( \iint |q_\tau(\mathbf{x}_\tau)|^\alpha \, d\tilde{\mathbf{x}}_\tau d\mathbf{x}_\tau \right)^{\frac{1}{\alpha}} \left( \iint \left| \log \left\{ \frac{p_\tau^{\boldsymbol{\theta},\boldsymbol{\phi}^*,(l)}(\tilde{\mathbf{x}}_\tau)}{q_\tau(\tilde{\mathbf{x}}_\tau)} \right\} \right|^\beta \, d\tilde{\mathbf{x}}_\tau d\mathbf{x}_\tau \right)^{\frac{1}{\beta}} \left( \iint |T_\tau(\mathbf{x}_\tau \to \tilde{\mathbf{x}}_\tau)|^\gamma \, d\tilde{\mathbf{x}}_\tau d\mathbf{x}_\tau \right)^{\frac{1}{\gamma}} \tag{54}$$

Since there exist $\alpha, \beta$, and $\gamma$ such that $q_\tau(\mathbf{x}_\tau) \in \mathcal{L}^\alpha$, $\log \left( \frac{p_\tau^{\boldsymbol{\theta},\boldsymbol{\phi}^*,(l)}(\mathbf{x}_\tau)}{q_\tau(\mathbf{x}_\tau)} \right) \in \mathcal{L}^\beta$, and $T_\tau \in \mathcal{L}^\gamma$ holds, Eq. 52 is absolute convergence. Therefore, by Fubini's theorem, the order of integration can be interchanged.

$$= -\int \log \left( \frac{p_\tau^{\boldsymbol{\theta},\boldsymbol{\phi}^*,(l)}(\tilde{\mathbf{x}}_\tau)}{q_\tau(\tilde{\mathbf{x}}_\tau)} \right) \left[ \int q_\tau(\mathbf{x}_\tau) T(\mathbf{x}_\tau \to \tilde{\mathbf{x}}_\tau) \, d\mathbf{x}_\tau \right] d\tilde{\mathbf{x}}_\tau \tag{55}$$

$$= -\int \log \left( \frac{p_\tau^{\boldsymbol{\theta},\boldsymbol{\phi}^*,(l)}(\tilde{\mathbf{x}}_\tau)}{q_\tau(\tilde{\mathbf{x}}_\tau)} \right) q_\tau(\tilde{\mathbf{x}}_\tau) \, d\tilde{\mathbf{x}}_\tau \tag{56}$$

$$= D_{\mathrm{KL}}[q_\tau \,\|\, p_\tau^{\boldsymbol{\theta},\boldsymbol{\phi}^*,(l)}] \tag{57}$$

$$\square$$

The assumption in Theorem 4.4 is made solely to satisfy Fubini's theorem, and we note that the theorem is commonly adopted in prior works (De Bortoli et al., 2021; Lipman et al., 2023).

## B    METROPOLIS HASTINGS ALGORITHM IN JOINT SPACE

In DLG (Kim & Ye, 2023), a time classifier was proposed to detect whether a sample had left the manifold after Langevin dynamics at timestep $t$. The proposal distribution of DLG is as follows:

$$p_{\mathrm{proposal}}^{\boldsymbol{\theta},\boldsymbol{\psi}}(\tilde{\mathbf{x}}, \tilde{t} \mid \mathbf{x}, t) = p_{\mathrm{proposal}}^{\boldsymbol{\theta}}(\tilde{\mathbf{x}} \mid \mathbf{x}, t) \cdot p_{\mathrm{proposal}}^{\boldsymbol{\psi}}(\tilde{t} \mid \mathbf{x}, t, \tilde{\mathbf{x}}) = p_{\mathrm{proposal}}^{\boldsymbol{\theta}}(\tilde{\mathbf{x}} \mid \mathbf{x}, t) \cdot p_{\mathrm{proposal}}^{\boldsymbol{\psi}}(\tilde{t} \mid \tilde{\mathbf{x}}) \tag{58}$$

First, given $(\mathbf{x}, t)$, sample $\tilde{\mathbf{x}}$ using one step of Langevin dynamics. After that, sample $\tilde{t}$ using the time classifier conditioned on $\tilde{\mathbf{x}}$. The proposal distribution in the joint space depends not only on the score network but also on the time classifier.

However, as we showed in Proposition A.1, this approach cannot converge to the true joint distribution even when the time classifier is optimal, i.e., $p^{\boldsymbol{\psi}^*}(t|\mathbf{x}) = q(t|\mathbf{x})$. To address this issue, we perform the Metropolis-Hastings algorithm in the joint space of time and data. To compute the acceptance probability in the joint distribution, we extend the density ratio formulation presented in Theorem 4.1. This extension is proposed in the following corollary.

**Corollary B.1.** *Let $\mathbf{x}, \tilde{\mathbf{x}}$ be arbitrary samples at diffusion timesteps $t, \tau$, respectively, and let $\mathbf{x}_\tau$ be any fixed point at timestep $\tau$. If $\tau < \min(t, \tilde{t})$ is satisfied, the density ratio of the true joint distribution $q(\cdot, \cdot)$ is given by:*

$$\frac{q(\tilde{\mathbf{x}}, \tilde{t})}{q(\mathbf{x}, t)} = \frac{q_{\tilde{t}|\tau}(\tilde{\mathbf{x}}|\mathbf{x}_\tau)}{q_{t|\tau}(\mathbf{x}|\mathbf{x}_\tau)} \cdot \frac{L_{\tilde{t}}(\tilde{\mathbf{x}}, \tilde{t})}{L_t(\mathbf{x}, t)} \cdot \frac{p_{\tau|t}^{\boldsymbol{\theta}}(\mathbf{x}_\tau|\mathbf{x})}{p_{\tau|\tilde{t}}^{\boldsymbol{\theta}}(\mathbf{x}_\tau|\tilde{\mathbf{x}})} \tag{59}$$

---

**Algorithm 2** `JointMALAOneStep`$(\mathbf{x}, t, \mathbf{s}, L_t, \boldsymbol{\theta}, \boldsymbol{\phi}, \boldsymbol{\psi})$

---

**Input:** $\mathbf{x}, t, \mathbf{s}, L_t$
**Output:** Accepted sample $\tilde{\mathbf{x}}_\tau$
1: **repeat**
2:    Sample $\tilde{\mathbf{x}} \sim p_{\text{proposal},t}^{\boldsymbol{\theta}}(\cdot \mid \mathbf{x}, t)$
3:    Sample $\tilde{t} \sim p_{\text{proposal}}^{\boldsymbol{\psi}}(\cdot \mid \tilde{\mathbf{x}})$
4:    Compute $\tilde{\mathbf{s}} = \mathbf{s}^{\boldsymbol{\theta}}(\tilde{\mathbf{x}}, \tilde{t})$ and $\tilde{L}_{\tilde{t}} = L_{\tilde{t}}^{\phi}(\tilde{\mathbf{x}}, \tilde{t})$
5:    Compute acceptance probability: $\hat{\alpha}_{joint}(\mathbf{x}, \tilde{\mathbf{x}}, \mathbf{s}, \tilde{\mathbf{s}}, L_t, \tilde{L}_{\tilde{t}}, \tau)$
6:    Sample $u \sim \mathcal{U}(0, 1)$
7: **until** $u < \alpha$
8: **return** $\tilde{\mathbf{x}}, \tilde{t}, \tilde{\mathbf{s}}, \tilde{L}_{\tilde{t}}$

---

*Proof.* For $\forall \mathbf{x}_\tau$ with $\tau < t, \tilde{t}$, the below equation holds.

$$\frac{q(\tilde{\mathbf{x}}, \tilde{t})}{q(\mathbf{x}, t)} = \frac{q(\tilde{\mathbf{x}} \mid \tilde{t})}{q(\mathbf{x} \mid t)} \cdot \frac{q(\tilde{t})}{q(t)} \tag{60}$$

$$= \frac{q_{\tilde{t}}(\tilde{\mathbf{x}})}{q_t(\mathbf{x})} \tag{61}$$

$$= \frac{q_{\tilde{t}|\tau}(\tilde{\mathbf{x}} \mid \mathbf{x}_\tau)}{q_{t|\tau}(\mathbf{x} \mid \mathbf{x}_\tau)} \cdot \frac{q_{\tau|t}(\mathbf{x}_\tau|\mathbf{x})}{q_{\tau|\tilde{t}}(\mathbf{x}_\tau|\tilde{\mathbf{x}})} \cdot \frac{q_\tau(\mathbf{x}_\tau)}{q_\tau(\mathbf{x}_\tau)} \tag{62}$$

$$= \frac{q_{\tilde{t}|\tau}(\tilde{\mathbf{x}} \mid \mathbf{x}_\tau)}{q_{t|\tau}(\mathbf{x} \mid \mathbf{x}_\tau)} \cdot \frac{q_{\tau|t}(\mathbf{x}_\tau|\mathbf{x})}{q_{\tau|\tilde{t}}(\mathbf{x}_\tau|\tilde{\mathbf{x}})} \tag{63}$$

$$= \frac{q_{\tilde{t}|\tau}(\tilde{\mathbf{x}} \mid \mathbf{x}_\tau)}{q_{t|\tau}(\mathbf{x} \mid \mathbf{x}_\tau)} \cdot \frac{q_{\tau|t}(\mathbf{x}_\tau|\mathbf{x})}{p_{\tau|t}^{\boldsymbol{\theta}}(\mathbf{x}_\tau|\mathbf{x})} \cdot \frac{p_{\tau|\tilde{t}}^{\boldsymbol{\theta}}(\mathbf{x}_\tau|\tilde{\mathbf{x}})}{q_{\tau|\tilde{t}}(\mathbf{x}_\tau|\tilde{\mathbf{x}})} \cdot \frac{p_{\tau|t}^{\boldsymbol{\theta}}(\mathbf{x}_\tau|\mathbf{x})}{p_{\tau|\tilde{t}}^{\boldsymbol{\theta}}(\mathbf{x}_\tau|\tilde{\mathbf{x}})} \tag{64}$$

$$= \frac{q_{\tilde{t}|\tau}(\tilde{\mathbf{x}} \mid \mathbf{x}_\tau)}{q_{t|\tau}(\mathbf{x} \mid \mathbf{x}_\tau)} \cdot \frac{L_\tau(\mathbf{x}_\tau, \tau)}{L_t(\mathbf{x}, t)} \cdot \frac{L_{\tilde{t}}(\tilde{\mathbf{x}}, \tilde{t})}{L_\tau(\mathbf{x}_\tau, \tau)} \cdot \frac{p_{\tau|t}^{\boldsymbol{\theta}}(\mathbf{x}_\tau|\mathbf{x})}{p_{\tau|\tilde{t}}^{\boldsymbol{\theta}}(\mathbf{x}_\tau|\tilde{\mathbf{x}})} \tag{65}$$

$$= \frac{q_{\tilde{t}|\tau}(\tilde{\mathbf{x}} \mid \mathbf{x}_\tau)}{q_{t|\tau}(\mathbf{x} \mid \mathbf{x}_\tau)} \cdot \frac{L_{\tilde{t}}(\tilde{\mathbf{x}}, \tilde{t})}{L_t(\mathbf{x}, t)} \cdot \frac{p_{\tau|t}^{\boldsymbol{\theta}}(\mathbf{x}_\tau|\mathbf{x})}{p_{\tau|\tilde{t}}^{\boldsymbol{\theta}}(\mathbf{x}_\tau|\tilde{\mathbf{x}})} \tag{66}$$

$\square$

The density of the proposal distribution, $p_{\text{proposal}}^{\boldsymbol{\theta}}(\tilde{\mathbf{x}} \mid \mathbf{x}, t) \cdot p_{\text{proposal}}^{\boldsymbol{\psi}}(\tilde{t} \mid \tilde{\mathbf{x}})$, is tractable. In detail, $p_{\text{proposal}}^{\boldsymbol{\theta}}(\tilde{\mathbf{x}} \mid \mathbf{x}, t)$ is a Langevin proposal, which follows a Gaussian distribution and $p_{\text{proposal}}^{\boldsymbol{\psi}}(\tilde{t} \mid \tilde{\mathbf{x}})$ can be evaluated using the output of the time classifier. Therefore, the acceptance probability in the joint space can be computed. The acceptance probability is given as follows:

$$\hat{\alpha}_{joint}(\mathbf{x}, \tilde{\mathbf{x}}, \mathbf{s}, \tilde{\mathbf{s}}, L, \tilde{L}, \tau)$$

$$= \min\left( 1, \underbrace{\frac{q_{\tilde{t}|\tau}(\tilde{\mathbf{x}} \mid \hat{\mathbf{x}}_\tau)}{q_{t|\tau}(\mathbf{x} \mid \hat{\mathbf{x}}_\tau)}}_{\text{Forward term}} \cdot \underbrace{\frac{\tilde{L}}{L}}_{\text{Likelihood ratio}} \cdot \underbrace{\frac{p_{\tau|t}^{\boldsymbol{\theta}}(\mathbf{x}_\tau|\mathbf{x})}{p_{\tau|\tilde{t}}^{\boldsymbol{\theta}}(\mathbf{x}_\tau|\tilde{\mathbf{x}})}}_{\text{Transition kernel term}} \cdot \underbrace{\frac{p_{\text{proposal}}^{\boldsymbol{\theta}}(\mathbf{x} \mid \tilde{\mathbf{x}}, \tilde{t}) \cdot p_{\text{proposal}}^{\boldsymbol{\psi}}(t \mid \mathbf{x})}{p_{\text{proposal}}^{\boldsymbol{\theta}}(\tilde{\mathbf{x}} \mid \mathbf{x}, t) \cdot p_{\text{proposal}}^{\boldsymbol{\psi}}(\tilde{t} \mid \tilde{\mathbf{x}})}}_{\text{Proposal term}} \right) \tag{67}$$

$\tau, \hat{\mathbf{x}}_\tau$ can be any point. We choose the value of $\tau$ such that it does not deviate significantly from the original timestep $t$. In our experiments, we empirically set $\tau$ so that the standard deviation of $q_{\tau|0}(\mathbf{x}_\tau \mid \mathbf{x}_0)$ differs from that of $q_{\min(t,\tilde{t})|0}(\mathbf{x}_{\min(t,\tilde{t})} \mid \mathbf{x}_0)$ by 0.1, based on the VESDE(Song et al., 2021b) parameterization. We set $\hat{\mathbf{x}}_\tau = \frac{1}{2}(\boldsymbol{\mu}_t(\mathbf{x}, \mathbf{s}^{\boldsymbol{\theta}}(\mathbf{x}, t)) + \boldsymbol{\mu}_{\tilde{t}}(\tilde{\mathbf{x}}, \mathbf{s}^{\boldsymbol{\theta}}(\tilde{\mathbf{x}}, \tilde{t})))$. The detailed process is in Algorithm 2.

Since the timestep is proposed for every update of Alg. 2 , we need to reassign the starting timestep $t$ for denoising. We first fix the total number of steps $T$, and perform $T - t$ steps of denoising

Table 9: FID and NFE on unconditional CelebA-HQ 256×256 generation with ScoreSDE (Song et al., 2021b), DLG (Kim & Ye, 2023), AC (marginal), and AC (joint).

| | FID↓ | NFE↓ | FID↓ | NFE↓ | FID↓ | NFE↓ |
|---|---|---|---|---|---|---|
| ScoreSDE (KAR1) | 121.27 | 40 | 122.74 | 98 | 125.15 | 198 |
| +DLG | 20.19 | 21.21 | 29.12 | 47.21 | 30.72 | 107.21 |
| +AC (marginal) | 103.81 | 19.78 | 75.51 | 41.83 | 87.40 | 84.13 |
| +AC (joint) | **15.13** | **15.94** | **22.55** | **44.06** | **15.69** | **87.26** |
| ScoreSDE (KAR2) | 83.21 | 40 | 57.28 | 98 | 29.74 | 198 |
| +DLG | 17.92 | 21.21 | 12.12 | 47.21 | 8.14 | 107.21 |
| +AC (marginal) | 45.74 | 19.55 | 19.48 | 42.97 | 9.45 | 94.34 |
| +AC (joint) | **8.45** | **20.05** | **9.55** | **40.05** | **6.60** | **98.27** |

from $t$ down to the proposed timestep $\tilde{t}$. Then, we perform $t$ steps of denoising from $\tilde{t}$ to 0. Our methodology generalizes the approach of DLG (Kim & Ye, 2023). While DLG generates samples from the joint space of time and data using Gibbs sampling, we introduce the Metropolis-Hastings algorithm to correct samples toward the true data distribution by additionally training a time-dependent discriminator. We adopt the time classifier from the official code of DLG and use the $argmax$ of the classifier output as the proposed timestep, following their original approach. Since using the $argmax$ results in a deterministic time proposal distribution, we set $p^{\psi}(t \mid \mathbf{x}) = 1$ when computing the acceptance probability.

For fair comparison, we reproduced the experimental setting of DLG. We first obtained the best parameters for both the KAR1 and KAR2 samplers as reported in DLG, and then reproduced their performance using these optimal settings. Subsequently, we increased the number of denoising steps while keeping the remaining parameters unchanged.

## C  HYPERPARAMETER DETAIL

As described in the main text, our method treats the MH target diffusion timestep $\tau$ and the Langevin step size (controlled by the signal-to-noise ratio, SNR) as the primary parameters. In addition, we employ several auxiliary hyperparameters: the number of skipped steps $n_{\text{skip}}$, the burn-in length $n_{\text{burn-in}}$, and the number of parallel chains $n_{\text{chain}}$. Their roles are summarized as follows:

- $n_{\text{skip}}$: Controls how many intermediate steps are skipped between proposals.
- $n_{\text{burn-in}}$: Specifies the number of initial iterations discarded to reduce initialization bias.
- $n_{\text{chain}}$: Denotes the length of MCMC chains. With one initial point, we can get $n_{\text{chain}}$ samples.

Among the hyperparameters, we regard the choice of $\tau$ as the most critical. As $\tau$ decreases—i.e., as the state approaches the data distribution—the marginal distribution becomes sharper. This sharpness increases the computational burden of moving across the space via MCMC. While smaller $\tau$ brings the chain closer to the true data distribution (see proof of Theorem 4.2), it also requires a larger number of samples to sufficiently cover the support. Conversely, if $\tau$ is set too low, the effective reduction in NFE diminishes and distributional alignment becomes less pronounced. Therefore, selecting an appropriate $\tau$ is essential. Empirically, we found that setting $\tau$ between $\frac{1}{2}T$ and $\frac{3}{4}T$ achieves the most effective trade-off.

The second key parameter is the SNR, which controls the step size $\eta$ of the Langevin proposal:

$$\sqrt{\eta} = \text{SNR} \times \left( \frac{2 \cdot |\boldsymbol{\epsilon}|}{|\mathbf{s}|} \right). \tag{68}$$

A too-small SNR yields excessively small step sizes, limiting sample diversity, while a too-large SNR hampers convergence of the MH correction. Based on prior works that adopted Langevin sampling in diffusion models (e.g., Song & Ermon (2019); Song et al. (2021b)), we set the SNR in the range of 0.1–0.25.

Experimental results on varying these two key parameters are reported in Table 10. We conducted experiments with the total number of timesteps fixed at 18, while keeping $n_{\text{skip}}$, $n_{\text{burn-in}}$, and $n_{\text{chain}}$ constant. As shown in the Table, when $\tau$ is set too small, the MCMC chain tends to remain in a limited region of the space for a long time. Consequently, covering the entire distribution requires significantly higher computational cost, which is reflected in the degraded FID and Recall metrics. Furthermore, as the SNR increases, the acceptance probability gradually decreases. Across our overall experimental setup, the SNR satisfies $\text{SNR} \leq 0.25$, which allows us to maintain an acceptance probability of approximately $\alpha \gtrsim 0.25$ (see Table 10). Because $\alpha$ is sufficiently large, as argued in the main text, substantial NFE reduction can be achieved even with a small chain length $l$. These experimental results support the preceding analysis in Proposition 4.2.

| Method | $\tau$ | SNR | FID↓ | NFE↓ | Recall↑ | C.I of $\alpha$ |
|---|---|---|---|---|---|---|
| EDM (Heun) | - | - | 2.01 | 35 | 0.627 | - |
| +AC | 13 | 0.1 | 2.89 | 25.13 | 0.550 | $0.9150 \pm 0.0023$ |
| | | 0.2 | 2.06 | 25.65 | 0.620 | $0.6248 \pm 0.0033$ |
| | | 0.23 | 1.97 | 26.19 | 0.628 | $0.4703 \pm 0.0030$ |
| | | 0.27 | 2.09 | 28.11 | 0.625 | $0.2493 \pm 0.0019$ |
| +AC | 8 | 0.1 | 8.77 | 15.21 | 0.200 | $0.8689 \pm 0.0027$ |
| | | 0.2 | 6.14 | 15.75 | 0.422 | $0.6560 \pm 0.0039$ |
| | | 0.23 | 6.73 | 16.70 | 0.441 | $0.4671 \pm 0.0034$ |
| | | 0.27 | 9.45 | 18.66 | 0.448 | $0.2733 \pm 0.0022$ |
| +AC | 3 | 0.1 | 62.67 | 5.29 | 0 | $0.8525 \pm 0.0046$ |
| | | 0.2 | 46.60 | 5.75 | 0 | $0.6590 \pm 0.0054$ |
| | | 0.23 | 43.96 | 6.26 | 0 | $0.4812 \pm 0.0054$ |
| | | 0.27 | 39.44 | 9.13 | 0 | $0.2703 \pm 0.0022$ |

Table 10: Results for different $\tau$ values and SNR settings, including FID, NFE, Recall, and acceptance probabilities with 95% confidence intervals.

| Method | $n_{\text{skip}}$ | FID↓ | NFE↓ | Recall↑ |
|---|---|---|---|---|
| EDM (Heun) | - | 2.01 | 35 | 0.627 |
| + AC | 0 | 1.97 | 26.19 | 0.628 |
| | 1 | 1.98 | 28.25 | 0.631 |
| | 2 | 1.94 | 30.28 | 0.634 |
| | 3 | 1.97 | 32.30 | 0.638 |
| | 4 | 2.00 | 34.39 | 0.640 |
| | 5 | 2.02 | 36.38 | 0.623 |

Table 11: FID and NFE for different skip steps.

| Method | $n_{\text{chain}}$ | FID↓ | NFE↓ | Recall↑ |
|---|---|---|---|---|
| EDM (Heun) | - | 2.01 | 35 | 0.627 |
| + AC | 10 | 2.03 | 29.28 | 0.622 |
| | 50 | 2.00 | 26.69 | 0.630 |
| | 100 | 2.07 | 26.38 | 0.624 |
| | 300 | 1.97 | 26.19 | 0.629 |
| | 500 | 2.02 | 26.12 | 0.625 |

Table 12: FID and NFE for different chain steps.

| Method | $n_{\text{burn-in}}$ | FID↓ | NFE↓ | Recall↑ |
|---|---|---|---|---|
| EDM (Heun) | - | 2.01 | 35 | 0.627 |
| + AC | 0 | 1.99 | 36.23 | 0.640 |
| | 1 | 2.02 | 37.46 | 0.624 |
| | 2 | 2.01 | 38.69 | 0.633 |
| | 5 | 1.99 | 42.37 | 0.632 |
| | 10 | 1.97 | 48.54 | 0.629 |
| | 20 | 1.97 | 60.81 | 0.632 |
| | 50 | 1.98 | 97.58 | 0.629 |

Table 13: FID and NFE for different burn-in steps.

The following reports the results of varying each auxiliary parameter. Tables 11, 12, and 13 present the outcomes for changing $n_{\text{skip}}$, $n_{\text{chain}}$, and $n_{\text{burn-in}}$, respectively. Unless otherwise noted, all experiments are conducted with $T = 18$, $\tau = 13$.

Table 11 reports the effect of varying $n_{\text{skip}}$ while fixing $\text{SNR} = 0.23$, $n_{\text{chain}} = 300$, and $n_{\text{burn-in}} = 10$. The parameter $n_{\text{skip}}$ helps reduce autocorrelation between samples; however, excessively large values increase the NFE, limiting the achievable acceleration gain. Empirically, we set $n_{\text{skip}} = 0 \sim 1$ for CIFAR-10 and ImageNet, and maximum 4 for CelebA-HQ $256 \times 256$.

Table 12 shows the results obtained by varying $n_{\text{chain}}$ while fixing $\text{SNR} = 0.23$ and $n_{\text{burn-in}} = 10$. When SNR is too small, recall may vary with $n_{\text{chain}}$, but under reasonable SNR values the recall is largely insensitive to $n_{\text{chain}}$. Nevertheless, setting $n_{\text{chain}}$ too small can hinder effective NFE reduction.

Table 13 investigates the role of the burn-in process by varying $n_{\text{burn-in}}$ while fixing $\text{SNR} = 0.16$ and $n_{\text{chain}} = 1$. We set $n_{\text{chain}} = 1$ in order to isolate and examine the effect of correction on each sample. We observe that after about 10 burn-in steps, the chain sufficiently converges, indicating that the score-based proposal distribution indeed allows proper convergence. In practice, we set $n_{\text{burn-in}} \leq 10$.

# D   MCMC MIXING

Images are high-dimensional data, which makes direct statistical evaluation of Markov chain mixing challenging. To address this, we assess mixing indirectly by analyzing the class labels of generated

images. We trained a ResNet based classifier that achieves 95% accuracy on the CIFAR-10 test set, and used it to assign class labels to each generated image, thus forming a class sequence along the MCMC chain. We constructed Markov chains of length 300.

We measured Integrated Autocorrelation Time(IACT) (Birdsall et al., 1994) 30 times with a maximum lag of 100. With Table 15 we observed trends consistent with Recall metrics. Lower IACT values indicate better mixing, suggesting that our method yields well-mixed samples. However, we note that IACT is originally defined for continuous variables, and applying it to categorical class labels can be limiting.

To complement IACT, we also computed Cramér's V (Akoglu, 2018) 30 times to assess class autocorrelation in the discrete label space in Table 14. Under the best-performing setting ($\tau = 5, \text{SNR} = 0.23$), the value at lag 1 shows a relatively strong correlation (Akoglu, 2018), which is expected since our sampler proposes candidates based on local gradients. Nevertheless, both improvements in the Recall metric and our toy experiment 5.3 demonstrate that, despite such correlations, the chain is able to generate sufficiently diverse samples.

| Lag | Cramér's V $\pm$ std |
|---|---|
| 1 | $0.360 \pm 0.035$ |
| 2 | $0.259 \pm 0.037$ |
| 3 | $0.214 \pm 0.044$ |
| 4 | $0.182 \pm 0.042$ |
| 5 | $0.172 \pm 0.036$ |

Table 14: Cramér's V across lags.

| Method | $\tau$ | SNR | FID $\downarrow$ | Recall $\uparrow$ | IACT of class sequence $\downarrow$ |
|---|---|---|---|---|---|
| EDM (Heun) | - | - | 2.01 | 0.627 | - |
| +AC | 13 | 0.1 | 2.89 | 0.550 | $8.71 \pm 10.02$ |
| | | 0.2 | 2.06 | 0.620 | $2.45 \pm 3.23$ |
| | | 0.23 | 1.97 | 0.628 | $1.53 \pm 1.70$ |
| | | 0.27 | 2.09 | 0.625 | $1.89 \pm 1.35$ |
| +AC | 8 | 0.1 | 8.77 | 0.200 | $29.47 \pm 26.86$ |
| | | 0.2 | 6.14 | 0.422 | $21.65 \pm 24.84$ |
| | | 0.23 | 6.73 | 0.441 | $20.92 \pm 24.50$ |
| | | 0.27 | 9.45 | 0.448 | $24.39 \pm 25.55$ |
| +AC | 3 | 0.1 | 62.67 | 0 | $38.45 \pm 27.39$ |
| | | 0.2 | 46.60 | 0 | $34.93 \pm 27.83$ |
| | | 0.23 | 43.96 | 0 | $22.51 \pm 23.15$ |
| | | 0.27 | 39.44 | 0 | $24.45 \pm 32.90$ |

Table 15: IACT of class sequence for different $\tau$ values and SNR settings.

## E  METROPOLIS-HASTINGS ALGORITHM AND ALGORITHM 1

Algorithm 1 employs a propose-until-accept update: at each step, proposals are repeatedly drawn and subjected to the MH accept–reject test until one is accepted, and the accepted proposal is then emitted as the next sample. We adopted this design for empirical reasons, namely to mitigate stagnation and preserve sample diversity. In canonical Metropolis–Hastings, however, a rejection corresponds to a self-transition, which is essential for preserving detailed balance. Eliminating self-transitions by proposing until acceptance alters the transition kernel and can introduce stationary bias.

This variant can be interpreted as a Jump Markov chain (Rosenthal et al., 2021). In such chains, the target distribution is implicitly modified because the rejection mechanism no longer permits self-transitions. Following Rosenthal et al. (2021), the stationary distribution of the jump chain, denoted $\hat{\pi}$, can be expressed in terms of the original stationary distribution $\pi$ as

$$\hat{\pi}(x) = c\alpha(x)\pi(x), \tag{69}$$

where $\alpha(x) := 1 - P_{\text{transition}}(x|x)$ is the escape probability at state $x$, and $c = \mathbb{E}_{y \sim \pi}[\alpha(y)]^{-1}$ is a normalizing constant. Here $P_{\text{transition}}(\cdot|\cdot)$ denotes the transition probability of the original MH chain. The KL divergence between $\pi$ and $\hat{\pi}$ is then

$$D_{\text{KL}}[\pi||\hat{\pi}] = \mathbb{E}_{x \sim \pi}\left[\log \frac{\pi(x)}{\hat{\pi}(x)}\right] = \mathbb{E}_{x \sim \pi}\left[\log \frac{1}{c\alpha(x)}\right] = \mathbb{E}_{x \sim \pi}\left[\log \frac{\mathbb{E}_{y \sim \pi}[\alpha(y)]}{\alpha(x)}\right]. \tag{70}$$

This formulation shows that the jump chain introduces a KL divergence bias. When $\alpha(x)$ is constant over the support of $\pi$, no bias arises; otherwise, the deviation can be non-negligible. Despite the strong empirical performance of our method, a distributional gap remains. We leave a rigorous theoretical analysis of this gap to future work.

# F ADDITIONAL EXPERIMENT

## F.1 EXTEND AC-SAMPLER TO CORRECT EACH SAMPLES : MULTI STEP CORRECTION & REFINED PROPOSAL WITH A DISCRIMINATOR

After the burn-in process of the Metropolis-Hastings algorithm, the samples generated from our method can be regarded as samples drawn from the true distribution. This demonstrates that the Metropolis-Hastings algorithm can be used not only to accelerate sampling, but also to correct intermediate samples to better match the target distribution.

Focusing solely on the correction perspective, our proposed framework naturally incorporates the following methodological components: *multi-step correction* and *refined proposal with a discriminator*. As discussed in the main text, we initially present our algorithm using a single-step formulation for simplicity. However, applying our method in a multi-step setting is straightforward and does not pose any conceptual or technical difficulties. Therefore, we also conducted experiments under the multi-step setting.

Furthermore, DG (Kim et al., 2023) proposed correcting the score network using the gradient information from a discriminator. Since we adopt exactly the same training scheme for the discriminator as in DG, it is reasonable to apply a refined proposal based on the corrected score network. This implies that the discriminator trained at timestep $\tau$ not only provides a likelihood ratio estimate, but also enables refining the proposal distribution $p^{\theta}_{\text{proposal},\tau}$. It is possible to use $DG_p$ in accelerating, but $DG_p$ needs gradient calculation and this made sampling speed slow.

Table 16: Comparison of FID and sampling settings under different configurations

| Sampling | FID↓ | NFE↓ | $T$ | $\tau$ | SNR | $n_{\text{burn-in}}$ | $n_{\text{chain}}$ |
|---|---|---|---|---|---|---|---|
| EDM (Heun) | 1.97 | 35 | 18 | – | – | – | – |
| + PC | 2.18 | 51 | 18 | 1, 3, 5, 7 | 0.16 | 3 | 1 |
| | 2.13 | 51 | 18 | 7, 9, 11, 13 | 0.16 | 3 | 1 |
| | 2.00 | 51 | 18 | 11, 13, 15, 17 | 0.16 | 3 | 1 |
| + AC | 1.94 | 54.66 | 18 | 1, 3, 5, 7 | 0.16 | 3 | 1 |
| | 1.96 | 55.33 | 18 | 7, 9, 11, 13 | 0.16 | 3 | 1 |
| | 1.93 | 54.91 | 18 | 11, 13, 15, 17 | 0.16 | 3 | 1 |
| + AC with $DG_p$ | 1.98 | 54.65 | 18 | 1, 3, 5, 7 | 0.16 | 3 | 1 |
| | **1.87** | 54.56 | 18 | 7, 9, 11, 13 | 0.16 | 3 | 1 |
| | 1.92 | 54.17 | 18 | 11, 13, 15, 17 | 0.16 | 3 | 1 |

We present the results of both extensions in Table 16. We denote this discriminator-guided proposal scheme as $DG_p$. The result demonstrates the effect of MALA correction across various choices of the correction timestep $\tau$. Although the correction timestep increases while maintaining the same SNR, AC-Sampler either improves or maintains the baseline FID score. Furthermore, we observe that incorporating the DG scheme into AC-Sampler leads to a meaningful reduction in FID with shorter length of chain. In contrast, PC-Sampler (Song et al., 2021b) often fails to correct samples at large correction timesteps.

In the table, setting $n_{\text{chain}} = 1$ indicates that Metropolis-Hastings correction is applied for each individual sample. While this setting does not reduce the number of function evaluations (NFE), it effectively corrects each intermediate sample.

## F.2 DISCUSSION ABOUT TRADE-OFF IN CLASS-CONDITIONAL GENERATION

When performing class-conditional generation, the number of independent class samples plays an important role. For FID evaluation, we use 50K images. With $n_{\text{chain}} = 10$, the baseline involves 50,000 independent class samplings, whereas only 5,000 samplings occur with our method, which may lead to class imbalance. If this effect did not exist, the NFE reduction could be even more effective. To evaluate our method fairly under this setting, we generate 250K samples with $n_{\text{chain}} = 5$ (this setting makes total of 50K independent class samplings) and compute the FID five times using randomly selected subsets of 50K samples. We report the mean and standard deviation of the resulting FID. Ta-

Table 17: FID and NFE results on ImageNet 64×64 across different (SNR, $\tau$) settings.

| Method | SNR | $\tau$ | FID↓ | NFE↓ |
|---|---|---|---|---|
| EDM (Heun) | – | – | 2.30 | 61 |
| +AC | 0.12 | 8 | **2.28±0.05** | 52.91 |
| | 0.12 | 9 | 2.29±0.03 | **51.30** |

ble 17 presents this analysis, showing that our method can significantly reduce NFE while maintaining a comparable FID.

## F.3 CIFAR-10 WITH SCORESDE

We also conduct experiments with Score-SDE (Song et al., 2021b) and DLG (Kim & Ye, 2023). Table 18 and Figure 10 reports our reproductions of the base models (Score-SDE and DLG) alongside our method. Because the publicly released Score-SDE checkpoint is configured for sampling with roughly 1,000 NFE, achieving strong performance at substantially lower NFE is inherently challenging. Following the evaluation protocol described in the DLG paper, our reproduced results improved over the base checkpoint but did not exactly match the values reported in the original work. Under the same setting, applying our method yielded consistent distribution correction and quality improvements, even in the low-NFE regime.

| | FID↓ | NFE↓ | FID↓ | NFE↓ | FID↓ | NFE↓ |
|---|---|---|---|---|---|---|
| ScoreSDE (Base) | 27.35 | 16 | 26.58 | 26 | 26.72 | 36 |
| +DLG | 25.95 | 11.23 | 24.86 | 21.23 | 24.25 | 52.23 |
| +AC (Ours) | **25.18** | **10.95** | **23.83** | **19.13** | **23.14** | **29.07** |

Table 18: Experiment results on CIFAR-10 with ScoreSDE

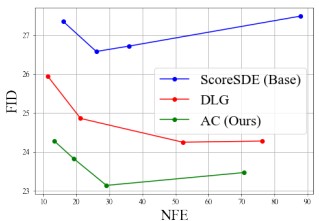

Figure 10: ScoreSDE base experiment on CIFAR-10

## F.4 ADDITIONAL ABLATION STUDIES

**Feature extractor ablation** To verify whether our proposed method relies on the semantic guidance of a pretrained classifier which is used in discriminator to extract feature, we conducted an ablation study using a randomly initialized classifier. In this setting, the discriminator possesses no classification ability or prior semantic knowledge. As shown in Table 19, although utilizing a pretrained ADM classifier yields some performance gain (FID 1.97), our method with a randomly initialized classifier ("AC w/o pre-trained") still achieves an FID of 1.99, outperforming the Base model (2.01) while maintaining improved sampling efficiency (NFE 28.19). This result demonstrates that our approach does not fundamentally depend on the availability of a pre-trained classifier or feature extractor. Note that the use of the ADM classifier as a feature extractor follows standard practices established in prior time-dependent discriminator frameworks, such as DG (Kim et al., 2023) and DiffRS (Na et al., 2024). We further extended this analysis to the Text-to-Image (T2I) model under the same randomly initialized setting. The results for the T2I experiments exhibit a similar trend and are detailed in Table 5.

Table 19: Ablation study on classifier dependency.

| Method | FID ↓ | NFE ↓ |
|---|---|---|
| EDM (Heun) | 2.01 | 35.00 |
| + AC (w/ Pre-trained) | **1.97** | **26.19** |
| + AC (w/ Random Init.) | 1.99 | 28.19 |

**Generating a small number of samples** Our method achieves acceleration gains as $n_{chain}$ increases; specifically, a larger $n_{chain}$ extends the MALA chain, which reduces the average NFE. However, when generating a small number of samples, $n_{chain}$ may not be sufficiently long. Therefore in this setting, our method may not accelerate sampling processes. To empirically quantify the impact of small $n_{chain}$ in this setting, we generated a small number of samples (4 for SD-v1.5 and 10 for CIFAR-10) with a batch size of 1 using various $n_{chain}$ values. As shown in Table 20, our method requires more time than the baseline when $n_{chain}$ is very small. However, once $n_{chain}$ exceeds a certain threshold, it generates samples faster than the baselines. Specifically, we achieved faster speeds than the baseline for CIFAR-10 when $n_{chain} \geq 3$, and for Stable Diffusion when $n_{chain} \geq 4$. While our method may not offer speed advantages for single or dual image generation, it delivers both acceleration and correctness as the number of samples increases ($\geq 2$ or $3$). This highlights its practical applicability for efficient and correct sampling in real-world scenarios.

Table 20: Wall-clock time and NFE analysis in the small sample, low-$n_{\text{chain}}$ regime. We compare the efficiency of our method (+AC) against baselines when generating a small number of images (4 for SD-v1.5, 10 for EDM CIFAR-10).

| (a) Stable Diffusion v1.5 (Generating 4 images) | | | | |
|---|---|---|---|---|
| **Method** | $n_{\text{chain}}$ | NFE | Total Time (s) | **# Images** |
| SD-v1.5 (DDIM) | - | 50 | 34 | 4 |
| + AC | 1 | 58.25 | 48 | 4 |
| + AC | 4 | **42.25** | **31** | 4 |

| (b) CIFAR-10 (Generating 10 images) | | | | |
|---|---|---|---|---|
| **Method** | $n_{\text{chain}}$ | NFE | Total Time (s) | **# Images** |
| EDM (Heun) | - | 35 | 6.82 | 10 |
| + AC | 1 | 37.14 | 7.44 | 10 |
| + AC | 2 | 35.89 | 7.13 | 10 |
| + AC | 3 | **33.98** | **6.81** | 10 |
| + AC | 5 | **32.52** | **6.47** | 10 |

# G  ADDITIONAL RELATED WORK

## G.1  SDE-BASED SAMPLING AND SOLVER-RELATED CORRECTIONS.

Several SDE-based methods mitigate numerical errors by improving the accuracy of the sampler itself: adaptive adjustment of SDE solver hyperparameters to reduce discretization error (Jolicoeur-Martineau et al., 2021), controlled-variance reverse SDEs that share the same marginals for more efficient sampling (Xue et al., 2023), and higher-order SDE solvers to further reduce solver error. (Gonzalez et al., 2023) These works focus on improving the numerical integration quality of the diffusion trajectory.

## G.2  APPROACHES ADDRESSING TRAINING–INFERENCE MISMATCH.

Other works instead target the mismatch between the training distribution and the inputs encountered during inference. Examples include: training diffusion models with input perturbations under an assumed Gaussian mismatch (Ning et al., 2023), and adversarially optimizing perturbations and model parameters to achieve distributionally robust diffusion training. (Wang et al., 2025) These approaches retrain the model to better handle off-manifold inputs during sampling.

## G.3  METROPOLIS-HASTINGS ALGORITHM IN DIFFUSION SAMPLING

The work in Aloui et al. (2024) applied the Metropolis-Hastings (MH) algorithm to the diffusion sampling process. In the paper, an acceptance network was trained via the loss based on detailed balance equation to calculate the acceptance probability, which was then utilized during the sampling procedure. However, their approach suffers from several key limitations:

1. No Distribution Correction: The acceptance probability calculation in (Aloui et al., 2024) involves a target distribution score term within its loss function. This term was approximated using the pre-trained score network. Consequently, the target distribution effectively becomes the model distribution itself, meaning that no true distribution correction is performed.

2. Dependency on Proposal Distribution: The loss function used in (Aloui et al., 2024) is dependent on the proposal distribution. This introduces a major practical drawback: if the proposal distribution is changed (e.g., switching to a different sampler step size or schedule), the acceptance function must be re-trained entirely.

Our proposed method addresses these limitations. By leveraging a discriminator and Theorem 4.1, we ensure that the acceptance probability can be computed regardless of the specific proposal distribution.

This is achieved solely by training the discriminator, eliminating the need to re-learn the acceptance function when the proposal distribution changes. Our method is applied on top of a base sampler, and therefore remains orthogonal to SDE-based solver improvements while preserving their advantages. Moreover, because our approach corrects generated samples solely through discriminator training, it mitigates the training–inference mismatch without requiring any additional training of the underlying diffusion model.

## H  EXPERIMENT SETTING

### H.1  EXPERIMENTAL SETUP

**Setups.** We evaluate on CIFAR-10, CelebA-HQ 256×256, ImageNet 64×64, and ImageNet 256×256. On CIFAR-10, we assess EDM (Karras et al., 2022) and DDO (Zheng et al., 2025) using the Heun sampler as in EDM, ScoreSDE (Song et al., 2021b) adopting samplers KAR1 (deterministic) and KAR2 (stochastic) (Kim & Ye, 2023). For ScoreSDE, refer Appendix F.3. On CelebA-HQ, we use the ScoreSDE (Song et al., 2021b) checkpoint within the DLG codebase (Kim & Ye, 2023) with KAR1, KAR2 sampler. On ImageNet 64×64, we use the EDM checkpoint with the SDE sampler from (Karras et al., 2022); on ImageNet 256×256, we use the DiT checkpoint (Peebles & Xie, 2023) with a DDPM sampler (Ho et al., 2020). For text-to-image generation, we use the SD v1.5 checkpoint (Peebles & Xie, 2023) with a DDIM sampler (Song et al., 2021a).

**Codebases and checkpoints.** Our experiments use the official repositories of EDM[2], DLG[3], DG[4],[5], DDO[6], ScoreSDE[7], DiT[8], and SD-v1.5[9].

**Discriminator training.** We train a time-dependent discriminator per network following DG (Kim et al., 2023) and DiffRS (Na et al., 2024), using the pre-trained ADM classifier (Dhariwal & Nichol, 2021) as the feature extractor. We use random initialized feature extractor with no training (same structure, no classifying ability at all) for T2I case since there is no ADM classifier for latent diffusion model. We only train discriminator. Compared to training a diffusion model, discriminator training is substantially cheaper.

**Metrics.** We report FID and the mean number of function evaluations (NFE) of the score network (as in DLG (Kim & Ye, 2023)), since NFE varies across samples in our method. FID is computed on 50K generated samples against the 50K test images; for CelebA-HQ 256×256, we report 10K FID. For COCO (Lin et al., 2014) generation task, we report 5K FID. Also we report Precision / Recall metric to assess both the fidelity and diversity of generated images. The computation of FID follows the official implementation provided by DG (Kim et al., 2023). We measure the Precision and Recall using the ADM codebase (Dhariwal & Nichol, 2021). For text-to-image generation, we evaluate Clip Score (Hessel et al., 2021), ImageReward (Xu et al., 2023a), and GenEval score (Ghosh et al., 2023).

All experiments were conducted on NVIDIA RTX 3090, 4090 GPU and A100 GPU using Python 3.8, PyTorch 1.12, and CUDA 11.4.

### H.2  DISCRIMINATOR DETAILS

To implement the time-dependent discriminator, we directly used the official DG codebase and followed their approach. On CIFAR-10, we used the discriminator checkpoint provided by DG only when the base diffusion model was EDM and the NFE of the EDM (Heun) sampler was set to 35. For all other cases, we trained the discriminator ourselves using the DG codebase.

---

[2] https://github.com/NVlabs/edm
[3] https://github.com/1202kbs/DMCMC
[4] https://github.com/aailabkaist/DG
[5] https://github.com/alsdudrla10/DG_imagenet
[6] https://github.com/NVlabs/DDO
[7] https://github.com/yang-song/score_sde_pytorch
[8] https://github.com/facebookresearch/DiT
[9] https://huggingface.co/stable-diffusion-v1-5/stable-diffusion-v1-5

Table 21: Sampling configuration and performance metrics (FID / NFE) with various diffusion and sampler combinations.

| Dataset | Task | Base Model | Base Sampler | $T$ | SNR | $n_{chain}$ | $n_{burn-in}$ | $n_{skip}$ | $\tau$ | FID | NFE |
|---|---|---|---|---|---|---|---|---|---|---|---|
| CIFAR-10 | Uncond. | EDM | EDM (Heun) | 18 | 0.23 | 50 | 10 | 0 | 11 | 2.10 | 22.78 |
| | | | | 18 | 0.23 | 300 | 10 | 0 | 13 | 1.97 | 26.19 |
| | | | | 18 | 0.23 | 500 | 10 | 0 | 13 | 2.02 | 26.12 |
| | | | | 18 | 0.23 | 3,4,5 | 5 | 4 | 13,14,15 | 1.93 | 44.40 |
| | | | | 14 | 0.2 | 50 | 0 | 0 | 6 | 2.38 | 15.82 |
| | | | | 10 | 0.2 | 11 | 0 | 0 | 5 | 3.24 | 10.57 |
| CIFAR-10 | Uncond. | DDO | EDM (Heun) | 16 | 0.175 | 2 | 0 | 2 | 13 | 1.41 | 29.41 |
| CIFAR-10 | Uncond. | EDM | DPM-Solver-v3 | 6 | 0.16 | 10 | 0 | 0 | 5 | 7.12 | 5.62 |
| | | | | 8 | 0.16 | 10 | 0 | 0 | 7 | 3.09 | 7.54 |
| | | | | 8 | 0.1 | 7 | 0 | 0 | 4 | 9.88 | 4.78 |
| | | | | 10 | 0.15 | 3 | 0 | 0 | 9 | 2.55 | 9.93 |
| CIFAR-10 | Uncond. | ScoreSDE | KAR1 | 18 | 0.23 | 250 | 10 | 1 | 13 | 23.14 | 29.08 |
| | | | | 18 | 0.23 | 250 | 10 | 1 | 11 | 23.80 | 25.18 |
| | | | | 18 | 0.23 | 250 | 10 | 0 | 14 | 23.83 | 19.13 |
| ImageNet 64×64 | Cond. | EDM | EDM (SDE) | 32 | 0.16 | 2 | 1 | 0 | 26 | 2.25 | 58.75 |
| | | | | 64 | 0.18 | 2 | 5 | 1 | 50 | 1.77 | 121.98 |
| | | | | 256 | 0.1 | 2 | 5 | 1 | 225 | 1.42 | 483.86 |
| CelebA-HQ 256×256 | Uncond. | ScoreSDE | KAR1 | 20 | 0.16 | 25 | 10 | 0 | 13 | 15.13 | 15.94 |
| | | | | 49 | 0.16 | 100 | 10 | 4 | 30 | 22.55 | 44.07 |
| | | | | 99 | 0.16 | 25 | 10 | 3 | 60 | 15.69 | 87.26 |
| CelebA-HQ 256×256 | Uncond. | ScoreSDE | KAR2 | 20 | 0.16 | 25 | 10 | 1 | 12 | 8.45 | 20.05 |
| | | | | 49 | 0.16 | 50 | 10 | 4 | 33 | 9.55 | 40.05 |
| | | | | 99 | 0.18 | 25 | 10 | 3 | 55 | 6.60 | 98.27 |
| ImageNet 256×256 | Cond. | DiT | DDPM | 250 | 0.12 | 2 | 190 | 0 | 10 | 2.31 | 234.38 |
| COCO | Cond. | Stable Diffusion 1.5 | DDIM | 50 | 0.18 | 4 | 0 | 5 | 22 | 23.16 | 45.24 |

Our discriminators were trained on a single NVIDIA RTX 3090 GPU. For the feature extractor, we used a commonly adopted (Na et al., 2024; Kim et al., 2023) pre-trained classifier from ADM[10] (Dhariwal & Nichol, 2021). The discriminator takes the features extracted by this network as input, and during training, we only updated the parameters of the discriminator network. Detailed training settings are provided in Table 22.

Indeed, while our method requires training an additional discriminator, we would like to emphasize that the training cost is significantly lower compared to that of the score model. As summarized in the table 23, our discriminator is much smaller and faster to train than the pre-trained score network. Compared to fine-tuning a pre-trained diffusion model, our approach introduces substantially lower computational overhead and does not modify the pre-trained model in any way. Importantly, we enable both acceleration and correction purely through discriminator training only. Table.24 shows the robustness of our time-dependent discriminator. To evaluate the performance of our discriminator at different timesteps, we conducted experiments with $n_{chain} = 1$ ensuring that each sample is corrected independently. (No acceleration was applied, as our goal was to isolate the effect of the discriminator across timesteps) Using a fixed SNR, we applied the AC-Sampler at various $\tau$ values with the same random seed. The results consistently showed improvements in FID across timesteps, suggesting that the discriminator effectively approximates the density ratio $\frac{q_\tau}{p_\tau^\theta}$ at multiple temporal locations.

Table 24: FID and NFE comparison of AC with different $\tau$.

| Method | $\tau$ | FID↓ | NFE↓ |
|---|---|---|---|
| EDM | - | 2.01 | 35 |
| +AC | 3 | 2.02 | 48.51 |
| | 5 | 1.97 | 48.53 |
| | 7 | 1.99 | 48.54 |
| | 10 | 2.00 | 48.72 |
| | 12 | 1.92 | 49.89 |
| | 15 | 2.00 | 49.03 |

---

[10]https://github.com/openai/guided-diffusion

Table 22: Configurations of the discriminator.

| | CIFAR-10 | | | ImageNet64 | CelebA-HQ256 | ImageNet256 | COCO |
|---|---|---|---|---|---|---|---|
| **Diffusion Backbone** | | | | | | | |
| Model | EDM | DDO | ScoreSDE | EDM | ScoreSDE | DiT-XL/2 | Stable Diffusion |
| Conditional model | ✗ | ✗ | ✗ | ✔ | ✗ | ✔ | ✔ |
| **Feature Extractor** | | | | | | | |
| Model | ADM | ADM | ADM | ADM | ADM | ADM | ADM |
| Architecture | U-Net encoder | U-Net encoder | U-Net encoder | U-Net encoder | U-Net encoder | U-Net encoder | U-Net encoder |
| Pre-trained | ✔ | ✔ | ✔ | ✔ | ✔ | ✔ | ✗ |
| Depth | 4 | 4 | 4 | 4 | 4 | 4 | 4 |
| Width | 128 | 128 | 128 | 128 | 128 | 128 | 128 |
| Attention Resolutions | 32,16,8 | 32,16,8 | 32,16,8 | 32,16,8 | 32,16,8 | 32,16,8 | 32,16,8 |
| Input shape (data) | (B,32,32,3) | (B,32,32,3) | (B,32,32,3) | (B,64,64,3) | (B,256,256,3) | (B,32,32,3) | (B,64,64,4) |
| Output shape (feature) | (B,8,8,512) | (B,8,8,512) | (B,8,8,512) | (B,8,8,512) | (B,8,8,512) | (B,8,8,384) | (B,8,8,512) |
| **Discriminator** | | | | | | | |
| Model | ADM | ADM | ADM | ADM | ADM | ADM | ADM |
| Architecture | U-Net encoder | U-Net encoder | U-Net encoder | U-Net encoder | U-Net encoder | U-Net encoder | U-Net encoder |
| Depth | 2 | 2 | 2 | 2 | 2 | 2 | 2 |
| Width | 128 | 128 | 128 | 128 | 128 | 128 | 128 |
| Attention Resolutions | 32,16,8 | 32,16,8 | 32,16,8 | 32,16,8 | 32,16,8 | 32,16,8 | 32,16,8 |
| Input shape (feature) | (B,8,8,512) | (B,8,8,512) | (B,8,8,512) | (B,8,8,512) | (B,8,8,512) | (B,8,8,384) | (B,8,8,512) |
| Output shape (logit) | (B,1) | (B,1) | (B,1) | (B,1) | (B,1) | (B,1) | (B,1) |
| **Discriminator Training** | | | | | | | |
| Time scheduling | VP | VP | VP | Cosine VP | VP | VP | Cosine VP |
| Time sampling | Importance | Importance | Importance | Importance | Importance | Importance | Importance |
| Time weighting | $\frac{g^2}{\sigma^2}$ | $\frac{g^2}{\sigma^2}$ | $\frac{g^2}{\sigma^2}$ | $\frac{g^2}{\sigma^2}$ | $\frac{g^2}{\sigma^2}$ | $\frac{g^2}{\sigma^2}$ | $\frac{g^2}{\sigma^2}$ |
| Batch size | 128 | 128 | 128 | 128 | 20 | 512 | 128 |
| # data samples | 50,000 | 50,000 | 50,000 | 50,000 | 30,000 | 50,000 | 5000 |
| # generated samples | 50,000 | 50,000 | 50,000 | 50,000 | 30,000 | 50,000 | 5000 |
| # Epoch | 60 | 70 | 60 | 20 | 50 | 50 | 30 |

Table 23: Training cost comparison of the score model and discriminator. CIFAR-10

| Training | Parameter Size | Training GPU | Training Time |
|---|---|---|---|
| Score (EDM) | 55.7M | 8×V100 GPUs | ∼2 days |
| Discriminator | 2.9M | 1×RTX 3090 | <2 hours |

To evaluate the robustness of our method under an imperfect discriminator, we conducted experiments using partially trained discriminators in table 25. We observed that as the discriminator training progressed, the quality of the generated samples consistently improved. Moreover, our method significantly outperformed where all proposals are accepted without a discriminator. These results indicate that even an imperfectly trained discriminator can still yield meaningful performance gains. This supports the theoretical claim in Appendix A.4, where we show that continued discriminator training leads to improvements in KL divergence.

Table 25: Ablation on discriminator epoch.

| Method | Epoch | FID↓ | NFE↓ |
|---|---|---|---|
| EDM (Heun) | - | 3.23 | 17.00 |
| + AC w.o. MH | - | 3.40 | 15.26 |
| + AC | 1 | 2.56 | 15.80 |
| | 2 | 2.59 | 15.80 |
| | 5 | 2.64 | 15.81 |
| | 10 | 2.39 | 15.81 |
| | 20 | 2.43 | 15.82 |
| | 60 | 2.38 | 15.81 |

# I   THE USE OF LARGE LANGEUAGE MODELS (LLMS)

We acknowledge the use of a Large Language Model (LLM) during the preparation of this manuscript. The LLM was employed solely as a general-purpose writing assistant to improve readability, grammar, and clarity of exposition. It was not involved in the ideation of research questions, the design of experiments, the development of methods, or the interpretation of results. The scientific contributions of this work, including problem formulation, methodology, theoretical analysis, and empirical evaluation, were conceived and carried out entirely by the authors. The role of the LLM was limited to helping refine the presentation of the text, and it did not contribute substantively to the research process itself.

# J   GENERATED IMAGES

We provide sample images at Figure 11, 12, 13, 14, 15 generated by our sampler. These are samples generated by applying our algorithm to the baseline models and samplers, used in our experiments.

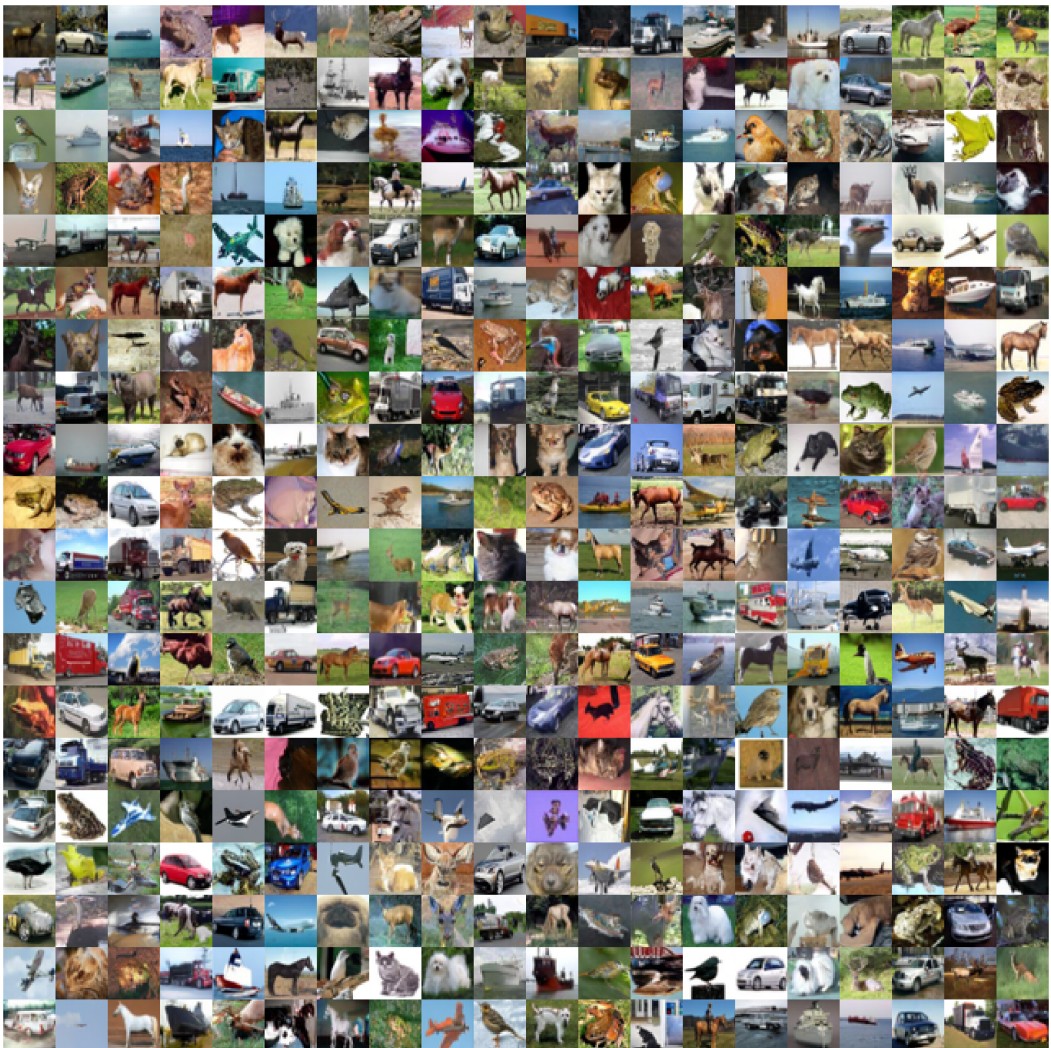

Figure 11: The uncurated generated images of AC-sampler on unconditional CIFAR-10 with EDM (EDM(Heun) sampler, NFE=26.19, FID=1.97).

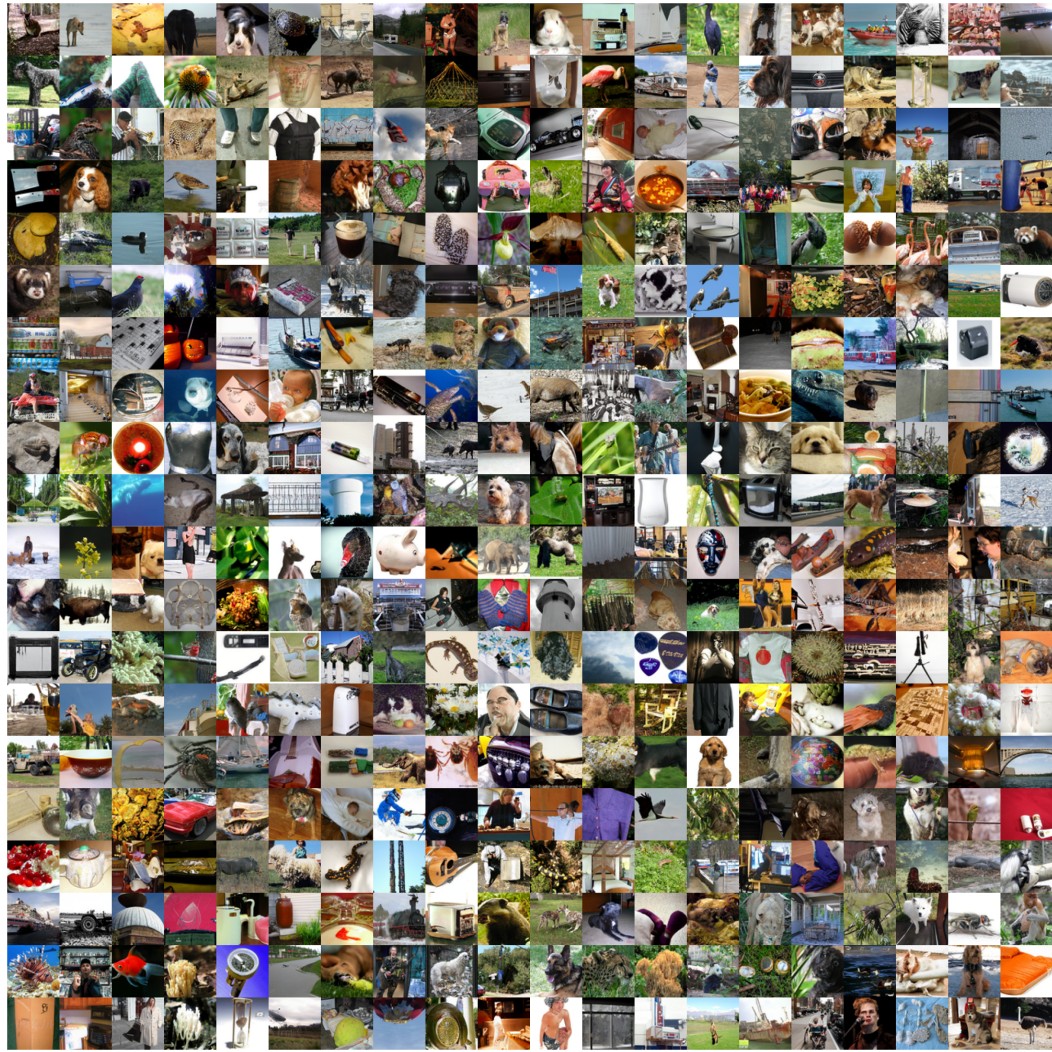

Figure 12: The uncurated generated images of AC-sampler on conditional ImageNet 64×64 with EDM (EDM(SDE) sampler, NFE=59.30, FID=2.27).

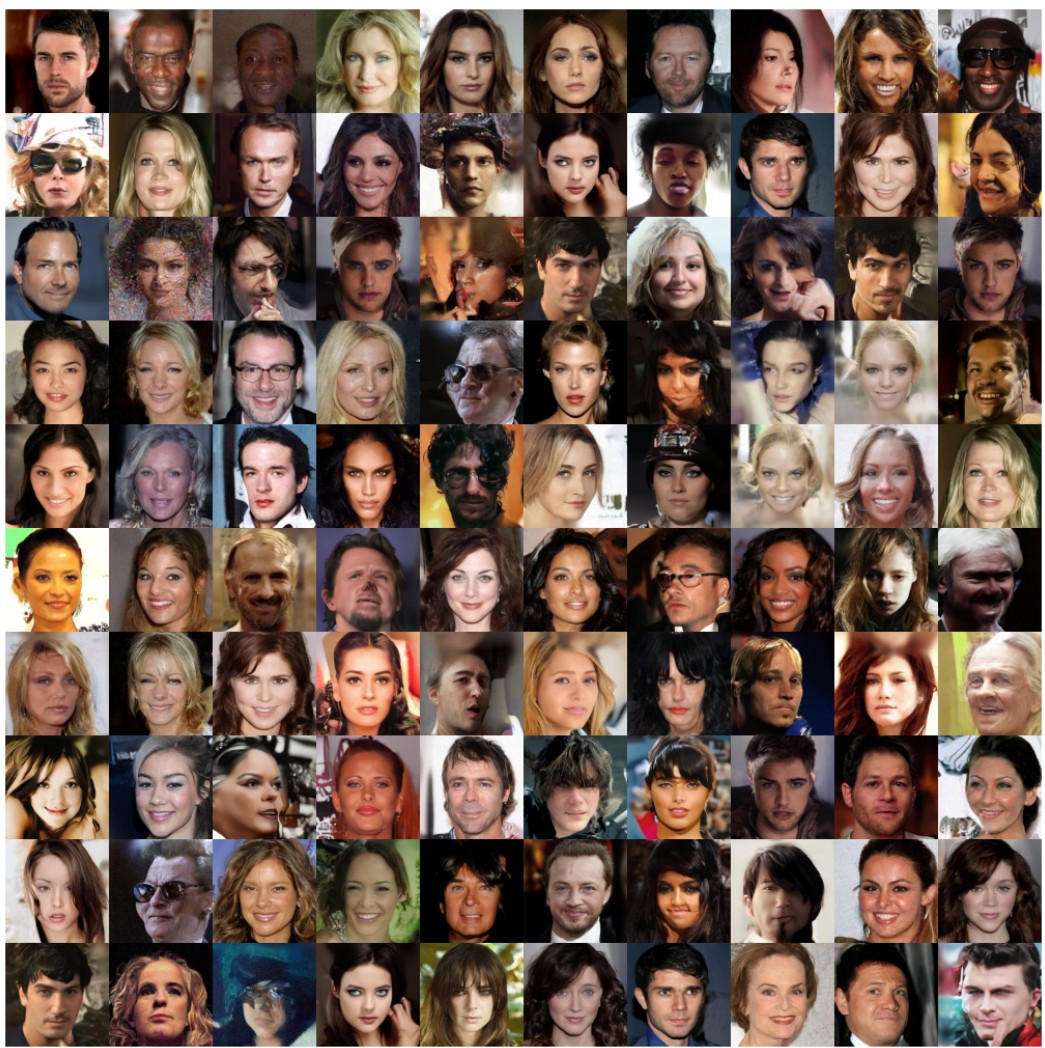

Figure 13: The uncurated generated images of AC-sampler on unconditional CelebA-HQ 256×256 with ScoreSDE (KAR1 sampler, NFE=15.94, FID=15.13).

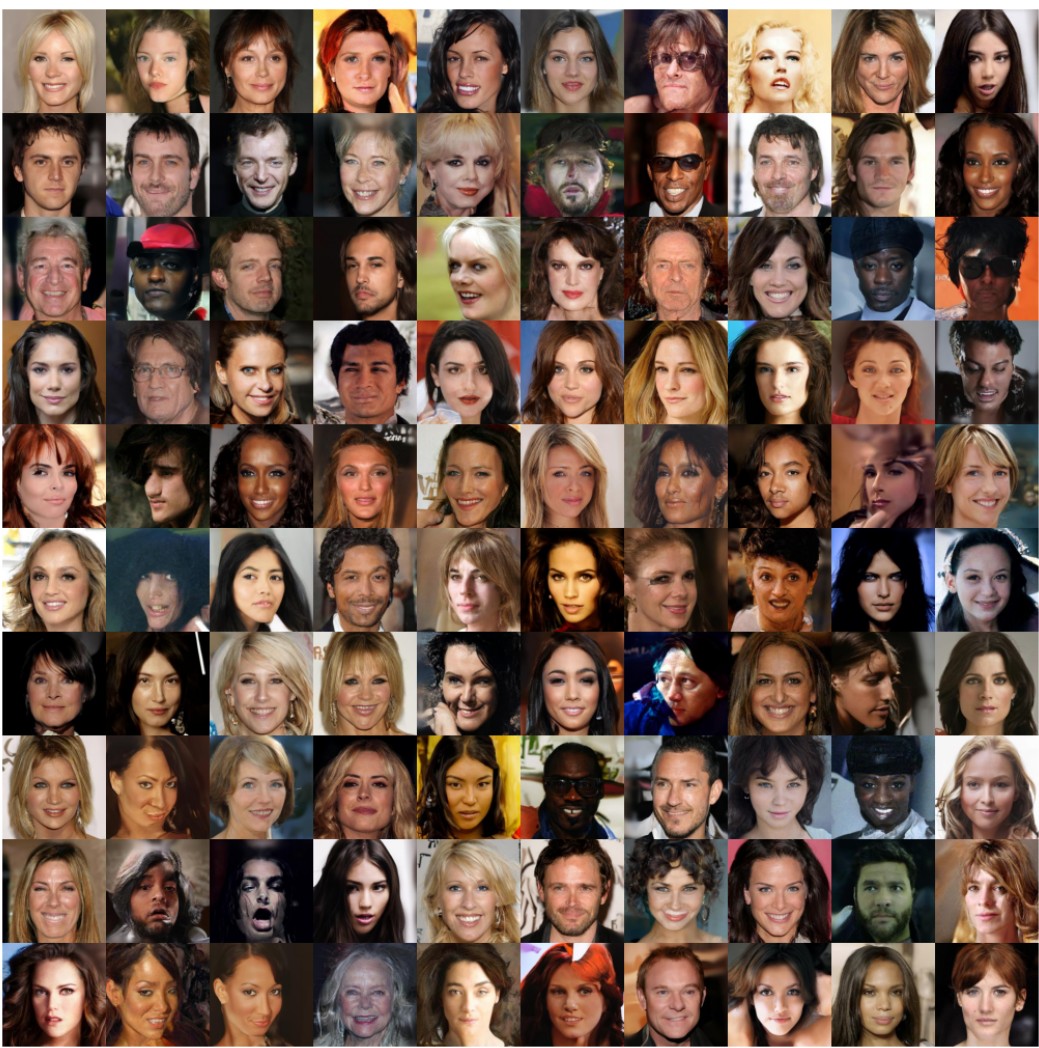

Figure 14: The uncurated generated images of AC-sampler on unconditional CelebA-HQ 256×256 with ScoreSDE(KAR2 sampler, NFE=20.05, FID=8.45).

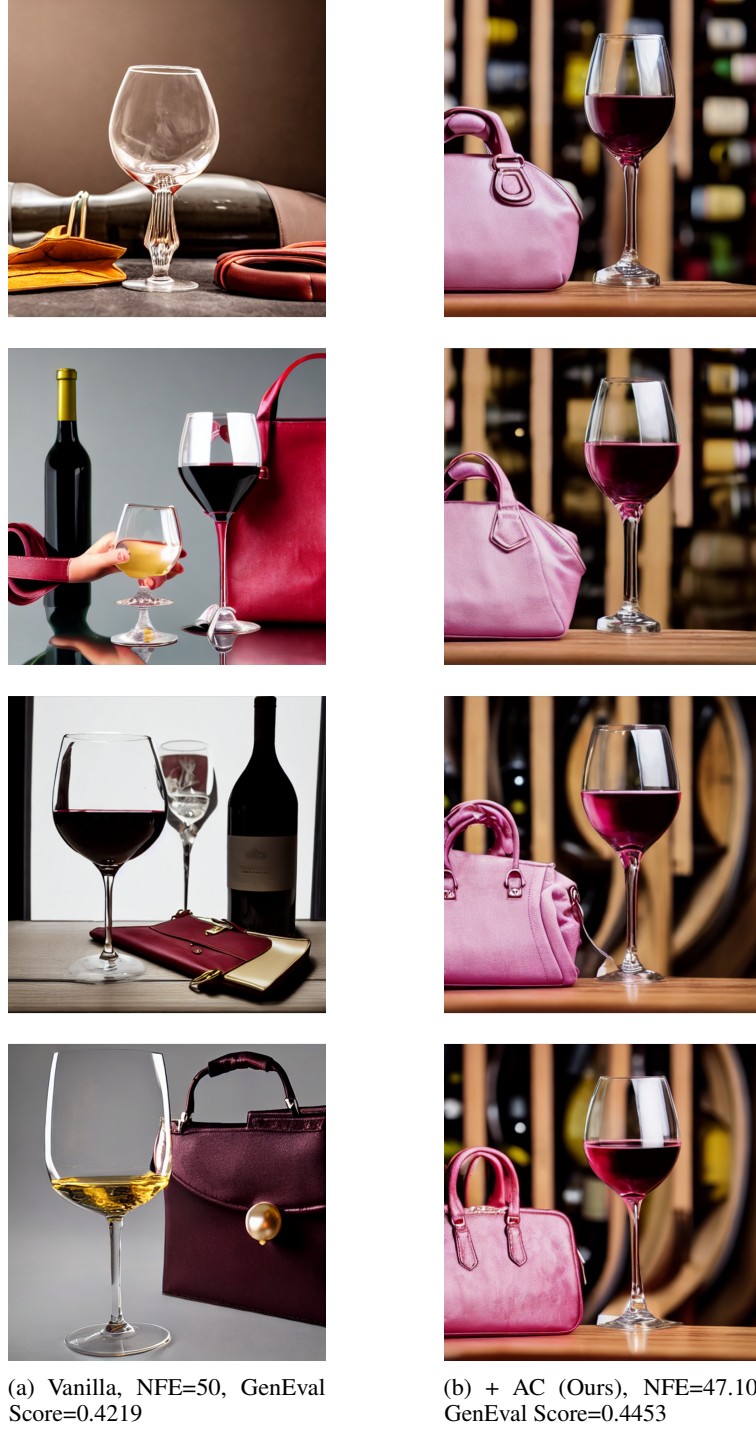

(a) Vanilla, NFE=50, GenEval Score=0.4219

(b) + AC (Ours), NFE=47.10, GenEval Score=0.4453

Figure 15: Generated images from Stable Diffusion v1.5 and AC-sampler using the prompt "a photo of a wine glass and a handbag" (DDIM sampler).

