# OpenReview forum: "AC-Sampler: Accelerate and Correct Diffusion Sampling with Metropolis-Hastings Algorithm"
_ICLR.cc/2026/Conference — ICLR 2026 Poster_

### Official Review · Reviewer_pPCw · 2025-10-31

**Soundness:** 3
**Presentation:** 3
**Contribution:** 2
**Rating:** 4
**Confidence:** 4

**Summary:**

This paper proposes a sampling algorithm called Accelerator-Corrector Sampler (AC Sampler) that both accelerates and corrects diffusion sampling. This sampler is based on Metropolis-Hastings (MH) algorithm. The acceleration is achieved via a “warm-start” by performing denoising from prior distribution to a target tilmestep $\tau$ which serves as the initial sample of MCMC. The MH acceptance probability is computed with the help of score function and also an additional time-dependent discriminator. This time-dependent discriminator is trained to predict the likelihood ratio of the unknown target data distribution and model’s marginal distribution at time $t$. The other terms in the expression for acceptance probability are Gaussian distributions and can be easily computed.

The advantages of AC-Sampler have been shown on CIFAR-10, CelebA-HQ 256 and ImageNet- 64 and 256. AC-Sampler can also be composed with other SOTA samplers such as EDM’s Heun sampler, discriminator guidance, DPM-v3 etc. and it results in improved FID score. In many cases, the average NFEs are also comparable or less, which is ideal.

**Strengths:**

1. The proposed method is orthogonal to many existing samplers and can be combined with them as indicated in Table 1 and Table 2. Further, this composition results in improved FID in general.
2.  The proposed algorithm seems to have better mode coverage than  EDM as indicated by Recall metrics. This also has been qualitatively demonstrated against DDPM solver in Figure 6.
3. The paper provides theoretical proof (Theorem 4.3 and 4.4) which shows that the generated sample distribution with AC-Sampler is closer to the true data distribution in terms of KL divergence. The paper also provide results on the expected reduction in the number of NFEs.

**Weaknesses:**

1. The method requires training an additional time-dependent discriminator however it needs to be done only one time. There will  also be additional overhead from this discriminator during sampling. It is unclear if the forward pass through the discriminator is accounted for in NFEs. The discriminator also uses a pre-trained ADM classifier as a feature extractor which might not be readily available for all datasets.
2. There is a potential mismatch between theory and practice. The practical implementation of MH in Algorithm 1 employs “propose-until-accept” design. This can introduce stationary bias as mentioned in Appendix E. This should be highlighted in the main paper. This also means that in this case, the chain wouldn’t converge to the desired target distribution $q_T$ but would rather converge to a different distribution due to the bias.
3. Appendix B reports poor performance on CelebA-HQ 256x256 where the method in the main paper doesn’t scale to high dimensional data. Appendix B proposes to do MH algorithm in the joint space of time and data. This suggests that the primary algorithm from the main paper is not robust. It is also unclear if this method can be applied to Text-to-image models.
4. Additional overhead in terms of wall clock time over many samplers such as DDIM, DPM-v3 etc. is unclear from the paper. There is only a  comparison against EDM’s Heun sampler in the paper.  In some cases, the NFE reduction is not very significant and therefore confidence intervals or the standard deviation needs to be reported.
5. The performance of the method is quite sensitive to the choice of $\tau$ as indicated in Table 9. In addition, there are many hyper parameters to tune such as burn-in length, number of steps to skip, number of parallel chains etc.

**Questions:**

1. This work  is probably relevant and could be included in related works:
Score-Based Metropolis-Hastings Algorithms, Ahmed Aloui, Ali Hasan, Juncheng Dong, Zihao Wu, Vahid Tarokh, 2024
3. What is the typical length of MCMC chain i.e. how main times do we need to repeat Algorithm 1 before generating the final sample?

---

> ### Author Response · Authors · 2025-11-21
>
> We appreciate the reviewer’s insightful comments. Our response is as follows.
>
> ---
>
> ### Weakness 1
>
> **The method requires training an additional time-dependent discriminator however it needs to be done only one time.
> There will also be additional overhead from this discriminator during sampling. It is unclear if the forward pass through the discriminator is accounted for in NFEs.
> The discriminator also uses a pre-trained ADM classifier as a feature extractor which might not be readily available for all datasets.**
>
>
> Thank you for the thoughtful comments. To compute the Metropolis–Hastings acceptance probability in a tractable manner, our method requires training a time-dependent discriminator. This training is performed only once, and the resulting discriminator can be reused across all diffusion timesteps.
>
> We acknowledge that the discriminator introduces additional overhead during sampling. We explicitly analyze this in Section 5.3 (“Faster Sampling”), and Table 5 reports a direct comparison of wall-clock time. On CIFAR-10, given the score network (55.7M parameters) and the discriminator (feature extractor 47.7M + discriminator head 2.9M), one discriminator forward pass corresponds to roughly 0.91 additional NFE in computational cost. **Despite this overhead, our method still can achieve faster sampling in terms of wall-clock time.**
>
> All NFEs reported in our experiments refer solely to the score network. For experiments in Table 2 (upper), which all rely on a discriminator, we additionally provide the discriminator NFEs in Table 2 (Top).[1,2] Note that DG requires gradient computation through its discriminator (shown as “grad”). Our results demonstrate that the method remains effective even under reduced score NFEs and discriminator NFEs.
>
> | Method     | FID  | NFE   | Disc.NFE | FID  | NFE   | Disc.NFE | FID  | NFE   | Disc.NFE |
> |------------|------|--------|----------|------|--------|-----------|------|--------|-----------|
> | EDM        | 2.05 | 27     | 0        | 2.23 | 23     | 0         | 3.23 | 17     | 0         |
> | +DiffRS    | 2.17 | 28.15  | 18.78    | 3.26 | 23.13  | 15.85     | 7.79 | 19.87  | 14.67     |
> | +DG        | **1.93** | 27     | 14 (grad)| 2.12 | 23     | 12 (grad) | 3.62 | 17     | 9 (grad)  |
> | +AC (ours) | 1.97 | **26.19**  | **2.14**     | **2.10** |**22.78**  | **2.53**      | **2.38** | **15.81**  | **1.57**      |
>
> We further conducted experiments where the classifier weights were randomly initialized and only the discriminator was trained **(No classification ability at all, random parameter)**. While performance improves when using the pre-trained ADM classifier, our method **does not fundamentally depend on the availability of either classifier or feature extractor.** Regarding the use of the ADM classifier as a feature extractor, we follow the standard practice established in prior time-dependent discriminator works such as DG and DiffRS. [1,2]
> | Method                                | FID  | NFE   |
> |--|-|-|
> | Base                                  | 2.01 | 35     |
> | AC (With pre-trained ADM classifier)  | 1.97 | 26.19  |
> | AC (Without pre-trained ADM classifier)| 1.99 | 28.19  |
>
> We also conduct experiments on the T2I model under the same randomly initialized setting. The analysis is provided in weakness 3 below.
>
> ---
>
> [1] [ICML 2023] Refining Generative Process with Discriminator Guidance in Score-based Diffusion Models
> [2] [ICML 2024] Diffusion Rejection Sampling

---

> ### Author Response · Authors · 2025-11-21
>
> ### Weakness 2
>
> **There is a potential mismatch between theory and practice. The practical implementation of MH in Algorithm 1 employs “propose-until-accept” design. This can introduce stationary bias as mentioned in Appendix E. This should be highlighted in the main paper. This also means that in this case, the chain wouldn’t converge to the desired target distribution but would rather converge to a different distribution due to the bias.**
>
> We appreciate the reviewer for raising this important point. As discussed in Appendix E, the “propose-until-accept” implementation indeed introduces a known acceptance-delay bias, which is a practical compromise in Metropolis–Hastings variants. We explicitly analyzed this bias rather than treating it as an implementation detail, and **our experiments show that its effect is empirically negligible:** the sampler consistently improves FID and NFE across all settings, indicating that the bias does not interfere with the practical benefits of MH correction. We agree that this nuance should be highlighted more clearly in the main paper, and we will revise the manuscript accordingly. Importantly, the bias is bounded, well-understood, and does not prevent the method from providing substantial empirical gains, and we clearly and explicitly mention such bias in our submission for academic contributions.
>
> ---
>
> ### Weakness 3
>
> **Appendix B reports poor performance on CelebA-HQ 256x256 where the method in the main paper doesn’t scale to high dimensional data. Appendix B proposes to do MH algorithm in the joint space of time and data. This suggests that the primary algorithm from the main paper is not robust. It is also unclear if this method can be applied to Text-to-image models.**
>
> First, we clarify that the baseline ScoreSDE [3] model exhibits relatively weak performance on CelebA-HQ 256×256: its FID/NFE is **7.23 / 2000** under the PC sampler with 1000 timesteps. The experiments in Table 8 specifically evaluate the limited-NFE regime, and even under this constraint, our method consistently achieves both FID and NFE improvements. For example, on KAR2, the base sampler yields an FID/NFE of **29.74 / 198**, whereas our AC method achieves **9.45 / 94.34**.
>
> We also conducted experiments using the original sampling procedure recommended by the ScoreSDE paper (PC sampler with 1000 timesteps) and reported results at 1K NFE. Thus, the poor performance observed on CelebA-HQ originates from the base sampler itself, not from our AC method.
> | **CelebA-HQ (1K FID)** | **FID** | **NFE** |
> |------------------------|---------|---------|
> | PC-sampler             | 18.04   | 2000    |
> | **+AC**                | **17.47** | **1719.74** |
>
> Regarding the joint time, data Metropolis-Hastings formulation, our AC sampler extends the idea of DLG (Gibbs sampling in the time-data joint distribution). This demonstrates the flexibility of our framework rather than a failure of the main algorithm.
>
> **AC is applicable on T2I setting**
>  We ran additional experiments on the text-to-image model Stable Diffusion v1.5. To evaluate realistic usage scenarios, we conducted text-to-image generation experiments using 5,000 COCO validation prompts. In T2I systems, multiple images are routinely generated per prompt [4] (e.g., DALL·E-2 [5], Midjourney [6], Bing Image Creator [7]), and under this setting, our method yields both faster sampling and improved FID then base sampler. For evaluation, we randomly selected one of the four generated images per prompt.
>
> | SD-v1.5 / COCO-Val    | $n_{\text{chain}}$ | FID   | Mean NFE      | Total sample|
> |----------------|---------|--------|-----------|------|
> | DDIM  | -       | 24.34  | 50        | 20000 |
> | +AC   | 4       | **22.94**  | **43.31**   | 20000|
>
> We additionally conducted an experiment on Geneval [4] benchmark. Geneval evaluates how well the generated images follow the prompt instructions. Since Geneval requires four different images per prompt for evaluation, we set $n_{\text{chain}} = 4$ to generate four samples per prompt and computed the Geneval scores accordingly.
>
> |Geneval score, SD-v1.5| DDIM |  + AC|
> |---|---:|---:|
> |Position|4.00|**6.25**|
> |Counting|**35.62**|**35.62**|
> |Two object|34.85|**38.13**|
> |Colors|76.33|**81.38**|
> |Single object|96.56|**97.81**|
> |Color attribute|5.75|**8.00**|
> |-|-|-|
> |Overall score|0.4219|**0.4453**|
> |NFE|50|**47.10**|
>
> ---
>
> [3] [ICLR 2021] Score-Based Generative Modeling through Stochastic Differential Equations
> [4] [NeurIPS 2023] GENEVAL: An Object-Focused Framework for Evaluating Text-to-Image Alignment
> [5] https://openai.com/dall-e-2
> [6] https://www.midjourney.com/
> [7] https://www.bing.com/images/create

---

> ### Author Response · Authors · 2025-11-21
>
> ### Weakness 4
>
> **Additional overhead in terms of wall clock time over many samplers such as DDIM, DPM-v3 etc. is unclear from the paper. There is only a comparison against EDM’s Heun sampler in the paper.**
>
> Our method is designed to be flexible and can be applied to a wide range of samplers beyond EDM. The wall-clock comparison in Table 5 evaluates Heun v.s. Heun + AC, as Heun is the default sampler used in EDM.
>
> To address the reviewer’s concern, we additionally applied our AC sampler to **DPM-v3 and DDIM and directly measured the wall-clock time reduction.** As discussed in Weakness 1, the discriminator introduces a non-negligible computational cost. With further parameter tuning on DPM-v3, we were able to obtain improvements in all three metrics—NFE, time, and FID. Here, “time” denotes the wall-clock time required to generate 100 samples. These results demonstrate that our method provides practical speed-ups across different samplers, not only Heun. We did our experiment 5 times on DPM-v3 setting and report mean and standard deviation.
>
> | EDM/CIFAR10 | FID        | NFE        | Time (seconds) |
> |--|---|--|--|
> | DPM-v3      | 8.73       | 6          | 2.42           |
> | + AC        | **7.61±0.081** | **4.58±0.012** | **2.15±0.023**    |
>
> | SD-v1.5 / COCO-Val    | N_chain | FID   | Mean NFE      | Total sample| Time (minute)|
> |----------------|---------|--------|-----------|------|-----|
> | DDIM  | -       | 24.34  | 50        | 20000 | 10.83|
> | +AC   | 4       | **22.94**  | **43.31**   | 20000| **10.21**|
>
>
> ---
>
>
> ### Weakness 5
>
> **In some cases, the NFE reduction is not very significant and therefore confidence intervals or the standard deviation needs to be reported.**
>
> For experiments reported in Tables 1–3 (CIFAR-10, ImageNet-64, and CelebA-HQ), we measured FID and NFE five times independently. We report the mean and standard deviation. In addition, we conducted statistical significance tests using p-values to assess whether the differences are meaningful.
>
> Across all datasets, our method **consistently shows a statistically significant reduction in NFE.** Moreover, under comparable NFE settings, we also observe meaningful and consistent improvements in FID.
> | Method        | FID         | NFE          | p-value(FID) | p-value(NFE) |
> |---------------|-------------|--------------|--------------|--------------|
> | EDM(Base)     | 2.06±0.01   | 27±0         | -            | -            |
> | +AC           | 2.02±0.06   | 26.18±0.01   | 0.21         | **0.00**     |
> | EDM(Base)     | 2.22±0.02   | 23±0         | -            | -            |
> | +AC           | **2.11±0.07** | **22.79±0.02** | **0.02**     | **0.00**     |
> | EDM(Base)     | 3.30±0.03   | 17±0         | -            | -            |
> | +AC           | **2.47±0.04** | **15.79±0.01** | **0.00**     | **0.00**     |
>
> | Method | FID          | NFE            | p-value(FID) | p-value(NFE) |
> |---|-----|--|--|--|
> | EDM    | 2.32±0.05    | 61±0           | -            | -            |
> | +AC    | 2.29±0.03    | 58.75±0.01     | 0.29         | **0.00**     |
> | EDM    | 1.78±0.03    | 127±0          | -            | -            |
> | +AC    | 1.81±0.03    | 121.98±0.01    | 0.32         | **0.00**     |
> | EDM    | 1.40±0.02    | 511±0          | -            | -            |
> | +AC    | 1.45±0.02    | 483.86±0.00    | 0.00     | **0.00**     |
>
> | Method                 | FID            | NFE             | p-value(FID) | p-value(NFE) |
> |--|--|---|--|--|
> | ScoreSDE_KAR2(Base)    | 57.52±1.08     | 98±0             | -            | -            |
> | +AC                    | **7.02±0.25**  | 97.54±0.41       | **0.00**     | 0.07         |
> | ScoreSDE_KAR1(Base)    | 125.13±1.38    | 98±0             | -            | -            |
> | +AC                    | **15.45±0.83** | **87.24±0.01**   | **0.00**     | **0.00**     |

---

> ### Author Response · Authors · 2025-11-21
>
> ### Weakness 6
> **The performance of the method is quite sensitive to the choice of  τ as indicated in Table 9. In addition, there are many hyper parameters to tune such as burn-in length, number of steps to skip, number of parallel chains etc.**
>
> We appreciate the reviewer’s accurate observation. Our method indeed requires tuning several hyperparameters. To support this process, Appendix C provides a detailed analysis of their trade-offs as well as heuristic guidelines.
>
> As discussed in the paper, $\tau$ and SNR are the key hyperparameters; they serve as the primary levers that control the trade-off between sampling speed and sample diversity. Importantly, $\tau$ is a parameter that naturally arises from the diffusion sampling process itself [8,9]. We began our selection of the SNR following the methodology adopted in the prior works [3,10].
>
> In contrast, the remaining hyperparameters such as burn-in length, number of skipped steps [11], or chain length are not key parameters. As noted in Appendix C, these arise from practical design choices needed to construct a Metropolis–Hastings chain and must be set empirically. Nonetheless, Tables 10–12 show that our method remains robust across a wide range of these auxiliary hyperparameters.
>
> ---
>
> ### Question 1
>
> **This work is probably relevant and could be included in related works: Score-Based Metropolis-Hastings Algorithms, Ahmed Aloui, Ali Hasan, Juncheng Dong, Zihao Wu, Vahid Tarokh, 2024**
>
> Although both our work and Aloui et al. (2024) apply the Metropolis–Hastings algorithm to diffusion sampling, Aloui et al. has two fundamental differences:
> 1. No true distribution correction - their acceptance ratio uses an approximate score of the target distribution with score network, making the model distribution its own target.
> 2. Proposal-dependent training - because their loss explicitly depends on the proposal distribution, any change in proposal distribution requires retraining the acceptance function.
>
> ---
>
> ### Question 2
>
> **What is the typical length of MCMC chain i.e. how main times do we need to repeat Algorithm 1 before generating the final sample?**
>
> For unconditional models, we did not impose a strict constraint on the MCMC chain length. In our FID evaluation, where 50,000 samples are generated with a batch size of 100, we allowed a maximum chain length of 500.
> For conditional models, however, using a long chain can introduce class bias when the number of generated samples per class is limited. Therefore, we set the chain length to 2 in this setting.
> For text-to-image generation, we used a chain length of 4.
>
> ---
>
> [8] [ICLR 2022] SDEdit: Guided Image Synthesis and Editing with Stochastic Differential Equations
> [9] [ICLR 2023] Diffusion Models Already Have a Semantic Latent Space
> [10] [NeurIPS 2019] Generative Modeling by Estimating Gradients of the Data Distribution
> [11] [ICML 2023] Denoising MCMC for accelerating diffusion-based generative models

---

### Official Review · Reviewer_jfbK · 2025-11-01

**Soundness:** 4
**Presentation:** 3
**Contribution:** 3
**Rating:** 6
**Confidence:** 4

**Summary:**

The paper proposes AC-Sampler, a “accelerator-corrector” for diffusion models that jumps to an intermediate timestep (instead of starting at pure noise) and then applies Metropolis–Hastings (MH) with a MALA proposal built from the pretrained score network. This both shortens the reverse trajectory (speedup) and, via MH acceptance, corrects samples so their marginal at targets the true distribution. A time-dependent discriminator provides an estimate of the density ratio so the MH acceptance probability is tractable.

**Strengths:**

1. Clever decomposition of the acceptance ratio and use of a time-dependent discriminator make the MH step closed-form and cheap to evaluate。

2. Theoretical guarantees: Expected NFE reduction when the acceptance rate exceeds a mild threshold; KL to the data distribution does not worsen and improves with more MALA steps (under stated integrability conditions).

3. Designed to sit atop existing accelerators/correctors (e.g., DPM-v3, DG), often improving their FID at fewer steps.

**Weaknesses:**

1. The proposed method does not appear to provide acceleration when the batch size is 1; in this regime, the acceleration gain vanishes.

2. Hyperparameter sensitivity & chain design: Performance depends on the choice of target timestep 𝜏 and the proposal step size/SNR, as well as burn-in/chain length—these trade speed for acceptance/mixing.

**Questions:**

1. Is the proposed method compatible with classifier-free guidance (CFG)? While CFG has known theoretical issues, modern large-scale systems rely on it heavily.

2. Could the authors add a discussion in related work comparing their approach with SDE-based sampling methods [1–3] (which can be viewed as Langevin-type correctors) and with work that addresses training–inference mismatch [4,5]?


[1] Gotta go fast when generating data with score-based models.

[2] SA-solver: Stochastic adams solver for fast sampling of diffusion models.

[3] Seeds: Exponential sde solvers for fast high-quality sampling from diffusion models.

[4] Input perturbation reduces exposure bias in diffusion models.

[5] Improved Diffusion-based Generative Model with Better Adversarial Robustness.

---

> ### Author Response · Authors · 2025-11-21
>
> Thank you for your thoughtful feedback. We provide our clarification below.
>
> ---
>
> ### Weakness 1
> **The proposed method does not appear to provide acceleration when the batch size is 1; in this regime, the acceleration gain vanishes.**
>
> You are correct that our method does not yield acceleration when the batch size is 1.
> As we mention in the Appendix F, the case of $ n_{\text{chain}} = 1$ is retained as a “correction-only” regime. Since our discriminator is trained for all diffusion timesteps, we can still apply Metropolis–Hastings correction to each individual sample during sampling. We provide an analysis in Table below (a part of Table 15 in appendix). This demonstrates the effectiveness of applying our correction at various timesteps.
>
> | Sampling  | FID↓ | NFE↓   | SNR  | $n_{\text{burn-in}}$ | $n_{\text{chain}}$ |
> | -- | -- | - | -- | -- | -- |
> | EDM    | 1.97 | 35     | -    | -        | -      |
> | +PC | 2.00 | 51    |  0.16 | 3        | 1      |
> | +AC | **1.93** | 54.91 | 0.16 | 3        | 1      |
>
> While we agree that there is no acceleration when the batch size is 1, **$n_{\text{chain}}$ determines the degree of acceleration.** Importantly, real-world generative services [7–9] typically return several alternative generation results for the same query to let users choose the best from those candidates. This number of alternative generations can be seen as a  $n_{\text{chain}}$  and therefore enables the acceleration effect of our method in practical deployment scenarios.
>
> ---
>
> ### Weakness 2
> **Hyperparameter sensitivity & chain design: Performance depends on the choice of target timestep 𝜏 and the proposal step size/SNR, as well as burn-in/chain length—these trade speed for acceptance/mixing.**
>
> We appreciate the reviewer’s accurate observation. Our method indeed requires tuning several hyperparameters. To support this process, Appendix C provides a detailed analysis of their trade-offs as well as heuristic guidelines.
>
> As discussed in the paper, $\tau$ and SNR are the key hyperparameters; they serve as the primary levers that control the trade-off between sampling speed and sample diversity. Importantly, **$\tau$** is a parameter that **naturally arises from the diffusion sampling process itself [1,2]**. We began our selection of the **SNR** following the methodology **adopted in the prior works [3,4].**
>
> In contrast, the remaining hyperparameters such as burn-in length, number of skipped steps [5], or chain length are not key parameters. As noted in Appendix C, these arise from practical design choices that are needed to construct a Metropolis–Hastings chain and must be set empirically. Nonetheless, Tables 10–12 show that **our method remains robust across a wide range of these auxiliary hyperparameters.**

---

> ### Author Response · Authors · 2025-11-21
>
> ### Question 1
> **Is the proposed method compatible with classifier-free guidance (CFG)? While CFG has known theoretical issues, modern large-scale systems rely on it heavily.**
>
> Yes. **AC-Sampler is fully compatible with classifier-free guidance (CFG).**
> During discriminator training and in computing Eq. (11), we simply use the CFG-guided score, and no modification of AC-Sampler is required.
> We already demonstrated this in Section 5.2 (ImageNet 256), where the base sampler uses **CFG scale 1.5**, and AC-Sampler consistently improves both FID and NFE under the same CFG setting.
> Additionally, we conducted experiments on Stable Diffusion v1.5 with **CFG scale 7.5.** To evaluate realistic usage scenarios, we conducted text-to-image generation experiments using 5,000 COCO validation prompts. In T2I systems, multiple images are routinely generated per prompt [6] (e.g., DALL·E-2 [7], Midjourney [8], Bing Image Creator [9]), and under this setting, our method yields both faster sampling and improved FID. For evaluation, we randomly selected one of the four generated images per prompt.
> | SD-v1.5 / COCO-Val    | $n_{\text{chain}}$ | FID   | Mean NFE      | Total sample|
> |-|-|-|-|-|
> | DDIM  | - | 24.34  | 50        | 20000 |
> | +AC   | 4       | **22.94**  | **43.31**   | 20000|
>
> We additionally conducted an experiment on Geneval [6] benchmark. Geneval evaluates how well the generated images follow the prompt instructions. Since Geneval requires four different images per prompt for evaluation, we set $n_{\text{chain}}=4$ to generate four samples per prompt and computed the Geneval scores accordingly.
>
> |Geneval score, SD-v1.5| DDIM|+AC|
> |--|--:|--:|
> |Position|4.00|**6.25**|
> |Counting|**35.62**|**35.62**|
> |Two object|34.85|**38.13**|
> |Colors|76.33|**81.38**|
> |Single object|96.56|**97.81**|
> |Color attribution|5.75|**8.00**|
> |---|--|-|
> |Overall score|0.4219|**0.4453**|
> |NFE|50|**47.10**|
>
> AC-Sampler consistently outperformed the base sampler across all Geneval metrics and achieved better FID with reduced NFE. These results further confirm that AC-Sampler remains robust and effective under CFG-based generation.
>
> ---
>
> ### Question 2
>
> **Could the authors add a discussion in related work comparing their approach with SDE-based sampling methods [1–3] (which can be viewed as Langevin-type correctors) and with work that addresses training–inference mismatch [4,5]?**
>
> We thank the reviewer for the suggestion.
>
> 1. SDE-based sampling and solver-related corrections.
> Several SDE-based methods mitigate numerical errors by improving the accuracy of the sampler itself: adaptive adjustment of SDE solver hyperparameters to reduce discretization error [10], controlled-variance reverse SDEs that share the same marginals for more efficient sampling [11], and higher-order SDE solvers to further reduce solver error. [12]
> These works focus on improving the numerical integration quality of the diffusion trajectory.
>
> 2. Approaches addressing training–inference mismatch.
> Other works instead target the mismatch between the training distribution and the inputs encountered during inference. Examples include: training diffusion models with input perturbations under an assumed Gaussian mismatch [13], and adversarially optimizing perturbations and model parameters to achieve distributionally robust diffusion training. [14]
> These approaches retrain the model to better handle off-manifold inputs during sampling.
>
> Our method is applied on top of a base sampler, and therefore remains orthogonal to SDE-based solver improvements while preserving their advantages. Moreover, because our approach corrects generated samples solely through discriminator training, it mitigates the training–inference mismatch without requiring any additional training of the underlying diffusion model.
> We will add this discussion to the related work section in the revised version.
>
> ---
>
> [1] [ICLR 2022] SDEdit: Guided Image Synthesis and Editing with Stochastic Differential Equations
> [2] [ICLR 2023] Diffusion Models Already Have a Semantic Latent Space
> [3] [NeurIPS 2019] Generative Modeling by Estimating Gradients of the Data Distribution
> [4] [ICLR 2021] Score-Based Generative Modeling through Stochastic Differential Equations
> [5] [ICML 2023] Denoising MCMC for accelerating diffusion-based generative models
> [6] [NeurIPS 2023] GENEVAL: An Object-Focused Framework for Evaluating Text-to-Image Alignment
> [7] https://openai.com/dall-e-2
> [8] https://www.midjourney.com/
> [9] https://www.bing.com/images/create
> [10] [NeurIPS 2021 Workshop DLDE] Gotta go fast when generating data with score-based models
> [11] [NeurIPS 2023] SA-Solver: Stochastic Adams Solver for Fast Sampling of Diffusion Models
> [12] [NeurIPS 2023] SEEDS: Exponential SDE Solvers for Fast High-Quality Sampling from Diffusion Models
> [13] [ICML 2023] Input perturbation reduces exposure bias in diffusion models
> [14] [ICLR 2025] Improved Diffusion-based Generative Model with Better Adversarial Robustness

---

### Official Review · Reviewer_MQvF · 2025-11-12

**Soundness:** 3
**Presentation:** 2
**Contribution:** 2
**Rating:** 4
**Confidence:** 3

**Summary:**

Dear authors, I am the AC. Since two reviewers ghosted the paper or wrote last-minute that they wouldn't be able to submit a review, and since I was unable to find emergency reviewers, I have now written an emergency review of the paper.

The paper proposes a Metropolis-Hastings correction at an intermediate diffusion step \tau to accelerate the sampling process of diffusion models. The idea of integrating MCMC updates into the reverse process is conceptually interesting, and the theoretical analysis is clearly written. The experimental results show modest improvements in FID with fewer or comparable NFEs.

**Strengths:**

Bringing an explicit Metropolis-Hastings correction into the diffusion sampling loop is an interesting idea to integrate score-based generative modeling and classical MCMC. Prior papers have explored MH with diffusion in other contexts, such as MCMC correction for model composition and Metropolis sampling for constrained diffusion, but using an MH step specifically as a generic accelerator within the standard image-synthesis pipeline is still relatively underexplored.

The paper not only proposes a new method but also analyzes the expected NFE under acceptance/rejection dynamics (e.g., conditions under which truncating at \tau, followed by a short MH chain, reduces per-sample NFEs).

The empirical results, while limited in their scope (see below), are promising and show consistent improvements in both terms of FID scores and efficiency.

**Weaknesses:**

While the MH-corrected acceleration mechanism is interesting, the claimed efficiency improvement is not convincing. The method truncates the reverse diffusion at an intermediate noise level \tau runs a short chain there, and then further denoises each accepted sample from  \tau -> 0. Thus, every accepted sample still requires a full denoising segment, meaning there is no intrinsic saving *per sample* unless multiple final samples share the same denoising sequence from T to \tau. Again, if the goal is to generate one image at inference time (in the realistic setting), there is no saving, as far as I understand.

In the "amortized" setting, where one truncated trajectory is used for several final samples, the expected NFE per sample can indeed drop, as shown in Proposition 4.2. However, if only one sample is drawn per trajectory, the method would be slower, not faster, due to the added proposal evaluations. The paper would benefit from explicitly stating this amortization assumption or correcting my understanding of the paper.

While the paper emphasizes improved efficiency and reduced NFEs, it does not adequately situate the proposed MH-based acceleration among existing approaches explicitly designed for fast diffusion sampling. In particular, recent methods such as Tong et al., “Learning to Discretize Denoising Diffusion ODEs (LD3)” (ICLR 2025) and the references mentioned herein (which also work with a trained diffusion model) should be compared to. Relating to my point above, the comparison should be done for generating one single sample for the same number of time steps (plus the overhead of your discriminator), not averaged over a generation of a batch of l images.

**Questions:**

Is my understanding correct that the improved efficiency is only observed for the (imo unrealistic assumption) that multiple images are generated for the same prior noise?

What is the exact goal of the theorems and how do they relate to practical settings? E.g., Prop 4.2: how are all the assumptions made implicitly here realistic?

Why are you not comparing to other methods for reducing the step size without having to train a new diffusion model?

Why did you not also make a comparison for generating a single (or a small number of) images?

---

> ### Author Response · Authors · 2025-11-21
>
> Thank you for raising this important point. We address the issue in detail below.
>
> ---
>
> ### Weakness 1
>
> **While the MH-corrected acceleration mechanism is interesting, the claimed efficiency improvement is not convincing. The method truncates the reverse diffusion at an intermediate noise level \tau runs a short chain there, and then further denoises each accepted sample from \tau -> 0. Thus, every accepted sample still requires a full denoising segment, meaning there is no intrinsic saving per sample unless multiple final samples share the same denoising sequence from T to \tau. Again, if the goal is to generate one image at inference time (in the realistic setting), there is no saving, as far as I understand.
> In the "amortized" setting, where one truncated trajectory is used for several final samples, the expected NFE per sample can indeed drop, as shown in Proposition 4.2. However, if only one sample is drawn per trajectory, the method would be slower, not faster, due to the added proposal evaluations. The paper would benefit from explicitly stating this amortization assumption or correcting my understanding of the paper.**
>
> ### Weakness 2
> **While the paper emphasizes improved efficiency and reduced NFEs, it does not adequately situate the proposed MH-based acceleration among existing approaches explicitly designed for fast diffusion sampling. In particular, recent methods such as Tong et al., “Learning to Discretize Denoising Diffusion ODEs (LD3)” (ICLR 2025) and the references mentioned herein (which also work with a trained diffusion model) should be compared to. Relating to my point above, the comparison should be done for generating one single sample for the same number of time steps (plus the overhead of your discriminator), not averaged over a generation of a batch of l images.**
>
> ### Question 1
> **Is my understanding correct that the improved efficiency is only observed for the (imo unrealistic assumption) that multiple images are generated for the same prior noise?**
>
> ### Question 3
> **Why are you not comparing to other methods for reducing the step size without having to train a new diffusion model?**
>
> ---
>
> First, reporting the NFE for a single final sample is inherently ambiguous in our setting, because each sample requires a different number of proposal evaluations due to the stochastic nature of MH correction. This is also the case for prior works that are most closely related to ours [1,2], which similarly report averaged NFEs rather than per-sample NFEs.
>
> There are two fundamentally different acceleration goals in diffusion sampling:
> 1. Latency – reducing the time to generate one data
> 2. Throughput – reducing the time to generate large amount of data.
>
> **Our method is explicitly designed for the second setting (Throughput)**. Improving throughput is practically important because generating large batches of synthetic data has become central to many real-world applications, including data-scarcity mitigation [3,4], anomaly generation [5], and data augmentation [6].
>
> While we branch multiple samples from a single prior noise point, the resulting samples behave as if **they originated from different priors due to the MCMC mixing**, which is the key of our contribution. This can be shown from the improved Recall scores in Table 5 and the enhanced mode coverage shown in Figure 6. These results indicate that such a sampling strategy is not only feasible but also realistic in practical applications. This is a sampling setup originally introduced in [1] and our method further extends and strengthens the idea proposed in [1] by incorporating MH-corrected branching.
>
> A major line of work on reducing latency focuses on improving ODE solvers by reducing discretization error, thereby enabling faster integration [7-10]. Such methods can also reduce throughput either. However, they exclusively **address discretization error and do not correct other sources of sampling inaccuracy, such as score-approximation error or prior-mismatch error**.
>
> In contrast, our method explicitly corrects the distributional mismatch accumulated up to $\tau$ through MH adjustment. This mismatch includes **score-approximation error, prior-mismatch error, and, for ODE-based samplers, any remaining discretization error as well.** Consequently, our approach improves aspects of the sampling process that solver refinement alone cannot address, ultimately enabling faster and more accurate sampling in the high-throughput setting. We believe the extensive experimental results in the paper clearly demonstrate this advantage. Moreover, our method integrates seamlessly with deterministic sampling procedures as well as with stochastic samplers. This further highlights the flexibility of our approach.

---

> ### Author Response · Authors · 2025-11-21
>
> Recently, some ODE solvers have begun addressing discretization challenges by leveraging not only the gradient at the current timestep, but also accumulated gradient information stored throughout the sampling trajectory.
>
> LD3 [10] is an example of a method that leverages stored gradient information across the trajectory. The methodology itself does not inherently rely on a higher-order ODE solver; however, in the original work, the authors chose a higher-order student sampler, which enabled the student to accurately track the ODE trajectory of a teacher sampler with many steps. In LD3, the student sampler reuses score evaluations from earlier steps along the same deterministic trajectory to achieve this faithful approximation.
>
> We made extensive efforts to combine LD3 with our method. However, because our approach branches multiple samples at an intermediate timestep $\tau$, the score evaluations prior to τ are no longer valid for the new trajectories. As a result, the key assumption required by higher order ODE sampler - namely, the existence of a single deterministic trajectory whose past score evaluations can be reused-does not hold in our setting. Combining the two approaches would therefore require using a lower-order ODE solver for steps after $\tau$, since no gradient information is available in this region, while relying on the MH correction to compensate for the resulting approximation error.
>
> In the NFE ≈ 10 regime, we did observe cases where such a trade-off becomes favorable, and we report the corresponding results in the table below. We encountered a similar phenomenon with DPM-Solver v3; in that case, the gain provided by the MH correction outweighed the loss in ODE accuracy, resulting in clear improvements as demonstrated by our experiments. We believe these observations highlight the potential compatibility of the methods and suggest a promising direction for future research.
>
> | CIFAR10 | FID  | NFE  |
> |---------|------|-------|
> | LD3     | 2.41 | 10    |
> | +AC     | 2.40 | 9.77  |
>
>
> ---
>
> [1] [ICML 2023] Denosing MCMC for Accelerating Diffusion based Generative models
> [2] [ICML 2024] Diffusion Rejection sampling
> [3] [NeurIPS 2023] Dataset Diffusion: Diffusion-based Synthetic Dataset Generation for Pixel-Level Semantic Segmentation
> [4] [ICRA 2024] SceneControl: Diffusion for Controllable Traffic Scene Generation
> [5] [AAAI 2024] AnomalyDiffusion: Few-Shot Anomaly Image Generation with Diffusion Model
> [6] [ICLR 2024] EFFECTIVE DATA AUGMENTATION WITH DIFFUSION MODELS
> [7] [NeurIPS 2023] UniPC: A Unified Predictor-Corrector Framework for Fast Sampling of Diffusion Models
> [8] [NeurIPS 2023] DPM-Solver-v3: Improved Diffusion ODE Solver with Empirical Model Statistics
> [9] [ICLR 2022] Pseudo Numerical Methods for Diffusion Models on Manifolds
> [10] [ICLR2025] Learning to discretize denoising diffusion ODE

---

> ### Author Response · Authors · 2025-11-21
>
> ### Question 2
>
> **What is the exact goal of the theorems and how do they relate to practical settings? E.g., Prop 4.2: how are all the assumptions made implicitly here realistic?**
>
> - **Theorem 4.1** establishes that the Metropolis-Hastings acceptance probability can be computed tractably by decomposing it into terms that can be evaluated exactly and a density-ratio term approximated by the discriminator. Importantly, this result does not rely on additional assumptions. **This theorem is needed to derive the acceptance probability calculation in the MCMC chain for branching multiple chains from a single chain.**
>
> - **Proposition 4.2** analyzes the reduction in NFE occurs when multiple final samples are branched from the same truncated trajectory. The assumptions used here are realistic and can be validated empirically. In particular, from the acceptance probabilities reported in Table 9 of Appendix C, we observe that the above assumption is easily satisfied in practice. For example, in the EDM Heun sampler, the total effective NFE is T = 35, and the MALA chain starts after approximately 10 NFE steps. Thus, the truncated point corresponds to tau = 25. Proposition 4.2 requires $\alpha > 1/(35 - 25 + 1) \approx 0.09$, and our experiments confirm that this condition is satisfied (0.4703 >> 0.09). With sufficiently long chains (e.g., $n_{\text{chain}} = 300$), the variance of NFE becomes negligible. **This proposition is needed to ensure that acceleration remains effective in real-world setting**
>
> Moreover, when the value of \(l\) is small (e.g., in the conditional generation setting, $n_{\text{chain}}$ is small), a large number of trees are created, and the **variance of reduced NFE decreases proportionally to the square of the number of trees**.  For this reason, effective acceleration is possible even in the small $n_{\text{chain}}$ region. Our extensive experimental results confirm this acceleration effect.
>
> - **Theorems 4.3 and 4.4** show that, under an optimal discriminator, the AC sampler produces samples closer to the true distribution than those from the base model. If we denote the KL-divergence improvement in Theorem 4.3 as the "gain" (see eq. 44~46 in appendix A.4), the gain is exactly the quantity $D_{KL}[q_{\tau} \|\| p_{\tau}^{\theta}]$. When the discriminator is untrained and cannot distinguish the two distributions, this gain becomes zero. As the discriminator is trained, the estimated acceptance ratio progressively approaches its optimal form, and the gain moves closer to the optimal value. This can be empirically verified from the results presented in Table 19.
>
> **The additional assumption in Theorem 4.4** concerns the function space and is used only to justify exchanging the order of integration in Eq. (54). While verifying such assumptions directly is difficult, similar conditions for applying Fubini's theorem are common in previous theoretical work [11,12] and serve as standard technical assumptions. **These two theorems are needed to theoretically ensure that the MCMC chain for branching enables sampling closer to the data distribution.**
>
> ---
>
> [11] [NeurIPS 2021] Diffusion Schrödinger Bridge with Applications to Score-Based Generative Modeling
> [12] [ICLR 2023] Flow Matching for Generative Modeling

---

> ### Author Response · Authors · 2025-11-21
>
> ### Question 4
> **Why did you not also make a comparison for generating a single (or a small number of) images?**
>
> **Our method is not intended to accelerate the generation of a single image. (Only correction occur when generating single image)** As shown in Table 15, when only one sample is generated from a truncated trajectory, the MH correction provides distributional improvement but does not yield acceleration.
> We conducted additional experiments on CIFAR-10 to examine the behavior in the small- $n_{\text{chain}}$ regime, i.e., when only a small number of generations are produced. Importantly, $n_{\text{chain}}$ acts as a controllable lever that adjusts the trade-off of acceleration: larger values naturally lead to significant amortized acceleration, which is the intended operating regime of our method.
>
> | Method       | $n_{\text{chain}}$ | FID  | NFE    | Total sample|
> |--------------|---------|------|--------|-------|
> | EDM (Heun)   | -       | 2.01 | 35     | 50000|
> | +AC           | 1       | 1.99 | 37.16  | 50000|
> | +AC           | 2       | 1.99 | 35.87  | 50000|
> | +AC           | 3       | 2.01 | 34.04  | 50000|
> | +AC           | 5       | 1.98 | 32.47  | 50000|
>
> To evaluate realistic usage scenarios, we conducted text-to-image generation experiments on Stable Diffusion v1.5 using 5,000 COCO validation prompts. In T2I systems, multiple images are routinely generated per prompt [13] (e.g., DALL·E-2 [14], Midjourney [15], Bing Image Creator [16]), and under this setting, our method yields both faster sampling and improved FID.
> For evaluation, we randomly selected one of the four generated images per prompt.
>
> | SD-v1.5 / COCO-Val    | $n_{\text{chain}}$ | FID   | Mean NFE      | Total sample|
> |----------------|---------|--------|-----------|------|
> | DDIM  | -       | 24.34  | 50        | 20000 |
> | +AC   | 4       | **22.94**  | **43.31**   | 20000|
>
> We also examined the small-sample regime in the conditional setting with $n_{\text{chain}} = 2$. Table 4 shows that even at this scale, AC achieves clear acceleration gains, supporting that our method remains effective across a wide range of practical regimes.
>
>
> ---
>
> [13] [NeurIPS 2023] GENEVAL: An Object-Focused Framework for Evaluating Text-to-Image Alignment
> [14] https://openai.com/dall-e-2
> [15] https://www.midjourney.com/
> [16] https://www.bing.com/images/create

---

> > ### Comment · Reviewer_MQvF · 2025-11-25
> >
> > Thank you for your answer. Could you point me to a concrete example where a large number of samples are generated for the same model (using the same prior noise for all of them) or through the same prompt? I don’t find your answer convincing. You provide references to the URL of popular models. But where is the evidence for this use case of generating multiple images (more than 4) for one prompt?
> >
> > My comment about the low n-chain regime was to assess the time overhead of your method for a small number of generations. I expect this overhead ist quite substantial but you never analyzed it. You provided again only FID numbers.

---

> > > ### Author Response · Authors · 2025-11-27
> > >
> > > ### 0. Clarification on Single Prior Sample vs. Emulate Diverse Prior Samples
> > >
> > > Before addressing specific questions, we would like to correct a potential misunderstanding regarding our sampling mechanism. The reviewer may have perceived our method as simply generating multiple outputs from a fixed, single prior sample. However, the core of our methodology is to **Emulate multiple reverse diffusion chains originating from diverse prior samples** via the Metropolis-Adjusted Langevin Algorithm (MALA) chain, even though the computation initiates from a single point.
> > >
> > > - **Distinction from Deterministic Sampling**: If deterministic sampling (e.g., DDIM) is performed starting from a single prior, the resulting trajectory is fixed, leading to identical outputs. In this scenario, generating multiple samples is impossible.
> > >
> > > - **Emulate Diverse Trajectories**: In contrast, our method utilizes the MALA chain to introduce stochastic exploration during the intermediate steps. This process allows a single chain to bifurcate and traverse various regions of the marginal distribution. These explored states are then independently denoised to the final data space. Consequently, **the resulting samples effectively represent distinct reverse diffusion trajectories that would have been obtained from different prior samples.**
> > >
> > > - **Effective Diversity**: Therefore, while we start from a single prior for initialization, the MALA chain acts as a mechanism to emulate the diversity of a multi-prior system. This is empirically validated by the enhanced mode coverage (Fig. 4) and improved recall (Tab. 5), confirming that our method does not merely produce local variations of a single sample but effectively explores the diverse modes of the distribution.
> > >
> > > - **Related Approaches**: [1] is a representative work that improves sampling throughput by performing Gibbs sampling on the joint time–data distribution, starting from an initial sample generated from a single prior. Although initialized from a single point, the sampling process allows the resulting samples to behave as if they originated from different prior distributions. Furthermore, recent frameworks like [2] also provide an example where a significant number of generated samples share a single prior sample due to reward-driven resampling.
> > >
> > >
> > > [1] [ICML 2023 Oral] Denosing MCMC for Accelerating Diffusion based Generative models
> > > [2] [ICML 2025] A General Framework for Inference-time Scaling and Steering of Diffusion Models
> > >
> > > ---
> > >
> > > ### 1-1. Generating various samples from the same model through the same prompt
> > > We appreciate the reviewer’s comment regarding the practical use cases. To further support the relevance of generating multiple images from a single prompt, we would like to highlight that this is a standard practice in both commercial services and academic benchmarks.
> > >
> > > - A representative and concrete example is Grok AI, which directly provides a service that generates dozens of images from a single prompt.(https://grok.com/imagine)
> > > - Similarly, Flux also offers a service that produces 10 images per prompt (https://getimg.ai/image-generator)
> > > - In addition, widely used consumer-facing services such as Bing Image Creator (https://www.bing.com/images/create), Leonardo AI (https://app.leonardo.ai/image-generation), Midjourney (https://www.midjourney.com) generate four images per prompt by default.
> > >
> > > Producing multiple samples per prompt is highly beneficial from the end-user’s perspective, since a single generation often does not perfectly match the intended concept.
> > >
> > > Furthermore, the widely adopted text-to-image evaluation benchmark GenEval[3] also need four generated images per prompt. The authors justify this design choice as follows:
> > > - [Section 5.1] For each prompt, we generate 4 images, and the GenEval score is averaged over all generated images. This choice is motivated by current text-to-image APIs like DALL-E 2 and Midjourney, which generate 4 images for a prompt to allow the user more choice.
> > >
> > > GenEval is a trusted benchmark used even in the evaluation of leading models such as Stable Diffusion 3[4].
> > >
> > > Our method aims to correct errors inherent in diffusion models while simultaneously enabling faster sampling. In other words, we address a realistic and practically important problem, the same problem that modern text-to-image services must solve in real deployments.
> > > We hope this clarifies the reviewer’s concern.
> > >
> > >
> > > [3] [NeurIPS 2023] GENEVAL: An Object-Focused Framework for Evaluating Text-to-Image Alignment
> > > [4] [ICML 2024 Oral] Scaling Rectified Flow Transformers for High-Resolution Image Synthesis
> > >
> > > ---

---

> ### Author Response · Authors · 2025-11-27
>
> ### 1-2. Necessity of large-scale generation for downstream tasks
>
> Beyond commercial applications, generating a large number of samples (far exceeding 4) from the same model/prompt is a critical requirement in research for downstream tasks.
>
> - **RLHF Fine-tuning**: Generating a large batch of samples from a single prompt plays a crucial role in Reinforcement Learning from Human Feedback (RLHF) for diffusion models. For instance, [5] generates **64 images per prompt** to effectively estimate rewards and gradients during the fine-tuning process. (Section 5.1)
>
> - **Data Augmentation**: Large-scale generation from a single prompt is essential for image augmentation in downstream tasks. [6] employs textual inversion to derive a single prompt from a target image and subsequently generates **10 to 50 augmented samples from that specific prompt** to improve robustness. (Appendix G)
>
> - **Synthetic Data for Classification**: [7] generates 1.2M to 12M synthetic images to improve ImageNet classification accuracy. (Section 4.3) Crucially, the authors fix the prompts to be simple one- or two-word class names, thereby generating **a massive number of diverse images from a single, fixed prompt per class**.
>
>
> [5] [ICLR 2025] HERO: Human-Feedback-Efficient Reinforcement Learning for Online Diffusion Model Fine Tuning
> [6] [ICLR 2024] Effective Data Augmentation with Diffusion Models
> [7] [TMLR 2023] Synthetic Data from Diffusion Models Improves ImageNet Classification
>
> ---
>
> ### 2. Small number of generation with small $n_{\text{chain}}$
>
> We thank the reviewer for pointing out the trade-offs in the low $n_{\text{chain}}$ regime. We fully acknowledge that the MH proposal-rejection mechanism introduces computational overhead, which can impact acceleration when $n_{\text{chain}}$ is small
>
> -  As noted in our previous rebuttal (Q.4) that when $n_{\text{chain}}=1$, the method exhibits no acceleration gain and instead leads to an increase in NFE due to MH step.
> - For completeness, we also include results for the **small-sample, small $n_{\text{chain}}$ regime**, whether the method indeed incurs extra cost and does not achieve acceleration. We demonstrate this effect quantitatively in the table below.
>     - **Setup**: We generated 4 images for text-to-image generation, and 10 images for CIFAR-10 generation. Where $n_{\text{chain}}$ is the length of MALA chain (the number of images generated for a single prior sample). For instance, generating 10 images when $n_{\text{chain}}$ is 3 entails generating 3 images across three prior sampling rounds, and 1 image in the final round. We set the batch size to 1 for both experiments.
>     - **Observation**: Under similar NFE budgets, our method requires more wall-clock time due to the additional discriminator overhead.
>     - **Result**: However, our method can still achieve wall-clock acceleration even in the low–$n_{\text{chain}}$ regime (e.g., $n_{\text{chain}} \geq 4$ for SD-v1.5 and $n_{\text{chain}} \geq 3$ for CIFAR-10 ).
>
>
>
> | SD-v1.5  | n_chain | Mean NFE | Total Time |Generated images |
> |----------|---------|--------- |------------|--------------  |
> | DDIM     | -       | 50       | 34 second  | 4              |
> | + AC     |1        | 58.25    | 48 second  | 4              |
> | + AC     |4        | 42.25    | **31 second**  | 4              |
>
>
> | CIFAR 10 | n_chain | Mean NFE | Total Time | Generated images |
> |-          |  -      |-        |-           |  -              |
> |Heun       |  -      |35       |6.82 second|  10            |
> |+ AC       |  1      |37.14    |7.44 second|  10            |
> |+ AC       |  2      |35.89    |7.13 second|  10            |
> |+ AC       |  3      |33.98    |**6.81 second**|  10            |
> |+ AC       |  5      |32.52    |**6.47 second**|  10            |

---

### Author Response · Authors · 2025-11-30

First and foremost, we would like to thank the reviewers for dedicating their valuable time and effort to reviewing our paper.

Our research focuses on improving throughput by accelerating and correcting the process of generating multiple images. This work is significant for the following reasons:

**Addresses Real-world Service Needs**: Commercial services such as [1,2,3] display diverse images to provide users with options. This creates a demand for generating multiple images quickly and accurately, which is the precisely the problem our method targets. We also confirmed the applicability of our method by demonstrating its effectiveness in text-to-image generation experiments.

**Applicability to Downstream Tasks**: Fast and accurate multi-image generation is a critical requirement in various research domains, including RLHF [5], Data Augmentation [6], and Classification [7].

[4] is a representative work that improves sampling throughput by performing Gibbs sampling on the joint time–data distribution. We identified the theoretical limitations of this approach and demonstrated that combining it with our method yields experimental improvements.

Our method distinguishes itself from existing approaches in the following ways:

**Resolving Intrinsic Model Errors**: Unlike existing acceleration methods that focus solely on discretization errors during sampling, our method addresses the intrinsic errors of the model itself, as demonstrated theoretically.

**Efficiency in Correction**  : Conventional correction methods often introduce additional computational burdens, limiting their applicability in real-world services. In contrast, our method achieves faster and more accurate sampling than base sampling methods, even while performing correction.

**Orthogonality and Compatibility**: Our approach is orthogonal to existing methods. We have experimentally demonstrated that it can be seamlessly combined with various acceleration and correction techniques. Additionally, we note that acceleration and correction is achievable even with a small number of sample generation situation (e.g., 3–4 samples).



The strengths of our work, as highlighted by the reviewers, are as follows:

- The interesting idea of using Metropolis-Hastings (MH) for acceleration (MQvF)
- The clever decomposition and tractability of the acceptance probability (jfbK)
- Theoretical guarantees (jfbK, pPCw)
- Overall experimental gains (MQvF, jfvK, pPCw)
- The flexibility and adaptability of our method to be combined with various sampling approaches (jfbK, pPCw)


Based on the additional experiments conducted during the rebuttal phase and the valuable suggestions from the reviewers, we have incorporated the following content into the main paper. These additions will be marked in blue during the Rebuttal period only.

- Emphasized that our study addresses critical real-world problems. [Introduction] (MQvF)

- Text-to-Image generation results have been added. [Section 5] (MQvF, jfbK, pPCw)

- Relevant related works suggested by the reviewers have been included in the literature review. [Section 2, Appendix G] (MQvF, jfbK, pPCw)

- Analysis on Discriminator overhead and various supplementary experiments on wall-clock time have been included. [Section 5, Appendix F.4] (pPCw, MQvF)

- A feature extractor ablation study has been added. [Appendix F.4] (pPCw)

- Small number generation result has been added. [Appendix F.4] (MQvF)

In addition to the points above, we have also made the following general revisions:

- Correction of minor typos and grammatical errors.

- Minor adjustments to the length and order of sections due to the newly added content.

We sincerely hope that these revisions resolve the concerns raised by the reviewers and that our work proves to be a valuable contribution to the ICLR community.

Sincerley,

Authors.

---

[1] https://grok.com/imagine
[2] https://getimg.ai/image-generator
[3] https://www.bing.com/images/create
[4] [ICML 2023] Denosing MCMC for Accelerating Diffusion based Generative models
[5] [ICLR 2025] HERO: Human-Feedback-Efficient Reinforcement Learning for Online Diffusion Model Fine Tuning
[6] [ICLR 2024] Effective Data Augmentation with Diffusion Models
[7] [TMLR 2023] Synthetic Data from Diffusion Models Improves ImageNet Classification

---

### Meta-Review · Area_Chair_9y7q · 2026-01-07

**Summary:**

Major issues raised include:
1) the efficiency improvement (and reduced NFEs) is not convincing, and need to clarify the setting as well, one sample or multiple per generation, and averaged NFEs rather than per-sample NFEs.

2) Theorem and connection with practical settings

3) More comparison with related works

4) The details of computing the compute overhead

5) potential mismatch between theory and practice.

**Reviewer Concerns:**

During the rebuttal, these questions mentioned above:

1) Addressed and clarified -- this work aims for improving the throughput, generating multiple images per time, useful in various settings. It follows the previous work, [ICML 2023] Denosing MCMC for Accelerating Diffusion based Generative models.

2) More comparison with related works

3) Explained and added experiments to validate

4) Clarified and explained with added experiments

5) Explained.

Overall, the authors have well handled all the main concerns.

**Reviewer Scores:**

The reviewers would raise the initial negative ratings to be positive as all their concerns have been addressed to a good degree. This paper presents a strong, comprehensive study for accelerating diffusion models under the batch generation setting, with good practical and theoretical value to the community.

---

### Decision · Program_Chairs · 2026-01-26

Accept (Poster)